# Towards Diverse Device Heterogeneous Federated Learning via Task Arithmetic Knowledge Integration

**Mahdi Morafah**[*1], **Vyacheslav Kungurtsev**[2], **Hojin Chang**[1], **Chen Chen**[3], **Bill Lin**[1]

[1]University of California San Diego (UCSD), [2]Czech Technical University in Prague,
[3]University of Central Florida (UCF)
[*]Correspondence: `mmorafah@ucsd.edu`

## Abstract

Federated Learning (FL) has emerged as a promising paradigm for collaborative machine learning, while preserving user data privacy. Despite its potential, standard FL algorithms lack support for diverse heterogeneous device prototypes, which vary significantly in model and dataset sizes—from small IoT devices to large workstations. This limitation is only partially addressed by existing knowledge distillation (KD) techniques, which often fail to transfer knowledge effectively across a broad spectrum of device prototypes with varied capabilities. This failure primarily stems from two issues: the dilution of informative logits from more capable devices by those from less capable ones, and the use of a single integrated logits as the distillation target across all devices, which neglects their individual learning capacities and and the unique contributions of each device. To address these challenges, we introduce TAKFL, a novel KD-based framework that treats the knowledge transfer from each device prototype's ensemble as a separate task, independently distilling each to preserve its unique contributions and avoid dilution. TAKFL also incorporates a KD-based self-regularization technique to mitigate the issues related to the noisy and unsupervised ensemble distillation process. To integrate the separately distilled knowledge, we introduce an adaptive *task arithmetic* knowledge integration process, allowing each student model to customize the knowledge integration for optimal performance. Additionally, we present theoretical results demonstrating the effectiveness of task arithmetic in transferring knowledge across heterogeneous device prototypes with varying capacities. Comprehensive evaluations of our method across both computer vision (CV) and natural language processing (NLP) tasks demonstrate that TAKFL achieves state-of-the-art results in a variety of datasets and settings, significantly outperforming existing KD-based methods. Our code is released at `https://github.com/MMorafah/TAKFL`.

## 1   Introduction

Federated Learning (FL) has rapidly gained traction as a promising approach to train machine learning models collaboratively across multiple devices, while preserving the privacy of user data. Standard federated learning methods, such as FedAvg [33], however, are primarily designed for unrealistic *device-homogeneous* scenarios, where all devices are assumed to have identical compute resource and can train the same neural network architecture [28, 33, 49, 21, 27, 48, 31]. Therefore, standard FL cannot support the participation of *heterogeneous devices*, all of which could significantly contribute to model training due to their unique and invaluable local datasets. To address this gap, knowledge distillation (KD) techniques have emerged as a promising approach to establish federation among heterogeneous device prototypes and facilitate knowledge transfer between them. In this approach, locally updated client models from different device prototypes, collectively termed as ensembles, serve as teachers to distill their knowledge into each device prototype's server student model using an unlabeled public dataset.

---

Project Website: `https://mmorafah.github.io/takflpage`.

38th Conference on Neural Information Processing Systems (NeurIPS 2024).

Despite their success, however, existing KD-based methods for device heterogeneous FL are primarily designed for scenarios where device prototypes are in the same-size with similar capabilities, i.e. same model and dataset sizes. However, in practice, *device capabilities vary widely*, ranging from small devices like IoTs with small models and small datasets to large devices like workstations with large models and large datasets. This diversity, often overlooked in the existing literature, results in device prototypes with varying strengths and information qualities. Unfortunately, existing methods struggle to establish effective knowledge transfer in these challenging, real-world device heterogeneous settings, primarily due to two reasons: ① Existing methods often disregard the individual strengths and information quality of each device prototype's ensembles and integrate their logits into a single distillation target. *This approach dilutes the richer, more informative logits from larger, more capable devices with less informative logits from smaller, less capable ones.* ② Additionally, these methods employ this single integrated distillation target to transfer knowledge across all different size student models. *This one-size-fits-all approach fails to provide customized knowledge integration based on the unique learning capacities of each student and the specific helpfulness of each device prototype's ensembles.*

Moreover, the heterogeneous ensemble distillation process can inadvertently lead student models into erroneous learning directions, causing them to forget their self-knowledge acquired through averaged locally updated parameters. This issue arises primarily due to two reasons: ① The distillation process introduces noise, *as the ensembles' logits are inferred on an unfamiliar public dataset*, distinct from their original training data. Additionally, the presence of data heterogeneity and the insufficient training of some ensembles, due to computational constraints, can further exacerbate this noise. ② The distillation process *lacks supervision from the actual private datasets, which are the ultimate learning objectives.* Consequently, these factors, combined with the limitations outlined earlier, result in *suboptimal knowledge transfer* in device heterogeneous settings. This underscores the urgent need for a more effective knowledge transfer framework.

In this paper, we introduce TAKFL, a novel *"**T**ask **A**rithmetic **K**nowledge Transfer Integration for **F**ederated **L**earning"* framework, designed to overcome the fundamental limitations in the existing methods and improve knowledge transfer in scenarios where device prototypes vary in size—both model and dataset—and consequently, in strength. TAKFL treats knowledge transfer from each device prototype's ensembles as separate tasks, distilling them independently to ensure that each prototype's unique contributions are accurately distilled without interference. To tackle the challenges associated with noisy and unsupervised ensemble distillation, we incorporate a KD-based self-regularization technique into this individual knowledge transfer process. Subsequently, to selectively integrate the separately distilled knowledge from heterogeneous prototypes' ensembles, we introduce an adaptive *task arithmetic* knowledge integration method by extending the notion of task vectors from centralized learning to federated learning. Our approach enables the student model to strategically customize the knowledge integration process based on the quality of knowledge from each prototype's ensembles and its intrinsic capacity, aiming to achieve optimal performance. We present theoretical results, grounded on the established theoretical learning properties of over-parametrized neural networks, that conceptualize knowledge distillation as the allocation of device prototypes' capacities to accurately fit the chosen logits. These results demonstrate the advantages of employing task arithmetic for knowledge transfer in terms of overall accuracy, coverage, and efficiency, as well as the adaptive knowledge integration based on the capacity of the student prototype. Furthermore, we comprehensively evaluate our method across both computer vision (CV) and natural language processing (NLP) tasks, utilizing various datasets and architectures, and demonstrate that TAKFL consistently achieves state-of-the-art (SOTA) performance.

The contribution of our paper is as follows:
1. We formalize and review the important considerations of the problem statement of federated learning with heterogeneous device prototypes.
2. We introduce TAKFL, a novel KD-based method designed to overcome the fundamental limitations of existing approaches, effectively facilitating knowledge transfer across diverse heterogeneous device prototypes with varying capabilities.
3. We present the first theoretical model for device heterogeneous KD, and demonstrate the effectiveness and efficiency of TAKFL compared to the standard alternatives that do not adapt to the student's self-knowledge quality and available learning capacity.
4. Our comprehensive experimental evaluations on both CV and NLP tasks, spanning various datasets and architectures, reveal that TAKFL consistently achieves SOTA performance, outperforming existing KD-based methods.

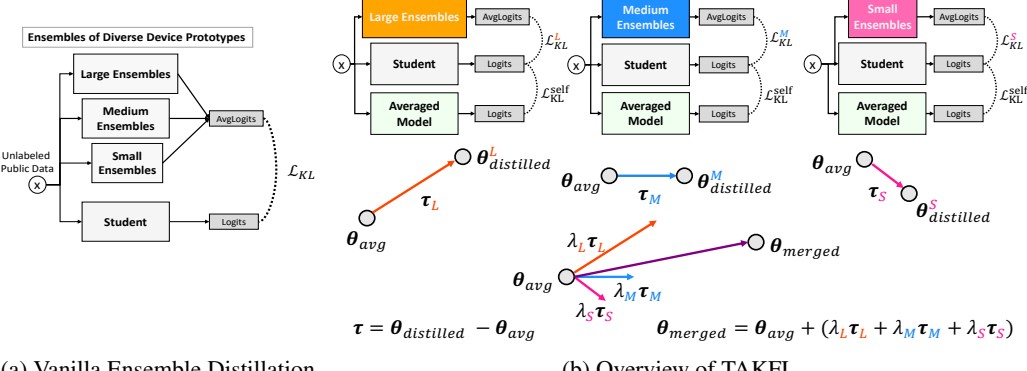

(a) Vanilla Ensemble Distillation        (b) Overview of TAKFL

Figure 1: **Overview of our approach and its distinction from prior works.** (a) This figure illustrates the vanilla ensemble distillation process, where logits from ensembles of various sizes are averaged and used as the distillation target across all prototypes. This approach leads to the dilution of information and suboptimal knowledge transfer (refer to Sections 6 and 7 for details). (b) This figure depicts our approach, TAKFL, which treats knowledge transfer from each prototype's ensemble as a separate task and distills them independently. Additionally, a KD-based self-regularization technique is introduced to mitigate issues related to the noisy and unsupervised ensemble distillation. Finally, the heterogeneously distilled knowledge is strategically integrated using an adaptive task arithmetic operation, allowing for customized knowledge integration based on each student prototype's needs.

## 2 Related Works

**Device Heterogeneous FL.** Prior works on device heterogeneous FL have considered two distinct approaches with different objectives and settings. The first array of studies focuses on accommodating devices with varying compute resources, aiming to train a single global model. Techniques such as static and rolling-based partial model training allow devices to train a sub-model of the global model tailored to their compute resources [11, 18, 3, 1]. However, this approach does not fully reflect real-world scenarios. In practice, device prototypes such as IoTs and smartphones have unique neural network architectures designed for their specific configurations and underlying tasks, which may not support training varying neural architectures. This highlights a significant limitation in accommodating the full spectrum of device heterogeneity in this approach. The second array of studies addresses a more practical scenario where device prototypes with heterogeneous model architectures participate in FL to enhance their global model performance through mutual knowledge sharing [30, 43, 6]. In this context, KD techniques are used to transfer knowledge among prototypes, where locally updated client models, termed as ensembles, serve as teachers to distill their knowledge into each server's student model using an unlabeled public dataset. For example, FedDF [30] uses vanilla logit averaging, while Fed-ET [6] applies an uncertainty-weighted logit averaging, enhanced by a diversity regularization technique. *However, existing works typically focus on settings where prototypes have similar capabilities—both model and dataset sizes—and thus neglecting the challenges in more diverse settings with varying capabilities. This oversight leaves their effectiveness in such settings largely unexplored. In this paper, we aim to study the underexplored diverse heterogeneous device settings.* See Appendix A for a more detailed discussion on the related works.

**Model Editing via Task Arithmetic.** Traditional methods for model editing often involve expensive joint fine-tuning across multiple tasks, which can limit scalability and democratization [62]. Recently, a promising technique called task arithmetic has emerged as a cost-effective and scalable method for updating pre-trained models with new information or refining undesired behavior [53, 37, 32]. The concept of "task vectors" introduced by Wortsman et al. [53] plays a pivotal role in these techniques. For any given task $t$, a task vector is derived by subtracting the model's pre-trained weights $\boldsymbol{\theta}_{pre}$ from its fine-tuned weights $\boldsymbol{\theta}_{ft}^{t}$ on task $t$, i.e. $\boldsymbol{\tau}_t = \boldsymbol{\theta}_{ft} - \boldsymbol{\theta}_{pre}$. These task vectors act as unique representations for specific tasks. Furthermore, researchers have demonstrated that by summing multiple task vectors $\{\boldsymbol{\tau}_t\}_{t=1}^{T}$, and integrating them into a pre-trained model via $\boldsymbol{\theta} = \boldsymbol{\theta}_{pre} + \lambda \sum_{t=1}^{T} \boldsymbol{\tau}_t$, one can effectively create a model capable of handling multiple tasks [53, 57]. *To the best of our knowledge, this work is the first to extend the notion of task vectors to the federated learning setting, introducing a task arithmetic for knowledge distillation across diverse heterogeneous device prototypes.*

# 3  Problem Statement: FL with Heterogeneous Device Prototypes

Consider a cross-device FL setup with a set of $M$ distinct device prototypes $\mathbb{M}$, i.e., $M = |\mathbb{M}|$. Each device prototype $m_j \in \mathbb{M}$ has a distinct neural network architecture $f^j(\cdot; \boldsymbol{\theta}^j)$ parameterized by $\boldsymbol{\theta}^j \in \mathbb{R}^{n_j}$ and a set of clients $\mathbb{C}^j$, with $N^j = |\mathbb{C}^j|$ clients in total. Each client $c_k \in \mathbb{C}^j$ has a local private dataset $\mathbb{D}_k^j = \{(\boldsymbol{x}_i, y_i)\}_{i=1}^{n_{j,k}}$, where $n_{j,k} = |\mathbb{D}_k^j|$, and locally trains the parameters $\boldsymbol{\theta}^j$ of the neural network architecture $f^j$ on its local dataset. Furthermore, denote $\mathbb{D}^j = \cup_{k \in \mathbb{C}^j} \mathbb{D}_k^j$ to be the union of the private datasets for device prototype $j$. We assume $\mathbb{D}^j \sim \mathcal{D}^j$, that is a subsample from the population distribution $\mathcal{D}^j$ and similarly $\mathbb{D}_k^j \sim \mathcal{D}_k^j$. The union of the private datasets, i.e. $\mathbb{D} = \bigcup_{j \in \mathcal{M}} \mathbb{D}^j$, is sampled from the entire population $\mathcal{D}$, which is defined as an unknown mixture of the distributions each device prototype sampled its data from, i.e. generically non-i.i.d. We formalize this as a mixture of local clients data population, i.e., $\mathcal{D} = \sum_j \omega_{j,:} \mathcal{D}^j = \sum_j \sum_k \omega_{j,k} \mathcal{D}_k^j$, where $0 \leq \omega_{j,k} \leq 1$ and $\sum_{jk} \omega_{j,k} = 1$, and $\omega_{j,k}$ is unknown.

The ultimate objective is to minimize the test error and thus enable accurate inference for each device prototype $j$, aiming to obtain the optimal parameters for the population dataset:

$$\operatorname*{argmin}_{\boldsymbol{\theta}^j} \mathbb{E}_{(\boldsymbol{x},y) \sim \mathcal{D}}[\ell(f^j(\boldsymbol{x}; \boldsymbol{\theta}^j), y)] = \operatorname*{argmin}_{\boldsymbol{\theta}^j} \sum_{j=1}^{M} \sum_{k=1}^{N^j} \omega_{i,k} \mathbb{E}_{(\boldsymbol{x},y) \sim \mathcal{D}_k^j}[\ell(f^j(\boldsymbol{x}; \boldsymbol{\theta}^j), y)] \quad (1)$$

where $\ell(\cdot, \cdot)$ is the sample-wise loss function (e.g. cross entropy for image classification) and we decompose by total population loss with the linearity of expectation in the mixture. See Fig 4b for a visual illustration of heterogeneous device prototype FL.

# 4  Background: Federated Ensemble Distillation

To address the limitations of standard FL in device heterogeneous settings, Lin et al. [30] proposed ensemble knowledge distillation to transfer knowledge between heterogeneous device prototypes in FL. This procedure consists of two stages: (1) local per-prototype FL, and (2) server-side vanilla ensemble distillation. The details of each stage discussed in the following paragraphs.

**Local Per-Prototype FL.** In this context, at each round $r$ a subset of clients $\mathbb{C}_r^j$ from each device prototype $j \in \mathbb{M}$ is randomly selected by the server and download their corresponding model initialization $\boldsymbol{\theta}_r^j$. Each client $c_k^j \in \mathbb{C}_r^j$, starting from this model initialization, locally train the model $f^j$ on its local private data $\mathbb{D}_k^j$ by taking multiple steps of stochastic gradient descent. Then, they send back their updated parameters $\{\widehat{\boldsymbol{\theta}}_k^j\}_{k \in \mathbb{C}_r^j}$ to the server. The server aggregates the received clients parameters, and computes $\boldsymbol{\theta}_{avg}^j = \sum_{k \in \mathbb{C}_r^j} \frac{|\mathbb{D}_k|}{\sum_{k \in \mathbb{C}_r^j} |\mathbb{D}_k|} \widehat{\boldsymbol{\theta}}_k^j$. In classic federated learning formalism, the parameters $\boldsymbol{\theta}_{avg}^j$ satisfy,

$$\boldsymbol{\theta}_{avg}^j \in \operatorname*{argmin}_{\boldsymbol{\theta}^j} \sum_{k=1}^{N^j} \mathbb{E}_{(\boldsymbol{x},y) \sim \mathbb{D}_k^j} [\ell(f^j(\boldsymbol{x}; \boldsymbol{\theta}^j), y)] \quad (2)$$

**Vanilla Ensemble Distillation.** In this stage, each server model $f^j$ gets initialized with $\boldsymbol{\theta}^j$, and undergoes updates using ensemble knowledge distillation. Here, heterogeneous client models from heterogeneous device prototypes, collectively termed as ensembles, serve as teachers, i.e. $\mathcal{T} := \{f^i(\cdot, \widehat{\boldsymbol{\theta}}_k^i) | i \in \mathbb{M}, k \in \mathbb{C}^i\}$, transferring their knowledge to each server student model, i.e. $\mathcal{S}_i := f^i(\cdot, \boldsymbol{\theta}^i)$. For simplicity, we drop the index for each server student model, denoting it as $\mathcal{S}$. The ensemble distillation loss using a mini-batch of data from an unlabeled public dataset, i.e $\boldsymbol{x} \in \mathbb{D}^{public}$, can be defined by the following equation:

$$\mathcal{L}_{\text{ED}} = \text{KL}\left[\sigma\left(\frac{1}{|\mathcal{T}|} \sum_{\mathcal{F} \in \mathcal{T}} \mathcal{F}(\boldsymbol{x})\right), \sigma(\mathcal{S}(\boldsymbol{x}))\right], \qquad \text{(AvgLogits)} \quad (3)$$

where $\sigma(\cdot)$ is the softmax function. As illustrated in Eq. 3, vanilla ensemble distillation treats all heterogeneous device prototypes' ensembles equally by uniformly averaging their logits. This way of knowledge integration overlooks the individual strengths and informational value of each prototype's ensembles. As a result, the richer, more informative logits from stronger ensembles are diluted by less informative logits from weaker ensembles, leading to information loss. Furthermore,

this averaged logits is used as the distillation target across different-sized student models, irrespective of their intrinsic capacity and the helpfulness of each prototype's ensembles. Consequently, this leads to suboptimal knowledge transfer in device heterogeneous FL. See Section 6 for theoretical analysis and Section 7 for experimental observations.

## 5 Task Arithmetic Knowledge Transfer and Integration

In this section, we introduce TAKFL, designed to overcome the fundamental limitations of previous approaches and enhance knowledge transfer across diverse heterogeneous device prototypes, which vary in size—in terms of both model and dataset size. TAKFL consists of two main components: (1) individually transferring knowledge from each device prototype's ensembles, and (2) adaptively integrating knowledge via task arithmetic. Detailed descriptions of each component are provided in Section 5.1 and 5.2, respectively. An illustrative overview of TAKFL is presented in Figure 1b, and the full algorithm is detailed in Appendix B, Algorithm 1.

### 5.1 Knowledge Transfer from Individual Device Prototype

We begin by discussing our proposed knowledge transfer framework from each individual device prototype's ensembles. This process consists of two main components: ensemble knowledge transfer and self-regularization, each detailed in the subsequent paragraphs.

**Ensemble Knowledge Transfer.** Vanilla ensemble distillation integrates the knowledge of varying strength ensembles by uniformly averaging their logits. This approach can potentially transform or even degrade the overall quality of the knowledge being transferred, leading to suboptimal knowledge transfer. To effectively distill the unique knowledge and contributions of each prototype's ensembles, and to avoid dilution, information loss, and interference from other prototypes' ensembles, we propose transferring the knowledge from each prototype's ensembles separately and independently.

Specifically, let's consider $\mathcal{T}_i := \{f^i(\cdot, \widehat{\boldsymbol{\theta}}_k^i) \mid k \in \mathbb{C}^i\}$ denotes the ensembles of device prototype $i$ as teacher and $\mathcal{S}_j$ denotes the server student model of the device prototype $j$. Without loss of generality, we refer to each device prototype's server student model as just student denoted as $\mathcal{S}$. Therefore, the knowledge distillation loss between the teacher ensembles $\mathcal{T}_i$ and server student $\mathcal{S}$ ($\mathcal{T}_i \rightarrow \mathcal{S}$) is defined below:

$$\mathcal{L}_{KD}^{\mathcal{T}_i \rightarrow \mathcal{S}} = \text{KL}\left[\sigma\left(\frac{1}{|\mathcal{T}_i|}\sum_{\mathcal{F} \in \mathcal{T}_i}\mathcal{F}(\boldsymbol{x})\right), \, \sigma\left(\mathcal{S}(\boldsymbol{x})\right)\right]. \tag{4}$$

**Scaffolding Student from Noisy Ensemble Distillation.** The ensemble distillation process may adversely impact the student, causing it to forget its own knowledge acquired through averaged locally updated parameters and be drifted into erroneous directions. This is primarily due to two key factors: (1) The ensemble distillation process introduces noise, mainly because the ensembles' logits are inferred on an unfamiliar public dataset they have not been trained on. These ensembles are originally trained on local private datasets, which usually differ from the unlabeled public dataset used for distillation. Moreover, other factors such as the presence of data heterogeneity within FL and insufficient training of some ensembles due to limited computational resources can exacerbate this noise, particularly in the initial rounds of federation. (2) The ensemble distillation process lacks supervision from the actual private datasets, which is the ultimate learning objective.

To scaffold the student models from the noisy and unsupervised distillation process, which may cause them to drift into erroneous directions and forget their invaluable self-knowledge, we introduce a KD-based self-regularization technique. Our self-regularization technique mitigates these issues by enforcing similarity between the logits of the student and its initial logits (when the student is initialized with averaged parameters) using KL divergence loss defined below:

$$\mathcal{L}_{\mathcal{S}}^{\text{self}} = \text{KL}\left[\sigma\left(\mathcal{S}(\boldsymbol{x}; \boldsymbol{\theta}_{avg})\right), \, \sigma\left(\mathcal{S}(\boldsymbol{x})\right)\right]. \tag{5}$$

**Overall Knowledge Transfer Objective.** The overall knowledge transfer objective from teacher ensembles $\mathcal{T}_i$ of device prototype $i$ to the student $\mathcal{S}$ is the combination of the ensemble knowledge distillation loss $\mathcal{L}_{KD}^{\mathcal{T}_i \rightarrow \mathcal{S}}$ (Eq. 4) and the self-regularization loss $\mathcal{L}_{\mathcal{S}}^{\text{self}}$ (Eq. 5) defined in the following:

$$\mathcal{L}_{\mathcal{S}}^{\mathcal{T}_i} = \mathcal{L}_{KD}^{\mathcal{T}_i \rightarrow \mathcal{S}} + \gamma \cdot \mathcal{L}_{\mathcal{S}}^{\text{self}}. \tag{6}$$

Here, $\gamma$ is a hyperparameter controlling the effect of self-regularization term. We associate the knowledge transfer from each device prototype $i$ to a task $T_i$ with the loss $\mathcal{L}_{\mathcal{S}}^{\mathcal{T}_i}$.

## 5.2 Task Arithmetic Knowledge Integration

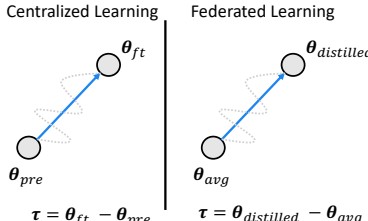

Figure 2: **Analogy between task vector in centralized learning and federated learning.**

Herein, we delve into the details of our proposed method for customized integration of the separately distilled knowledge from heterogeneous ensembles. Drawing inspiration from recent advances in model editing via task arithmetic [53], where a pre-trained model's knowledge can be edited via task-specific vectors using arithmetic operation, we propose a novel customizable knowledge integration method via task arithmetic. To do so we extend the notion of task vector from centralized learning to federated learning. We conceptualize the averaged locally updated parameters, i.e. $\boldsymbol{\theta}_{avg}$, as a "pre-trained", similar to those in centralized learning, and the parameters of the distilled model via knowledge transfer objective (Eq. 4), denoted as $\boldsymbol{\theta}_{distilled}$, as a "fine-tuned" version of the model (see Fig. 2). Consequently, the task vector $\boldsymbol{\tau}_i$ associated with the knowledge transfer task $\mathcal{L}_{\mathcal{S}}^{\mathcal{T}_i}$ can be defined by subtracting the distilled parameters from the averaged locally updated parameters as follows:

$$\boldsymbol{\tau}_i = \boldsymbol{\theta}_{distilled}^{\mathcal{T}_i \to \mathcal{S}} - \boldsymbol{\theta}_{avg}. \tag{7}$$

Essentially, task vectors serve as unique representations for the transferred knowledge from each prototype's ensembles to the student and encapsulate the distinct contributions of each prototype's ensembles to the student model. To selectively merge the knowledge of each prototype' ensembles into the student, we employ an adaptive task arithmetic operation as follows:

$$\boldsymbol{\theta}_{merged} = \boldsymbol{\theta}_{avg} + \sum_{i \in \mathbb{M}} \lambda_i \boldsymbol{\tau}_i, \tag{8}$$

where $\lambda_i$ denotes the merging coefficient associated with task vector $\boldsymbol{\tau}_i$, and they sum to one, i.e. $\sum_{i \in \mathbb{M}} \lambda_i = 1$. The merging coefficients determine the extent of knowledge integration from each prototype's ensembles. Essentially, they enable the student to have customized knowledge integration to achieve maximum performance. The student can determine these merging coefficients based on its own learning capacity and the relative knowledge and helpfulness of other device prototypes' ensembles. This approach provides an effective, low-cost, and scalable knowledge integration strategy in settings with diverse device heterogeneity. In our experiments, we considered this as a hyperparameter and tuned it manually or determined it using held-out validation sets which achieves similar results. More details can be found in Appendix F.3.

## 6 Theoretical Results

We present a theoretical understanding on the efficacy of knowledge distillation in device heterogeneous FL. We argue that vanilla ensemble distillation (VED) diffuses the information from logits, which presents a notable disadvantage for solving (1). This effect is particularly pronounced when the teacher ensembles are from a device prototype of small capacity, and the student model is from a device prototype of large capacity. By contrast, our proposed method of task arithmetic knowledge integration, mitigates the drawbacks of VED and is able to simultaneously incorporate information from differently sized heterogeneous ensembles, efficiently filling up the capacity of each student with the most informative knowledge, achieving optimal knowledge transfer.

**Assumptions and Preliminaries.** Standard practice, including the setting in consideration as well as the numerical experiments here, involves *overparametrized* neural networks, that is, the total number of weights far exceeds the training sample size. This implies that the set of weights that minimize the loss is non-unique, and moreover, it has been argued that they form a submanifold [8]. This submanifold structure of solution sets will provide the critical source of understanding the subsequent results. In particular, we shall consider knowledge distillation as filling up the capacity of device prototypes' models with basis vectors corresponding to submanifolds that minimize as many device prototypes' data distributions as possible.

Since we are interested in server-side distillation across heterogeneous device prototypes, we assume optimal conditions at the local per-prototype FL level, meaning that the perfect solution for local per-prototype FL is achieved. The formal details of the assumptions and statements are presented in Appendix C. In these Propositions we assume that the prototype $i$ and $j$ datasets are disjoint, i.e.

$\mathcal{D}^i \cap \mathcal{D}^j = \emptyset$, and $\bigcup_{i=1}^{M} \mathcal{D}^i = \mathcal{D}$. We show that the other cases are trivial and uninteresting in the appendix.

**Proposition 1.** *(information loss in VED, informal). Consider the VED procedure in the form of solving* (3). *Consider two device prototypes with a device capacity and solution dimension of $Q^1, Q^2$ and $W^1, W^2$, respectively, and with associated eigenbases $\mathcal{Q}^i, \mathcal{W}^i$. Denote $W^{i,j}, i, j = 1, 2$ as the capacity allocated by student $i$ in order to distill knowledge from teacher $j$'s logits.*

1. **Case 1:** *When the capacities are the same, that is $Q^1 = Q^2$ and $W^1 = W^2 = W^{1,2} = W^{2,1}$, then with VED, there will be some capacity, in the sense of eigenspace, of student prototypes that will be allocated with parameters that do not minimize the student's its own data distribution.*
2. **Case 2:** *Assume that $Q^1 > Q^2$ and $W^1 = W^{1,2} > W^2$. Then the phenomenon as for Case 1 holds. Moreover, there will be some capacity of student 1's model that will be allocated with parameters that do not minimize either of the teacher or student prototype's data distribution.*

An interesting key mechanism of the proof is that when VED is applied in distilling logits from a small device prototype to a large one, the modeling capacity of $W^{1,2}$ is structurally reduced to that of $W^2 < W^{1,2}$, i.e., it is an operation wasteful of the potential model capacity.

**Remark 1.** *This proposition proves that in general, VED is prone to diffuse knowledge already present in students, and leads to inefficient and inaccurate use of model capacity. Furthermore, under the case that device prototypes have different capacities, VED ends up leading to more erronous models entirely as the small information within the small teacher is transferred onto a larger capacity target.*

**Proposition 2.** *(improve knowledge transfer with task arithmetic, informal). Consider the TAKFL procedure as in the form of computing* (8). *Consider two device prototypes with a device capacity and solution dimension of $Q^1, Q^2$ and $W^1, W^2$, respectively, and with associated eigenbases $\mathcal{Q}^i, \mathcal{W}^i$.*

1. **Case 1:** *In the case that that $Q^1 \geq Q^2$ and $W^1 \geq W^2$, it holds that the TAKFL with prototype 1 as student preserves the eigenbasis associated to the parameters used to accurately fit the data $\mathcal{D}^1$.*
2. **Case 2:** *Assume that $Q^1 = Q^2$ and $W^1 = W^2$. TAKFL yields a solution for the student that is at the intersection of the subspaces corresponding to minimizing the two data distributions.*
3. **Case 3:** *Assume that $Q^1 > Q^2$ and $W^1 > W^2$. In the case of prototype 1 being the student, TAKFL yields a solution that:*
    (a) *retains the approximation accuracy on device 1's data distribution,*
    (b) *ensures approximation accuracy to the level of device 2's relative capacity*
    (c) *fills the remaining local capacity device 1 has allocated for device 2's logits with no informative new knowledge, unless enforced otherwise.*

**Remark 2.** *This proposition proves that in general, TAKFL promotes the most efficient allocation of the devices' capacity in order to accurately fit a diverse set of data distributions. With TAKFL, the previously acquired knowledge is entirely preserved. Even under the case that device prototypes have different capacities, TAKFL smartly transfers the most informative knowledge to each prototype's student model based on its own intrinsic capacity. Still, the final statement indicates that in the case that there are many different teachers, while a small device prototype serving as teacher will not be necessarily compromise information, it would still be preferable to allocate that capacity to a more informative, larger, teacher model.*

**Comments.** We comment on the complexity and convergence aspects of TAKFL briefly:

1. **Computation Time**: TAKFL's computation time is $O(1)$ (constant) due to parallelization, as all distillation processes occur simultaneously.
2. **Computation Load**: The overall computational load scales as $O(M)$ (linear) since the distillation tasks are performed independently for each prototype in parallel and merged into a singe task arithmetic operation (Eq. 8).
3. **Resource Usage (Memory)**: The resource usage scales as $O(M^2)$ (quadratic) because of the need to store and process multiple task vectors concurrently.
4. **Optimization Convergence Rate**: The convergence rate of the Knowledge Distillation procedure is as standard for training a nonconvex loss function, e.g. [42].

# 7 Experiments

## 7.1 Main Experimental Setup

**Dataset and Architecture.** We evaluate our method on computer vision (CV) and natural language processing (NLP) tasks. For CV, we train image classification using CIFAR10/100 [24], CINIC-10 [9], and TinyImagenet [25]. For NLP, we fine-tune pre-trained models for text classification on MNLI [52], SST-2 [45], MARC [22], and AG News [60]. Our architectures include ResNet [17], VGG [44], and ViT [12] for CV, and small BERT variants [47] (-Tiny, -Mini, -Small) for NLP. We simulate a federated non-i.i.d setting using a Dirichlet distribution $Dir(\alpha)$, where a lower $\alpha$ indicates higher heterogeneity [27, 36]. Further details can be found in Appendix F.1 and F.2.

**Implementation Details.** We use the Adam optimizer for both CV and NLP tasks. For CV, local training involves 20 epochs with a learning rate of 0.001, weight decay of 5e-5, and a batch size of 64. NLP training is conducted over 1 epoch with a learning rate of 3e-5, no weight decay, and a batch size of 32. For distillation, Adam is used with a learning rate of 1e-5 and weight decay of 5e-4 for CV, and 3e-5 with no weight decay for NLP. Batch sizes for distillation are 128 for CV and 32 for NLP. The softmax temperature is set at 3 for both tasks, with a temperature of 20 for self-regularization. Further details are provided in Appendix F.1 and F.2.

**Baselines and Evaluation Metric.** We compare our method against standard FL, i.e. FedAvg [33] and SOTA KD-based methods designed for heterogeneous device prototypes FL, including FedDF [30] and FedET [6]. The evaluation metric is the final top-1 classification accuracy of each device prototype's global model on the test dataset, as per the methodology described in [36]. We report the average results and the standard deviation over three independent runs, each with a different random seed.

*A more detailed version of the experiments, alongside additional experiments and ablation studies, is presented in Appendix D and E.*

## 7.2 Main Experimental Results

In this section, we evaluate the performance of our method, TAKFL, in a federated learning environment that mirrors real-world scenarios with diverse, heterogeneous device prototypes, as illustrated in Fig. 4b. Our experimental setup includes three different device prototype sizes: Small (S) with a small model and small dataset, Medium (M) with a medium-sized model and medium-sized dataset, and large (L) with a large model and large dataset.

**Performance on CV Task.** Table 1 presents the performance of TAKFL in the homo-family architecture setting on the CIFAR-10 and CIFAR-100 [24] datasets (for hetero-family architecture results, see Appendix D.1, Table 4). TAKFL consistently enhances performance across all device prototypes in various scenarios, achieving SOTA results. Notably, in the Dir(0.3) setting on CIFAR-10, TAKFL improves average performance across all prototypes by 8%, and by 4% on CIFAR-100. From Table 1, inconsistent performance improvements are observed with prior KD-based methods, especially for the L prototype. While S and M prototypes achieve gains, the L prototype suffers up to a 10% degradation compared to vanilla FedAvg, highlighting the dilution issue where valuable information from larger, more capable device prototypes is diluted by less informative outputs from smaller devices. Moreover, the significant performance improvements TAKFL achieves for each device prototype, particularly for S and M prototypes, illustrate the ineffectiveness of the one-size-fits-all approach used in the existing KD methods. These observations confirm the shortcomings of vanilla ensemble distillation and corroborate our theoretical findings in Remark 1 and 2. The effectiveness of our self-regularization technique is further supported by these experimental results. For more detailed and insightful analysis see Appendix D.1.1.

**Performance on NLP Task.** Table 2 presents the results on MNLI [52] and SST-2 [45] datasets (see Appendix D.3 for further experiments). Similar to the CV task, TAKFL has consistently improved performance across all device prototypes of varying sizes, achieving SOTA results: a 3% average increase on MNLI and 2% on SST-2. The suboptimality of existing KD methods, is evident from the results presented here as well. Notably, FedET suffers from a significant performance degradation compared to vanilla FedAvg. This issue stems from FedET's reliance on the confidence scores of neural networks for uncertainty estimates. However, neural networks, especially pretrained language

Table 1: **Performance Results for CV task on CIFAR-10 and CIFAR-100.** Training data is distributed among S, M, and L device prototypes in a 1:3:6 ratio, subdivided among clients using Dirichlet distribution. Public datasets are CIFAR-100 [24] for CIFAR-10 [24] and ImageNet-100 [10] for CIFAR-100. Client configurations include 100, 20, and 4 clients for S, M, and L, with sampling rates of 0.1, 0.2, and 0.5. Architectures are ResNet-8, ResNet-14, and ResNet-18 [17] for S, M and L, respectively. All models are trained from scratch for 60 rounds. See Appendix D.1 for additional experiments using hetero-family architecture and more details.

| Dataset | Baseline | Low Data Heterogeneity (Dir(0.3)) | | | | High Data Heterogeneity (Dir(0.1)) | | | |
|---|---|---|---|---|---|---|---|---|---|
| | | S | M | L | Average | S | M | L | Average |
| CIFAR-10 | FedAvg | $36.21_{\pm2.24}$ | $46.41_{\pm2.33}$ | $59.46_{\pm6.17}$ | 47.36 | $22.01_{\pm0.78}$ | $25.26_{\pm3.89}$ | $51.51_{\pm3.52}$ | 32.93 |
| | FedDF | $49.31_{\pm0.15}$ | $50.63_{\pm0.73}$ | $49.82_{\pm0.98}$ | 49.92 | $34.71_{\pm1.48}$ | $35.27_{\pm4.74}$ | $51.08_{\pm4.04}$ | 40.35 |
| | FedET | $49.21_{\pm0.72}$ | $55.01_{\pm1.81}$ | $53.60_{\pm6.47}$ | 52.60 | $29.58_{\pm3.00}$ | $30.96_{\pm4.70}$ | $45.53_{\pm6.46}$ | 35.36 |
| | TAKFL | $55.90_{\pm1.70}$ | $57.93_{\pm3.49}$ | $60.58_{\pm2.35}$ | 58.14 | $37.40_{\pm1.68}$ | $38.96_{\pm0.17}$ | $51.49_{\pm6.15}$ | 42.62 |
| | TAKFL+Reg | $\mathbf{56.37_{\pm0.46}}$ | $\mathbf{58.60_{\pm0.43}}$ | $\mathbf{65.69_{\pm1.28}}$ | **60.22** | $\mathbf{40.51_{\pm1.05}}$ | $\mathbf{40.12_{\pm1.24}}$ | $\mathbf{53.24_{\pm2.51}}$ | **44.62** |
| CIFAR-100 | FedAvg | $13.22_{\pm0.14}$ | $21.39_{\pm1.11}$ | $29.47_{\pm0.86}$ | 21.36 | $11.86_{\pm0.08}$ | $14.63_{\pm0.65}$ | $26.25_{\pm1.64}$ | 17.58 |
| | FedDF | $19.54_{\pm0.20}$ | $24.32_{\pm0.45}$ | $29.29_{\pm1.45}$ | 24.38 | $16.09_{\pm0.32}$ | $19.80_{\pm0.17}$ | $26.59_{\pm0.25}$ | 20.83 |
| | FedET | $19.67_{\pm0.35}$ | $25.27_{\pm0.66}$ | $31.10_{\pm1.53}$ | 25.35 | $11.18_{\pm1.68}$ | $18.22_{\pm0.35}$ | $26.40_{\pm0.65}$ | 18.60 |
| | TAKFL | $24.48_{\pm0.42}$ | $27.60_{\pm0.25}$ | $29.84_{\pm0.94}$ | 27.31 | $\mathbf{22.90_{\pm0.18}}$ | $23.63_{\pm0.72}$ | $26.98_{\pm0.13}$ | 24.50 |
| | TAKFL+Reg | $\mathbf{27.18_{\pm0.27}}$ | $\mathbf{29.14_{\pm0.20}}$ | $\mathbf{31.15_{\pm0.97}}$ | **29.15** | $22.88_{\pm0.37}$ | $\mathbf{23.92_{\pm0.57}}$ | $\mathbf{28.01_{\pm0.34}}$ | **24.94** |

Table 2: **Performance Results for NLP Task on MNLI and SST-2.** Training data distribution is similar to the CV task using only Dir(0.5) here. Public datasets are SNLI [2] for MNLI [52] and Sentiment140 [14] for SST-2 [45]. Client configurations are 8, 4, and 2 clients for S, M, and L, with sample rates of 0.3, 0.5, and 1.0, respectively. Architectures include Bert-Tiny, Bert-Mini, and Bert-Small [47] for S, M, and L, initialized from pre-trained parameters and fine-tuned for 20 communication rounds. See Appendix F.2 for more details.

| Baseline | MNLI | | | | SST-2 | | | |
|---|---|---|---|---|---|---|---|---|
| | S | M | L | Average | S | M | L | Average |
| FedAvg | $36.15_{\pm0.46}$ | $54.47_{\pm2.48}$ | $57.51_{\pm2.79}$ | 49.37 | $54.98_{\pm1.81}$ | $74.71_{\pm8.22}$ | $86.69_{\pm0.06}$ | 72.13 |
| FedDF | $54.21_{\pm0.15}$ | $60.44_{\pm1.91}$ | $66.71_{\pm1.09}$ | 60.45 | $74.41_{\pm2.62}$ | $80.71_{\pm1.63}$ | $84.35_{\pm1.66}$ | 79.82 |
| FedET | $48.03_{\pm6.32}$ | $50.33_{\pm7.87}$ | $53.80_{\pm6.18}$ | 50.72 | $66.63_{\pm9.14}$ | $65.89_{\pm16.35}$ | $70.05_{\pm15.83}$ | 67.52 |
| TAKFL | $57.43_{\pm0.21}$ | $63.58_{\pm0.31}$ | $68.74_{\pm0.12}$ | 63.25 | $74.73_{\pm0.55}$ | $82.17_{\pm0.31}$ | $86.93_{\pm0.42}$ | 81.28 |
| TAKFL+Reg | $\mathbf{57.61_{\pm0.89}}$ | $\mathbf{63.91_{\pm1.05}}$ | $\mathbf{68.96_{\pm1.10}}$ | **63.49** | $\mathbf{74.88_{\pm0.43}}$ | $\mathbf{82.40_{\pm0.83}}$ | $\mathbf{87.33_{\pm0.63}}$ | **81.54** |

models (PLMs), tend to be poorly calibrated and overconfident, undermining reliable uncertainty estimates [50, 15, 5, 55].

## 7.3 Scalability Evaluation

We evaluate the scalability of TAKFL across a spectrum of device prototypes, from extremely small (XXS) to extremely large (XXL), to see how well our method adapts from a uniform array of small-size prototypes to a diverse mix of sizes. Each prototype is equipped with appropriately scaled model and dataset sizes, simulating real-world variations in device capabilities.

Figure 3 illustrates TAKFL's ability to effectively scale from 3 to 7 device prototypes. In scenarios where all devices are similarly small, i.e. 3-device setup, TAKFL's performance is slightly better than FedDF. This is because when devices are homogeneously small and similar in capability, they

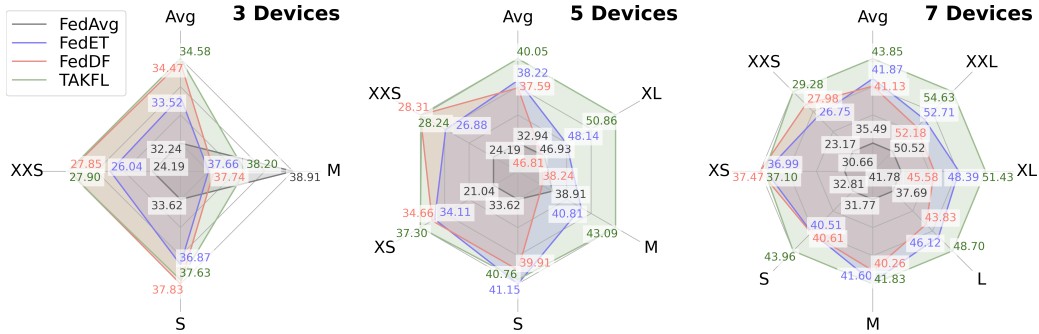

Figure 3: **Scalability Evaluation of TAKFL.** Image classification on CINIC-10 [9] dataset is used to evaluate TAKFL's scalability across device prototypes ranging from XXS to XXL. Training data is distributed among prototypes in a 1:2:3:4:5:6:7 ratio, further subdivided using Dir(0.5). Client configurations range from 35 for XXS to 5 for XXL. Architectures span from ResNet10-XXS for XXS to ResNet50 for XXL prototype, all initialized from scratch and trained over 30 communication rounds. The public dataset is CIFAR-100 [24]. See Appendix D.4 for more details.

do not offer unique contributions that could benefit from more complex distillation strategies. However, as the scenario expands to include larger devices like XL and XXL in the 5- and 7-device configurations, TAKFL significantly outperforms existing KD-based methods. This improvement is driven by the larger devices' ability to offer more significant and higher-quality knowledge, which TAKFL effectively distills across all prototypes, contrasting sharply with existing methods that fail to utilize this potential. These experimental observations, corroborated by our theoretical insights in Remark 2, demonstrate TAKFL's superior scalability and effectiveness.

## 8    Conclusion and Discussion

In this work, we addressed a fundamental issue in standard federated learning: the lack of support for heterogeneous device prototypes. Existing KD-based methods often fall short in real-world scenarios, where device capabilities vary widely. To address this, we introduced TAKFL, a novel KD-based method that treats knowledge transfer from each prototype's ensembles as separate tasks and distills them independently. TAKFL susequently integrates the knowledge using an adaptive task arithmetic technique for optimized performance. We also introduced a KD-based self-regulation technique to mitigate issues arising from noisy and unsupervised ensemble distillation. The effectiveness of our method is substantiated by both theoretical results and extensive experimentation across CV and NLP tasks, using various datasets and models.

Limitations remain, notably in real-world applicability. While TAKFL's effectiveness in an approximated real-world setup has been demonstrated, actual deployment on physical devices and in environments with extremely large models remains untested due to resource constraints. Experiencing TAKFL in genuine real-world settings could unveil additional challenges or limitations, providing further insights into its scalability and efficiency.

## 9    Acknowledgment

This work was partially supported by a research grant from Cisco Systems, Inc., Project Number 49790. V.K. acknowledges support from the Czech National Science Foundation under Project 24-11664S. We also gratefully acknowledge the use of the computational infrastructure provided by the OP VVV funded project CZ.02.1.01/0.0/0.0/16_019/0000765, "Research Center for Informatics," which enabled us to conduct the experiments presented in this work.

We would like to express our sincere gratitude to Ang Li, Matias Mendieta, and Guangyu Sun for their invaluable feedback and insightful discussions, which significantly contributed to the development and refinement of this work. Their thoughtful suggestions and careful review of earlier drafts were instrumental in enhancing the quality of this paper.

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

## Appendix

The supplementary materials are organized as follows:

- Appendix A: Provides more details on related works.
- Appendix B: Presents the full algorithm description of TAKFL.
- Appendix C: Presents formal theoretical statements, assumptions, and proofs supporting our method.
- Appendix D: Presents detailed experimental results including some additional experiments.
- Appendix E: Presents the ablation studies experiments.
- Appendix F: Presents hyper-parameters and implementation details.

## A   More Detailed Related Works

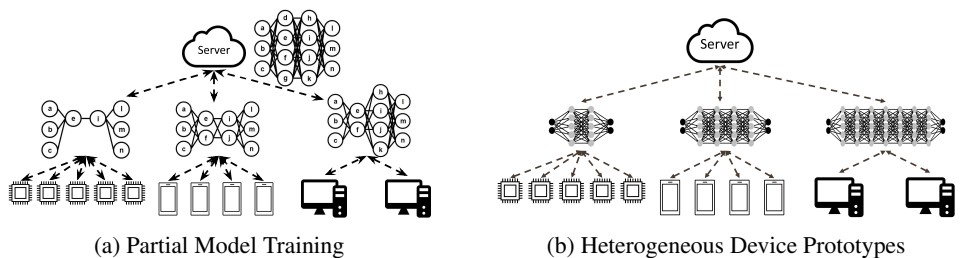

(a) Partial Model Training                    (b) Heterogeneous Device Prototypes

Figure 4: **Overview of Two Different Device Heterogeneous FL Settings.** (a) In the partial model training setting, the objective is to train a single global model where heterogeneous devices train a specific sub-model based on their computational resources. This approach necessitates device support for varying neural network architectures, which is impractical as devices typically have specialized architectures designed to match their hardware, software configurations, and underlying machine learning tasks. (b) In the heterogeneous device prototypes setting, device prototypes participate in FL to enhance the performance of their global model by transferring knowledge across prototypes. This setting is more feasible as it accommodates diverse device prototypes with their own specific configurations, including neural network architecture and dataset. However, establishing effective knowledge transfer between differently sized prototypes (like IoTs and workstations) and diverse configurations is challenging. In this paper, we address this issue.

Prior works on device heterogeneous FL have considered two distinct approaches with different objectives and settings. The first group of studies focuses on accommodating devices with varying compute resources, aiming to train a single global model [11, 3, 58, 54, 56]. Various partial model training techniques have been proposed for this setting, where devices are tasked with training a sub-model of a global model according to their compute resources. These include dropout-based [3], static [11, 18], and rolling-based sub-model extraction techniques [1]. Federated Dropout builds upon the concept of dropout [46] to extract smaller sub-models. Static sub-model extraction techniques like in HeteroFL [11] and FjORD [18] consistently extract designated portions of the global model, whereas FedRolex [1] introduces a more flexible rolling method for sub-model extraction. However, these approaches assume that devices can support various sub-model architectures for training, which does not fully reflect the real-world scenario. In practice, there exist a diverse spectrum of device prototypes such as IoT devices and smartphones each have unique and unhashable neural network architectures tailored to their specific hardware and software configurations and underlying machine learning tasks. Consequently, these device prototypes may not support training various neural network architectures, highlighting a significant limitation in accommodating the full spectrum of device heterogeneity in this setting.

The second array of studies tackles a more practical scenario where device prototypes with heterogeneous model architectures participate in FL to enhance their global model performance through mutual knowledge sharing. In this context, knowledge distillation techniques are employed to transfer knowledge among device prototypes [30, 6, 43]. Here, locally updated client models from various

device prototypes, collectively referred to as ensembles, serve as teachers to distill their knowledge into each server's student model using an unlabeled public dataset. For instance, FedDF [30] utilizes vanilla averaging of all ensemble logits as the distillation target for all server student models. In contrast, FedET [6] employs an uncertainty-weighted average of ensembles' logits as the distillation target for all server student models, complemented by a diversity regularization technique. However, methods like FedET rely on the neural networks' confidence scores for uncertainty estimates, overlooking the fact that neural networks are often poorly calibrated and prone to overconfidence, which compromises their ability to provide reliable uncertainty estimates [50, 15, 5, 55]. *These existing works typically focus on settings where device prototypes have similar capabilities, i.e. similar model and dataset sizes, thus neglecting the challenges presented in more diverse settings where device prototypes vary significantly in terms of model and dataset size. This oversight limits the effectiveness of these methods in truly diverse and heterogeneous environments. In this paper, we introduce TAKFL, which is designed to address the limitations of existing methods in these underexplored diverse device heterogeneous settings.*

Figure 1 illustrates the distinctions between these two different settings studied in the literature. For more information, we refer the reader to recent surveys [35, 26, 39, 4].

# B   Full Algorithm Description of TAKFL

The full algorithm description of TAKFL is presented in Algorithm 1.

---

**Algorithm 1** TAKFL Algorithm

---

**Require:** number of communication rounds ($R$), public unlabeled dataset $\mathbb{D}^{\text{public}}$, server training iterations $I$, heterogeneous device prototypes ($i \in \mathbb{M}$) with their associated clients ($\mathbb{C}^i$) and local datasets ($\{\mathbb{D}_k^i\}_{k \in \mathbb{C}^i}$), model architecture ($f^i$), local training iterations ($I_{local}$), local learning rate ($\eta_{local}$), server distillation iterations ($I_{distill}$), and server distillation learning rate ($\eta_{distill}$).

1: **Server Executes:**
2: Randomly initialize all device prototype's server model $\{\boldsymbol{\theta}_0^i\}_{i \in \mathbb{M}}$
3: **for** each round $r = 0, 1, \ldots, R - 1$ **do**
4:     $\mathbb{C}_r^i \leftarrow$ (randomly select clients from each device prototype) $\forall i \in \mathbb{M}$
5:     **for** each client $k \in \mathbb{C}_r^i$, $\forall i \in \mathbb{M}$ **in parallel do**
6:         $\widehat{\boldsymbol{\theta}}_k^i \leftarrow \texttt{ClientUpdate}(k; \boldsymbol{\theta}_r^i)$
7:     **end for**
8:     $\boldsymbol{\theta}_{avg}^i = \sum_{k \in \mathbb{C}_r^i} \frac{|\mathbb{D}_k^i|}{\sum_{k \in \mathbb{C}_r^i} |\mathbb{D}_k^i|} \widehat{\boldsymbol{\theta}}_k^i$
9:     **for** each device prototype's server student $i = 1, 2, \ldots, M$ **in parallel do**
10:         **for** each device prototype's teacher ensembles $j = 1, 2, \ldots, M$ **in parallel do**
11:             $\boldsymbol{\theta} \leftarrow \boldsymbol{\theta}_{avg}^i$
12:             **for** each server distillation iteration $t = 0, 1, 2, \ldots, I_{distill}$ **do**
13:                 $\boldsymbol{x} \leftarrow$ sample a mini-batch of data from public dataset $\mathbb{D}^{\text{public}}$
14:                 $\boldsymbol{\theta}^{t+1} \leftarrow \boldsymbol{\theta}^t - \eta_{distill} \cdot \nabla \mathcal{L}_{\mathcal{S}}^{\mathcal{T}_i}$ defined in Eq. 6.
15:             **end for**
16:             $\tau_j \leftarrow \boldsymbol{\theta}^{I_{distill}} - \boldsymbol{\theta}_{avg}^i$
17:         **end for**
18:         $\boldsymbol{\theta}_{r+1}^i \leftarrow \boldsymbol{\theta}_{avg}^i + \sum_{j=1}^M \lambda_j \tau_j$
19:     **end for**
20:     $\boldsymbol{\theta}_{r+1}^i \leftarrow \boldsymbol{\theta}^i$
21: **end for**

22: **function** $\texttt{ClientUpdate}(k, \boldsymbol{\theta}_r^i)$
23:     $\boldsymbol{\theta} \leftarrow \boldsymbol{\theta}_r^i$
24:     **for** each local update iteration $t = 0, 1, \ldots, I_{local} - 1$ **do**
25:         $\{\boldsymbol{x}, y\} \leftarrow$ sample a mini-batch of data from local dataset $\mathbb{D}_k^i$
26:         $\boldsymbol{\theta}^{t+1} \leftarrow \boldsymbol{\theta}^t - \eta_{local} \cdot \nabla \ell(f^i(\boldsymbol{x}; \boldsymbol{\theta}^t), y)$
27:     **end for**
28:     $\widehat{\boldsymbol{\theta}}_k^i \leftarrow \boldsymbol{\theta}^{I_{local}}$
29: **end function**

---

## C  Theoretical Results

### C.1  Proofs of the Main Propositions

First we present the formal assumptions associated with our theoretical derivations.

**Assumption 1.** Local federated averaging is performed with perfect test accuracy, i.e.,

$$\underset{\boldsymbol{\theta}^j}{\operatorname{argmin}} \sum_{j=1}^{M} \sum_{k=1}^{N^j} \mathbb{E}_{(\boldsymbol{x},y)\sim\mathbb{D}_k^j} \left[\ell(f^j(\boldsymbol{x};\boldsymbol{\theta}^j),y)\right] = \underset{\boldsymbol{\theta}^j}{\operatorname{argmin}} \mathbb{E}_{(\boldsymbol{x},y)\sim\mathcal{D}^j} \left[\ell(f^j(\boldsymbol{x};\boldsymbol{\theta}),y)\right] \tag{9}$$

That is, the training error on the datasets $\{\mathbb{D}_k^j\}$ for the computed $\theta_{avg}^j$ is the same as the test error on the population distribution $\mathcal{D}^j$. Moreover assume that we can write $\mathcal{T}_i = \left\{ \sum_{k=1}^{N^i} f^i(\cdot,\hat{\boldsymbol{\theta}}_k^i) | k \in \mathbb{C}^i \right\} = \{f^i(\cdot,\theta_{avg}^i)\}$. Finally, we assume that the same population distribution $\sum_j \omega_j \mathcal{D}^j$ is the same that the clients perform their testing on as the server performs distillation on.

These assumptions are made for mathematical practicality while at the same time not starkly unreasonable. The local FL the device prototypes perform is generically prone to imprecision, especially as the clients' data varies, but this discrepancy is bounded [16]. Similarly the difference in the average of logits and the logit of averages has a bounded difference norm [53]. Thus, violations of the Assumption add additional perturbations to quantities derived in the Theoretical analysis without having structural/qualitative effects, and thus would only present clutter in the presentation.

**Notations.** Now we present the notation defining the specific quantities we refer to in the derivations below. The set of important quantities is given in Table 3. Note that the formal definitions of the first two quantities are,

$$\boldsymbol{\Theta}^j := \underset{\boldsymbol{\theta}}{\operatorname{argmin}} \mathbb{E}_{(\boldsymbol{x},y)\sim\mathcal{D}^j} \left[\ell(f^j(\boldsymbol{x};\boldsymbol{\theta}),y\right], \ \boldsymbol{\Theta}^{j,k} := \underset{\boldsymbol{\theta}}{\operatorname{argmin}} \mathbb{E}_{(\boldsymbol{x},y)\sim\mathcal{D}^i} \left[\ell(f^j(\boldsymbol{x};\boldsymbol{\theta}),y\right]$$

Table 3: Notation and Definitions

| Notation | Definition |
|---|---|
| $\boldsymbol{\Theta}^j$ | Parameters in $j$'s device model that minimize the loss on its population distribution |
| $\boldsymbol{\Theta}^{j,k}$ | Parameters in $j$'s device model that minimize the loss on $i$'th population distribution |
| $Q^j = \dim(\boldsymbol{\theta}^j)$ | The total capacity of device prototype $j$ |
| $\mathcal{Q}^j = \{e_k^j\}_{k=1,\dots,Q^j}$ | Eigenbasis for the model of device prototype $j$ |
| $W^j = \dim(\boldsymbol{\Theta}^j)$ | Dimension of the solution submanifold $\boldsymbol{\Theta}^j$ |
| $W^{j,k} = \dim(\boldsymbol{\Theta}^{j,k})$ | Dimension of the solution submanifold $\boldsymbol{\Theta}^{j,k}$ |
| $\mathcal{W}^j = \{e_k^j\}_{k=1,\dots,W^j}$ | Eigenbasis the solution submanifold $\boldsymbol{\Theta}^j$ |
| $\mathcal{W}^{j,k} = \{e_l^{j,k}\}_{l=1,\dots,W^{j,k}}$ | Eigenbasis the solution submanifold $\boldsymbol{\Theta}^{j,k}$ |

We shall make use of the "Choose" combinatorial operator, defined to be $Ch(n,p) = \frac{n!}{p!(n-p)!}$. The standard $O(\cdot)$ notation indicates $a_k = O(b_k)$ to mean there exists $K$ and $C$ such that for $k \geq K$, $a_k \leq Cb_k$.

A recent finding that inspired the methodology in this work is the discovery of the weight disentanglement phenomenon underlying task arithmetic [37]. Indeed the *task arithmetic property* provides the ideal circumstance for federated knowledge transfer as we shall see below. Formally, adapting their definition to our notation:

(**Task Arithmetic Property**) holds for a set of vectors $\{\boldsymbol{\tau}_j\}$ if for all $j$ it holds that,

$$f^j\left(\boldsymbol{x};\boldsymbol{\theta}_{avg}^j + \sum_{i\neq j}\lambda_i\boldsymbol{\tau}_i\right) = \begin{cases} f^j(\boldsymbol{x};\boldsymbol{\theta}_{avg}^j + \lambda_i\boldsymbol{\tau}_i) & \boldsymbol{x} \in \mathcal{D}^i \\ f^j(\boldsymbol{x};\boldsymbol{\theta}_{avg}^j) & \boldsymbol{x} \in \mathcal{D}^j \setminus \cup_{i\neq j}\mathcal{D}^i \end{cases} \tag{10}$$

Let us define an important property of task arithmetic that we shall use in the sequel.

**(Weight disentanglement).**[37] A parametric function $f : \mathcal{X} \times \Theta \to \mathcal{Y}$ is weight disentangled with respect to a set of task vectors $T = \{\boldsymbol{\tau}_j\}_{j \in \mathbf{T}}$ and the corresponding supports $\mathcal{D}_T := \{\mathcal{D}_j\}_{j \in \mathbf{T}}$ if

$$f(\boldsymbol{x}; \boldsymbol{\theta}_0 + \sum_{i \in \mathbf{T}}^{T} \alpha_i \boldsymbol{\tau}_i) = \sum_{i \in \mathbf{T}} g_j(\boldsymbol{x}; \alpha_i \boldsymbol{\tau}_i) + g_0(\boldsymbol{x}),$$

where $g_i(x; \alpha_i \boldsymbol{\tau}_i) = 0$ for $\boldsymbol{x} \notin \mathcal{D}_i$ and $i \in \mathbf{T}$, and $g_0(\boldsymbol{x}) = 0$ for $\boldsymbol{x} \in \bigcup_{i \in \mathbf{T}} \mathcal{D}_i$.

We now present the formal statements as well as the proofs of the main propositions.

**Proposition 1.** *(Information Loss in VED). Let the prototype $i$ and $j$ datasets be disjoint, i.e. $\mathcal{D}^i \cap \mathcal{D}^j = \emptyset$, and $\bigcup_{i=1}^{M} \mathcal{D}^\rangle = \mathcal{D}$. Note that this implies the weight disentanglement property. Consider the VED procedure in the form of solving (3). Consider two device prototypes with a device capacity and solution dimension of $Q^1, Q^2$ and $W^1, W^2$, respectively, and with associated eigenbases $\mathcal{Q}^i, \mathcal{W}^i$. Let the solution set of VED with prototype $i$ as student be $\hat{\Theta}_{VED}^i$ with $\dim(\hat{\Theta}_{VED}^i) = W^{v_i}$ with eigenbasis $\mathcal{W}^{v_i}$. In addition, denote $W^{s,t}$, $s,t \in \{1,2\}$ the dimension of the solution set for the student model trained on the data from the teacher device's ensembles. We assume that self-distillation is executed appropriately, e.g., $W^{1,1} = W^1$ and $W^{2,2} = W^2$.*

1. ***Case 1:*** *Assume that $Q^1 = Q^2$ and $W^1 = W^2 = W^{1,2} = W^{2,1}$. Then it holds that, in expectation,*

$$\dim\left(\hat{\Theta}_{VED}^1 \cap [\mathcal{Q}^1 \setminus \mathcal{W}^1]\right) = O\left(\frac{(Q^1 - W^1)(W^1)!(Q^1 - W^{1,2})!}{Q^1!(W^1)!(Q^1 - W^1)! + Q^1!W^{1,2}!(Q^1 - W^{1,2})!}\right)$$

   *This corresponds to the expected capacity of prototype 1 that is taken up for fitting logits that are not in the span of $\mathcal{W}^1$, that is, that do not fit the data corresponding to prototype 1.*
2. ***Case 2:*** *Assume that $Q^1 > Q^2$ and $W^1 = W^{1,2} > W^2$. Then the same quantity as for Case 1 holds. Moreover,*

$$\dim\left(\hat{\Theta}_{VED} \cap [\mathcal{Q}^1 \setminus (\mathcal{W}^1 \cup \mathcal{W}^{1,2})]\right) = O\left(\frac{(Q^1 - W^1)(W^1!)(W^{1,2} - W^2)!}{Q^1!W^1!(Q^1 - W^1)! + Q^1!W^2!(W^{1,2} - W^2)!}\right)$$

   *This corresponds to capacity of client 1 that has been allocated but fits, in the model of prototype 1, neither the data of prototype 1, nor of the data of prototype 2.*

*Proof.* Formally,

$$\hat{\Theta}_{VED} := \underset{\theta \in \mathcal{Q}^1}{\operatorname{argmin}} \, \mathcal{L}_{\text{ED}} = \underset{\theta}{\operatorname{argmin}} \, \text{KL}\left[\sum_{i=1,2} \sigma\left(f^i(\boldsymbol{x}, \boldsymbol{\theta}_{avg}^i)\right), \sigma\left(\mathcal{S}(\boldsymbol{x})\right)\right]$$

Since by assumption $\boldsymbol{\theta}_{avg}^i$ solves the training problem on the data associated with device prototype $i$, the logit is accurate, and thus there is a map $\mathcal{O}(i,j) : \mathcal{T}_i \to \mathcal{T}_i^j \subseteq \mathcal{W}^{i,j}$. The self distillation, that is, $\mathcal{S}_j$ defines a bijective map from $\mathcal{W}^j$ to $\mathcal{W}^j$ and thus does not affect the capacity allocation.

**Case 1:** In this case, generically (that is, measure zero on some non-atomic sampling on a distribution of operators) $\mathcal{O}(i,j)$ is bijective. Now let us compute the expectation of the number of eigenvectors of, e.g. $\mathcal{W}^{1,2}$ that are in the complement of the span of $\mathcal{W}^1$. Assuming, for simplicity, independence, this would correspond to counting the possible choices within the capacity of $\mathcal{Q}^1 \setminus \mathcal{W}^1$ over the range of possible choices of filling the capacity of $\mathcal{Q}^1$ with vectors in $\mathcal{W}^1$ together with choices of filling it with vectors in $\mathcal{W}^{1,2}$:

$$\sum_{i=1}^{Q^1 - W^1} i \frac{Ch(Q^1 - W^1, i)}{Ch(Q^1, W^1) + Ch(Q^1, W^{1,2})}$$

For, e.g., $Q^1 = 4$ and $W^1 = W^{1,2} = 2$ this is $\frac{1}{3}$.

To derive a scaling rate we can write:

$$\sum_i i \frac{\frac{(Q^1 - W^1)!}{i!(Q^1 - W^1 - i)!}}{\frac{Q^1!}{(W^1)!(Q^1 - W^1)!} + \frac{Q^1!}{W^{1,2}!(Q^1 - W^{1,2})!}} = O\left(\frac{(Q^1 - W^1)(W^1)!(Q^1 - W^{1,2})!}{Q^1!(W^1)!(Q^1 - W^1)! + Q^1!W^{1,2}!(Q^1 - W^{1,2})!}\right)$$

**Case 2:** In this case, it must be that, at best almost surely, $\mathcal{O}(2,1)$ is injective, but not surjective. This means that distilling from 2 to 1 does not fill the capacity of $\mathcal{W}^{1,2}$, and is thus a fundamentally

wasteful operation, that is $|\mathcal{T}_j^j| = W^2 < W^{1,2}$. Now let us compute the expectation of the number of eigenvectors of, e.g. $\mathcal{W}^{1,2}$ that are in the complement of the span of $\mathcal{W}^1$. Since $\mathcal{W}^{1,2}$ are being structurally allocated for fitting, the combinatorial expression is the same:

$$\sum_{i=1}^{Q^1-W^1} i \frac{Ch(Q^1 - W^1, i)}{Ch(Q^1, W^1) + Ch(Q^1, W^{1,2})}$$

Thus for, e.g., $Q^1 = 4$ and $W^1 = W^{1,2} = 2$ this is, again, $\frac{1}{3}$. The scaling in this case is

$$O\left(\frac{(Q^1 - W^1)(W^1!)(Q^1 - W^{1,2})!}{Q^1!W^1!(Q^1 - W^1)! + Q^1!W^2!(Q^1 - W^2)!}\right)$$

However, we observe that there are vectors in the range of $\mathcal{W}^{1,2} \setminus \mathcal{O}(2,1)(\mathcal{W}^2)$ that have been allocated by the VED but lie in neither $\mathcal{W}^1$ nor in $\mathcal{W}^{1,2}$, that is, are garbage. We can compute those as the expected number of eigenvectors arising from allocating $\mathcal{W}^{1,2} \setminus \mathcal{O}(2,1)(\mathcal{W}^2)$ that intersect with $\mathcal{Q}^1 \setminus \mathcal{W}^1$ (that is, the spare capacity not used for fitting data $\mathcal{D}^1$). This is, using similar principles:

$$\sum_{i=1}^{W^{1,2}-W^1} i \frac{Ch(Q^1 - W^1, i)}{Ch(Q^1, W^1) + Ch(Q^1, W^{1,2} - W^2)}$$

This is for, e.g., $Q^1 = 4$, $W^{1,2} = 2$ and $W^2 = 1$, this would be $\frac{3}{10}$

The scaling here is

$$O\left(\frac{(Q^1 - W^1)(W^1!)(W^{1,2} - W^2)!}{Q^1!W^1!(Q^1 - W^1)! + Q^1!W^2!(W^{1,2} - W^2)!}\right)$$

∎

**Proposition 2.** *(Improve knowledge transfer with task arithmetic). Consider the TAKFL procedure as in the form of computing* (8). *Consider two device prototypes with a device capacity and solution dimension of $Q^1, Q^2$ and $W^1, W^2$, respectively, and with associated eigenbases $\mathcal{Q}^i, \mathcal{W}^i$. Let the solution set of TAKFL with prototype $i$ as student be $\hat{\Theta}_{TA}^i$ with $\dim(\hat{\Theta}_{TA}^i) = W^v$ with eigenbasis $\mathcal{W}^v$. In addition, denote $W^{s,t}$, $s,t \in \{1,2\}$ dimension of the solution set for the student model trained on the data from the teacher device's ensembles. . The following statements hold:*

*In the case that that $Q^1 \geq Q^2$ and $W^1 \geq W^2$, it holds that the TAKFL preserves that the eigenbasis used to model the data $\mathcal{D}^1$'s accuracy for device prototype 1, that is for student 1*

$$\dim\left(\mathcal{W}^v \cap [\mathcal{Q}^1 \setminus \mathcal{W}^1]\right) = 0$$

***Case 1:*** *Assume that $Q^1 = Q^2$ and $W^1 = W^2$. Then it holds that,*

$$\dim\left(\mathcal{W}^v \cap [\mathcal{Q}^1 \setminus (\mathcal{W}^1 \cup \mathcal{W}^{1,2})]\right) = 0$$

*Moreover, it holds that,*

$$\hat{\Theta}_{TA} \in Span(\mathcal{W}^1 \cap \mathcal{W}^{1,2})$$

*Thus, with equal capacity, no information is lost in Task Arithmetic aided knowledge ensemble distillation and capacity is efficiently used to model the data from both prototype 1 and prototype 2.*

***Case 2:*** *Assume that $Q^1 > Q^2$ and $W^1 > W^2$. Then it again holds that,*

$$\dim\left(\mathcal{W}^v \cap [\mathcal{Q}^1 \setminus (\mathcal{W}^1 \cup \mathcal{W}^{1,2})]\right) = 0$$

*However, while $\hat{\Theta}_{TA} \in Span(\mathcal{W}^1)$, it holds that $\dim\left(\mathcal{W}^v \cap \mathcal{W}^{1,2}\right) = W^2 < W^{1,2}$.*

*Proof.* We consider the case that the prototype $i$ and $j$ datasets be disjoint, i.e. $\mathcal{D}^i \cap \mathcal{D}^j = \emptyset$, and $\bigcup_{i=1}^M \mathcal{D}^\rangle = \mathcal{D}$. We shall see how the other cases are easier and the result remain the same.

We can see immediately from the weight disentanglement property of Task Arithmetic that,

$$f^1(\boldsymbol{x}; \boldsymbol{\theta}_{avg}^1 + \alpha_1\boldsymbol{\tau}_1 + \alpha_2\boldsymbol{\tau}_2) = g^{1,1}(\boldsymbol{x}; \alpha_1\boldsymbol{\tau}_1) + g^{1,2}(\boldsymbol{x}; \alpha_2, \boldsymbol{\tau}_2) + g^{1,0}(x)$$

with $g^{1,1}(\boldsymbol{x}; \alpha_1 \boldsymbol{\tau}_1)$ for $\boldsymbol{x} \notin \mathcal{D}^1$, $g^{1,2}(\boldsymbol{x}; \alpha_2 \boldsymbol{\tau}_2)$ for $\boldsymbol{x} \notin \mathcal{D}^2$ and $g^{1,0}(\boldsymbol{x}) = 0$ for $\boldsymbol{x} \in \mathcal{D}^1 \cup \mathcal{D}^2$. From this, we can immediately conclude the first statement of the Proposition as well as the expression

$$\dim \left( \mathcal{W}^v \cap \left[ \mathcal{Q}^1 \setminus (\mathcal{W}^1 \cup \mathcal{W}^{1,2}) \right] \right) = 0$$

and also, in the case of $W^1 = W^{1,2} = W^2$ implies

$$\hat{\Theta}_{TA} \in Span(\mathcal{W}^1 \cap \mathcal{W}^{1,2})$$

For the last statement we observe again as in the second Case in the Proposition describing VED, $\dim \left( \mathcal{O}(2,1)(\mathcal{W}^2) \right) < W^{1,2}$ from which we can conclude that, generically

$$\dim \left( \{v : v \in \mathcal{O}(2,1)(\mathcal{W}^2) \subseteq \mathcal{W}^v, \, \mathbb{E}_{(\boldsymbol{x},y)\sim\mathcal{D}^2} l(f^1(\boldsymbol{x}; v), y) > 0\} \right) = W^{1,2} - W^2$$

proving the final statement. ∎

We observe that a key mechanism of the proof is the dimension of the target space of the teaching operator $\mathcal{O}(i,j)$. As an informative model, we can consider coefficients $\lambda_j$ of task vectors as restricting the rank, relative to other teachers. For instance, in the previous Proposition, if $W^{1,2} = 2$ and $W^2 = 1$, then $\lambda_2 = 1/2$, so as to enforce one vector of $\mathcal{W}^{1,2}$ is a target for the map $\tilde{O}(2,1)$, would be appropriately sensible.

The prototype $i$ and $j$ datasets are fully overlapping, i.e. $\mathcal{D}^i \subseteq \mathcal{D}^j$, meaning that the two prototypes have the same classes $\mathcal{D}_c, \forall c \in D$, then these are the changes to the the proposition 1 and 2 in our theory.

Consider that in this case, the logits corresponding to $\mathcal{W}^{1,1}$ and $\mathcal{W}^{1,2}$ are the same. As such $\mathcal{W}^{1,2} \in \mathcal{Q}^1$, and so the propositions change to, trivially:

**Proposition 3.** (Information Loss in VED). *Consider the VED procedure. Consider two device prototypes with a device capacity and solution dimension of $Q^1, Q^2$ and $W^1, W^2$, respectively, and with associated eigenbases $\mathcal{Q}^i, \mathcal{W}^i$. Let the solution set of VED with prototype $i$ as student be $\hat{\Theta}^i_{VED}$ with $\dim(\hat{\Theta}^i_{VED}) = W^{v_i}$ with eigenbasis $\mathcal{W}^{v_i}$. In addition, denote $W^{s,t}, s,t \in \{1,2\}$ the dimension of the solution set for the student model trained on the data from the teacher device's ensembles. We assume that self-distillation is executed appropriately, e.g., $W^{1,1} = W^1$ and $W^{2,2} = W^2$.*

1. ***Case 1:*** *Assume that $Q^1 = Q^2$ and $W^1 = W^2 = W^{1,2} = W^{2,1}$. Then it holds that, in expectation,*

$$\dim \left( \hat{\Theta}^1_{VED} \cap \left[ \mathcal{Q}^1 \setminus \mathcal{W}^1 \right] \right) = 0$$

   *This corresponds to the expected capacity of prototype 1 that is taken up for fitting logits that do not fit the data corresponding to prototype 1.*

2. ***Case 2:*** *Assume that $Q^1 > Q^2$ and $W^1 = W^{1,2} > W^2$. Then the same quantity as for Case 1 holds. Moreover,*

$$\dim \left( \hat{\Theta}_{VED} \cap \left[ \mathcal{Q}^1 \setminus (\mathcal{W}^1 \cup \mathcal{W}^{1,2}) \right] \right) = 0$$

   *This corresponds to capacity of client 1 that has been allocated but fits, in the model of prototype 1, neither the data of prototype 1, nor of the data of prototype 2.*

Now consider that the prototype $i$ and $j$ datasets are partially overlapping, i.e. $\mathcal{D}^i \cap \mathcal{D}^j \neq \emptyset$, meaning that the two prototypes have the overlap over some of the classes $\mathcal{D}_c$, then these are the changes to the the the proposition 1 and 2 in our theory.

In this case, $\mathcal{W}^{1,2} = \mathcal{W}^{1,2}_1 \cup \mathcal{W}^{1,2}_2$, that is, the class labels across data prototypes partially overlap (with $W^{1,2}_1$ the overlapping dimensionality), the Proposition for VED changes to:

**Proposition 4.** (Information Loss in VED). *Consider the VED procedure. Consider two device prototypes with a device capacity and solution dimension of $Q^1, Q^2$ and $W^1, W^2$, respectively, and with associated eigenbases $\mathcal{Q}^i, \mathcal{W}^i$. Let the solution set of VED with prototype $i$ as student be $\hat{\Theta}^i_{VED}$ with $\dim(\hat{\Theta}^i_{VED}) = W^{v_i}$ with eigenbasis $\mathcal{W}^{v_i}$. In addition, denote $W^{s,t}, s,t \in \{1,2\}$ the dimension of the solution set for the student model trained on the data from the teacher device's ensembles. We assume that self-distillation is executed appropriately, e.g., $W^{1,1} = W^1$ and $W^{2,2} = W^2$.*

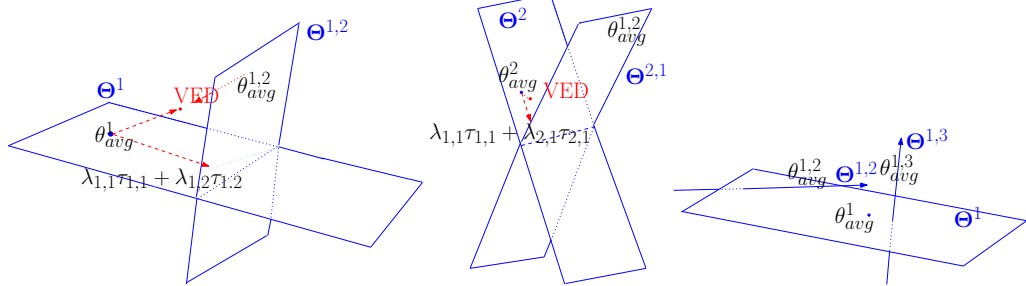

Figure 5: **Illustration of Geometric Intuition**. Each panel presents a different case example. The left and center panels present the geometric intuition of KD for vanilla ensemble distillation (VED) and TAKFL in the case where two different large device prototypes performing knowledge transfer. The planes represent the solution subspaces. The right panel presents a circumstance by which two small device prototypes (2, and 3) serve as teacher for transferring knowledge to a larger device prototype 1.

1. **Case 1:** *Assume that $Q^1 = Q^2$ and $W^1 = W^2 = W^{1,2} = W^{2,1}$. Then it holds that, in expectation,*

$$\dim\left(\hat{\Theta}^1_{VED} \cap \left[\mathcal{Q}^1 \setminus \mathcal{W}^1\right]\right) = O\left(\frac{(Q^1 - W^1)(W^1)!(Q^1 - W^{1,2}_2)!}{Q^1!(W^1)!(Q^1 - W^1)! + Q^1!W^{1,2}_2!(Q^1 - W^{1,2}_2)!}\right)$$

   *This corresponds to the expected capacity of prototype 1 that is taken up for fitting logits that do not fit the data corresponding to prototype 1.*

2. **Case 2:** *Assume that $Q^1 > Q^2$ and $W^1 = W^{1,2} > W^2$. Then the same quantity as for Case 1 holds. Moreover,*

$$\dim\left(\hat{\Theta}_{VED} \cap \left[\mathcal{Q}^1 \setminus (\mathcal{W}^1 \cup \mathcal{W}^{1,2})\right]\right) = O\left(\frac{(Q^1 - W^1)(W^1!)(W^{1,2}_2 - W^2)!}{Q^1!W^1!(Q^1 - W^1)! + Q^1!W^2!(W^{1,2}_2 - W^2)!}\right)$$

   *This corresponds to capacity of client 1 that has been allocated but fits, in the model of prototype 1, neither the data of prototype 1, nor of the data of prototype 2.*

Note that the presence of overlapping logits $W^{1,2}_1$ do not appear in the result, it is exactly the same as the original Proposition 1 in the original paper just with $W^{1,2}_2$ in place of $W^{1,2}$.

The Propositions for TAKFL remain the same.

### C.2 Geometric Intuition

In this section we aim to provide geometric intuition for the mechanism of VED and Task Arithmetic KD on three different cases. Figure 5 presents the geometry illustration for three different cases. We discuss each case in the following.

**Case I: KD between two large prototypes with different data distributions.** Consider Figure 5 left panel. This panel corresponds to a setting where two large device prototypes with similar total capacity, i.e. $Q^1 = Q^2 = 3$ perform knowledge transfer. We consider the solution dimensions of both prototypes to be the same, i.e. $W^1 = W^2 = 2$. These would correspond to planes in the ambient space. Therefore, one plane corresponds to the solution subspace of the prototype 1 trained on its own data, i.e. $\Theta^1$ subspace in the panel, and the other corresponds to the (theoretical) solution subspace of this prototype trained on prototype 2's data, i.e. $\Theta^{(1,2)}$ in the panel. In this case, since the data distributions of the prototypes are fairly disparate, this has resulted into near orthogonal subspaces corresponding to these solutions. As we can see from the panel, VED will lead to point which is far away from either of the planes corresponding to optimal solution subspaces, and far from the optimal set of parameters, which is their intersection, suggesting a loss of knowledge. By contrast, the TAKFL approach, by customizing the merging coefficients and putting each to half, i.e. $\lambda_{1,1}\lambda_{1,2} = 0.5$, can traverse in the tangent space of zero loss surface and get into the intersection subspace which is exactly the optimal solution ($\theta^*_{merged} = \theta^1_{avg} + \lambda_{1,1}\tau_{1,1} + \lambda_{1,2}\tau_{1,2}$).

**Case II: KD between two large prototypes with similar data distributions.** Consider Figure 5 center panel. Similar to Case I, this panel corresponds to a setting where two large device prototypes with similar total capacity, i.e. $Q^1 = Q^2 = 3$ performing knowledge transfer. We consider the solution dimensions of both prototypes to be the same, i.e. $W^1 = W^2 = 2$. These would correspond to planes in the ambient space. Therefore, one plane corresponds to the solution subspace of the prototype 1 trained on its own data, i.e. $\Theta^1$ subspace in the panel, and the other corresponds to the (theoretical) solution subspace of this prototype trained on prototype 2's data, i.e. $\Theta^{(1,2)}$ in the panel. In this case, since the data distributions of the prototypes are fairly close, this has resulted into non-orthogonal solution subspaces. As we can see from the panel, while VED could still lead to some information loss, by and large we expect straightforward KD. Our task arithmetic (TA) approach, again by customizing the merging coefficients $\lambda_{1,1}\lambda_{1,2} = 0.5$, can traverse in the tangent space of zero loss surface on each plane and get into the intersection subspace, corresponding to the most efficient allocation of device prototype capacity for fitting simultaneously the logits corresponding to accurate modeling of device prototype 1 as well as device prototype 2's data distribution.

**Case III: KD between two small prototypes and one large prototype.** Now consider the right panel in Figure 5. This panel corresponds to a setting where two small prototypes serve as teachers and one large prototype is the student. The $\boldsymbol{\theta}^1_{avg}$ plane corresponds to the solution subspace of the large prototype 1 on its own data, $\mathcal{D}^1$. The line $\boldsymbol{\theta}^{1,2}_{avg}$ line corresponds to the subspace of solutions in prototype 1's parameter space projected into the capacity of the information transferred from device prototype 2. Finally, the line labelled $\boldsymbol{\theta}^{1,3}_{avg}$ corresponds to the subspace of solutions in prototype 1's parameter space projected into the capacity of the information transferred from device prototype 3. Here, we can see from the relative angle of the lines with respect to the plane that the distribution $\mathcal{D}^1$ is closer to the distribution $\mathcal{D}^2$ than to $\mathcal{D}^3$. Comparing this case to the previous cases, $\boldsymbol{\theta}^{1,3}_{avg}$ is like case I and $\boldsymbol{\theta}^{1,2}_{avg}$ is like case II. We can apply the same conclusions here as well regarding the performance of vanilla ensemble distillation and our adaptive task arithmetic approach. We can see from the geometric visualization that knowledge distillation towards $\boldsymbol{\theta}^{1,3}_{avg}$ has more margin of error for prototype 1. Therefore, with the TAKFL approach large prototype 1 can strategically select which prototype to learn more from, and since $\boldsymbol{\theta}^{1,2}_{avg}$ has closer data distribution to prototype 1, TAKFL will prioritizes this by putting a larger merging coefficient, i.e $\lambda_{1,2} > \lambda_{1,3}$. By contrast, VED lacks this customization and results in sub-optimal knowledge distillation.

The geometric intuition discussed here is consistent with our detailed experimental analysis in D.1.1.

## C.3 Analytical Properties of Learning Dynamics

Here we provide additional insights from the literature as to the nature and properties of learning as it takes place on overparametrized models. Specifically, we comment on literature in the area of Stochastic Differential Equation (SDE) models for SGD training dynamics, and its correspondence to the results above. Overparametrization has been conjectured to be a significant factor in contributing to the (unexpected, by classical bias-variance tradeoffs) generalization ability of deep neural networks, from a number of perspectives [13].

Consider the diffusion model of SGD training for overparametrized NNs provided in [29]. Their analysis relies on the following two assumptions. For our purposes $L$ is shorthand for a client group's loss, $L(\boldsymbol{\theta}) = \sum_{k \in \mathbb{C}^j} \mathbb{E}_{(\boldsymbol{x},y)\sim\mathbb{D}^j_k} \left[ \ell(f^j(\boldsymbol{x};\boldsymbol{\theta}), y) \right]$ for some $j$, which will be identified from the context.

**Assumption 2.** $L : \mathbb{R}^Q \to \mathbb{R}$ is $C^3$ and the solution set $\Gamma$ is a $W$-dimensional $C^2$-submanifold of $\mathbb{R}^D$ for $0 \le W \le D$ and $\text{rank}(\nabla^2 L(\theta)) = Q - W$

**Assumption 3.** Assume that $U$ is an open neighborhood of $\Gamma$ satisfying that gradient flow starting in $U$ converges to some point in $\Gamma$

From these, [29] derives Theorem 4.6. This Theorem decomposes the random process of the parameter weights driven by SGD after it has reached the solution manifold, e.g., the diffusive random walk of $\theta^1_{avg}$ in Figure 5 along its respective solution manifold.

**Theorem 1.** *Given $L$, $\Gamma$ and $\theta_\mu(0) = \theta(0) \in U$ as by Assumptions 2 and 3 the SDE modeling the optimization of $F$ by SGD, that is, defining $\Phi(X)$ to be the gradient flow applied to the state random variable $X$, then for $T$ as long as $\mathbb{P}[Y(t) \in U, \forall 0 \le t \le T] = 1$, $\theta_\eta(\lfloor T/\eta^2 \rfloor$ converges in*

*distribution to the stochastic process $Y(T)$ as $\eta \to 0$, with $Y(t)$ given as,*

$$dY(t) = \Sigma_{\parallel}^{1/2}(Y)dW(t) - \tfrac{1}{2}\nabla^2 L(Y)^\dagger \partial^2(\nabla L)(Y)[\Sigma_{\parallel}(Y)]dt \\ - \tfrac{1}{2}\partial\Phi(Y)\left(2\partial^2(\nabla L)(Y)\left[\nabla^2 L(Y)^\dagger \Sigma_{\perp,\parallel}(Y)\right] + \partial^2(\nabla L)(Y)\left[\mathcal{L}_{\nabla^2 L}^{-1}(\Sigma_\perp(Y))\right]\right)dt \quad (11)$$

*where $\Sigma \equiv \sigma\sigma^T$ and $\Sigma_{\parallel}$, $\Sigma_{\parallel}$, $\Sigma_{\parallel}$ are given as,*

$$\Sigma_{\parallel} = \partial\Phi\Sigma\partial\Phi, \; \Sigma_\perp = (I_D - \partial\Phi)\Sigma(I_D - \partial\Phi), \\ \Sigma_{\parallel,\perp} = \partial\Phi\sigma(I_D - \partial\Phi) \quad (12)$$

This theorem indicates that the asymptotic flow of SGD on the client training can be decomposed into a covariance-driven random walk in the tangent space, drift to preserve the flow into the tangent plane, the tangent-normal portion of the noise covariance and noise in the normal direction.

This analytical expression provides the probabilistic foundations for the more higher level theoretical results above. In particular, local gradient dynamics, as employed by individual device prototypes $j$ using FedAvg on its local clients, yields a flow for the stochastic process defined by the weights. At this point of learning, the weights are traversing the solution set, with noise predominantly in the tangent directions. Thus knowledge distillation which preserves this noise structure is going to be more effective as far as preserving accuracy across data.

## D  Detailed Experimental Results

In this section we present a more detailed version of experimental results presented in the main paper Section 7. Additional experimental results are also presented here.

### D.1  Main Experimental Results on Computer Vision (CV) Task

**The experiments in this section complements the main experimental results in the main paper Section 7.2.**

**Experimental Setup.** For the evaluation on CV task, we employ CIFAR-10 and CIFAR-100 [24] datasets. For CIFAR-10, we use CIFAR-100 as the unlabeled public dataset, while ImageNet-100, a subset of ImageNet [10] with 100 classes (see Appendix F.1.1), is used for CIFAR-100. We distribute the training dataset among the device prototypes in a ratio of 1:3:6 for S, M, and L, respectively. Each device prototype's data portion is further distributed among its clients using a Dirichlet distribution. We apply two levels of data heterogeneity for a comprehensive evaluation: low heterogeneity, i.e. Dir(0.3), and high heterogeneity, i.e. Dir(0.1). Additionally, we configure the number of clients and their sampling rates as follows: 100 clients for S, 20 for M, and 4 for L, with sampling rates set at 0.1, 0.2, and 0.5 respectively. To comprehensively evaluate, we use two distinct architectural settings: the *"homo-family"* setting, where all device prototypes' architectures are from the same family—employing ResNet8, ResNet14, and ResNet18 [17] for S, M, and L, respectively; and the *"hetero-family"* setting, which diverse architectures are used—ViT-S [12] for S, ResNet14 for M, and VGG-16 [44] for L. All models are initialized from scratch, and the communication round is set at 60 rounds. Further details regarding hyper-parameters can be found in Table 13.

**Overview of Performance Results.** Table 4 presents the performance of TAKFL across diverse architecture settings on the CIFAR-10 and CIFAR-100 datasets. TAKFL consistently improves all device prototypes of different sizes in various cases by a significant margin compared to the baselines, achieving SOTA performance. Notably, in the homo-family architecture setting with Dir(0.3) on CIFAR-10, TAKFL improves average performance across all prototypes by 8%, and by 4% on CIFAR-100. In the hetero-family settings with Dir(0.1) on CIFAR-10 and Dir(0.3) on CIFAR-100, TAKFL enhances performance by ∼3% and 1%, respectively. Furthermore, we observe that our self-regularization technique has successfully mitigated issues associated with the noisy and unsupervised ensemble distillation process, thereby enhancing performance. Generally, the performance gains from self-regularization are more pronounced in low data heterogeneity cases, where prototypes' models perform better and possess higher quality self-knowledge. Thus, self-regularization proves more effective as it preserves this higher quality self-knowledge.

Table 4: **Performance Results for CV task on CIFAR-10 and CIFAR-100.** Training data is distributed among S, M, and L device prototypes in a 1:3:6 ratio, subdivided among clients using Dirichlet distribution. Public datasets are CIFAR-100 for CIFAR-10 and ImageNet-100 for CIFAR-100. Client configurations include 100, 20, and 4 clients for S, M, and L, with sampling rates of 0.1, 0.2, and 0.5. In homo-family settings, architectures are ResNet8, ResNet14, and ResNet18; in hetero-family settings, they are ViT-S, ResNet14, and VGG-16. All models are trained from scratch for 60 rounds. See Appendix F.1 for more details.

| | | Homo-family Architecture Setting | | | | | | | |
| Dataset | Baseline | Low Data Heterogeneity | | | | High Data Heterogeneity | | | |
| | | S | M | L | Average | S | M | L | Average |
|---|---|---|---|---|---|---|---|---|---|
| | FedAvg | $36.21_{\pm2.24}$ | $46.41_{\pm2.33}$ | $59.46_{\pm6.17}$ | 47.36 | $22.01_{\pm0.78}$ | $25.26_{\pm3.89}$ | $51.51_{\pm3.52}$ | 32.93 |
| | FedDF | $49.31_{\pm0.15}$ | $50.63_{\pm0.73}$ | $49.82_{\pm0.98}$ | 49.92 | $34.71_{\pm1.48}$ | $35.27_{\pm4.74}$ | $51.08_{\pm4.04}$ | 40.35 |
| CIFAR-10 | FedET | $49.21_{\pm0.72}$ | $55.01_{\pm1.81}$ | $53.60_{\pm6.47}$ | 52.61 | $29.58_{\pm3.00}$ | $30.96_{\pm4.70}$ | $45.53_{\pm6.46}$ | 35.36 |
| | TAKFL | $55.90_{\pm1.70}$ | $57.93_{\pm3.49}$ | $60.58_{\pm2.35}$ | 58.14 | $37.40_{\pm1.68}$ | $38.96_{\pm0.17}$ | $51.49_{\pm6.15}$ | 42.61 |
| | TAKFL+Reg | $\mathbf{56.37_{\pm0.46}}$ | $\mathbf{58.60_{\pm0.43}}$ | $\mathbf{65.69_{\pm1.28}}$ | **60.22** | $\mathbf{40.51_{\pm1.05}}$ | $\mathbf{40.12_{\pm1.24}}$ | $\mathbf{53.24_{\pm2.51}}$ | **44.62** |
| | FedAvg | $13.22_{\pm0.14}$ | $21.39_{\pm1.11}$ | $29.47_{\pm0.86}$ | 21.36 | $11.86_{\pm0.08}$ | $14.63_{\pm0.65}$ | $26.25_{\pm1.64}$ | 17.58 |
| | FedDF | $19.54_{\pm0.20}$ | $24.32_{\pm0.45}$ | $29.29_{\pm1.45}$ | 24.38 | $16.09_{\pm0.32}$ | $19.80_{\pm0.17}$ | $26.59_{\pm0.25}$ | 20.83 |
| CIFAR-100 | FedET | $19.67_{\pm0.35}$ | $25.27_{\pm0.66}$ | $31.10_{\pm1.53}$ | 25.35 | $11.18_{\pm1.68}$ | $18.22_{\pm0.35}$ | $26.40_{\pm0.65}$ | 18.60 |
| | TAKFL | $24.48_{\pm0.42}$ | $27.60_{\pm0.25}$ | $29.84_{\pm0.94}$ | 27.31 | $22.90_{\pm0.18}$ | $23.63_{\pm0.72}$ | $26.98_{\pm0.13}$ | 24.50 |
| | TAKFL+Reg | $\mathbf{27.18_{\pm0.27}}$ | $\mathbf{29.14_{\pm0.20}}$ | $\mathbf{31.15_{\pm0.97}}$ | **29.16** | $22.88_{\pm0.37}$ | $\mathbf{23.92_{\pm0.57}}$ | $\mathbf{28.01_{\pm0.34}}$ | **24.94** |

| | | Hetero-family Architecture Setting | | | | | | | |
| Dataset | Baseline | Low Data Heterogeneity | | | | High Data Heterogeneity | | | |
| | | S | M | L | Average | S | M | L | Average |
|---|---|---|---|---|---|---|---|---|---|
| | FedAvg | $27.53_{\pm0.83}$ | $47.30_{\pm3.17}$ | $55.10_{\pm8.60}$ | 43.31 | $20.93_{\pm1.54}$ | $25.62_{\pm6.04}$ | $36.80_{\pm5.47}$ | 27.78 |
| | FedDF | $34.15_{\pm0.87}$ | $54.06_{\pm1.06}$ | $69.07_{\pm4.99}$ | 52.43 | $24.20_{\pm0.74}$ | $34.07_{\pm3.08}$ | $39.81_{\pm5.45}$ | 32.69 |
| CIFAR-10 | FedET | $33.24_{\pm1.27}$ | $58.86_{\pm0.94}$ | $65.56_{\pm3.49}$ | 52.55 | $24.37_{\pm1.26}$ | $37.77_{\pm4.71}$ | $43.64_{\pm3.36}$ | 35.26 |
| | TAKFL | $33.29_{\pm0.15}$ | $57.64_{\pm0.19}$ | $68.44_{\pm0.66}$ | 53.12 | $24.92_{\pm1.32}$ | $38.07_{\pm3.19}$ | $48.01_{\pm3.99}$ | 37.00 |
| | TAKFL+Reg | $\mathbf{33.34_{\pm3.36}}$ | $\mathbf{59.01_{\pm3.12}}$ | $\mathbf{70.22_{\pm4.40}}$ | **54.19** | $\mathbf{25.10_{\pm1.87}}$ | $\mathbf{38.81_{\pm5.36}}$ | $\mathbf{50.26_{\pm6.42}}$ | **38.06** |
| | FedAvg | $8.51_{\pm0.37}$ | $22.11_{\pm0.58}$ | $37.91_{\pm2.60}$ | 22.84 | $7.01_{\pm0.47}$ | $14.94_{\pm0.96}$ | $28.51_{\pm1.46}$ | 16.82 |
| | FedDF | $10.46_{\pm0.17}$ | $23.46_{\pm0.65}$ | $36.81_{\pm0.82}$ | 23.58 | $7.76_{\pm0.40}$ | $18.92_{\pm0.39}$ | $29.81_{\pm1.09}$ | 18.83 |
| CIFAR-100 | FedET | $11.16_{\pm0.18}$ | $25.40_{\pm0.30}$ | $37.38_{\pm0.60}$ | 24.65 | $8.20_{\pm0.54}$ | $20.66_{\pm0.50}$ | $28.95_{\pm1.79}$ | 19.27 |
| | TAKFL | $10.29_{\pm0.11}$ | $27.14_{\pm0.89}$ | $\mathbf{39.15_{\pm0.88}}$ | 25.53 | $7.88_{\pm0.68}$ | $21.41_{\pm0.37}$ | $31.31_{\pm0.66}$ | 20.20 |
| | TAKFL+Reg | $\mathbf{11.25_{\pm0.37}}$ | $\mathbf{27.86_{\pm0.86}}$ | $38.68_{\pm0.45}$ | **25.93** | $\mathbf{8.45_{\pm0.20}}$ | $\mathbf{22.16_{\pm0.87}}$ | $\mathbf{31.95_{\pm1.13}}$ | **20.85** |

### D.1.1 Consistency Analysis: Experimental and Theoretical Correlations

In this part, we elaborate on our key experimental observations and their alignment with our theoretical findings.

**Insight 1:** From Table 4, it is evident that prior KD-based methods show inconsistent performance across various device prototypes, particularly for the large (L) prototype. For instance, in the CIFAR-10 homo-family setting with Dir(0.3), while small (S) and medium (M) prototypes see performance gains, the L prototype experiences up to a ∼10% performance decline compared to vanilla FedAvg, which lacks server-side knowledge distillation. This trend is consistent across other settings, such as CIFAR-10 Dir(0.1) homo-family and CIFAR-100 Dir(0.3) homo-family. These outcomes underline the dilution problem inherent in existing methods, where the valuable insights from larger, more capable device prototypes are overshadowed by less informative outputs from smaller devices, thereby degrading the performance of L prototypes. These empirical findings are supported by our theoretical insights as discussed in Remark 1. Specifically, Proposition 1 illustrates that vanilla ensemble distillation (VED) leads to knowledge dilution and inaccuracies due to misaligned device capacity allocations. Moreover, this issue becomes more significant when the smaller device prototype serve as teacher.

**Insight 2:** From Table 4, the suboptimality of existing KD-based methods is evident from the significant performance improvements of our method, especially for S and M prototypes across various settings. This underscores the ineffectiveness of the one-size-fits-all approach these methods employ, where a single averaged logits distillation target is used for all device sizes, proving to be suboptimal. Our experimental observations regarding the shortcomings of vanilla ensemble distillation methods align with our theoretical findings, as substantiated in Remark 1 and 2. It becomes evident that an efficient knowledge distillation process must allocate capacity in a manner that appropriately corresponds to the information value of the teacher ensemble prototypes.

**Insight 3**: Our experiments, detailed in Table 4, demonstrate TAKFL's adept handling of knowledge from various device prototypes under different data heterogeneity conditions. We observed consistent performance gains for small (S) and medium (M) prototypes across both low and high data heterogeneity, compared to vanilla FedAvg. However, in high heterogeneity settings, large (L)

prototypes show less improvement, prompting the question: *What can smaller device prototypes offer to larger ones?*

In low heterogeneity scenarios, large prototypes significantly benefit from the collective knowledge, showing enhanced performance. Conversely, in conditions of extreme heterogeneity, where smaller models contribute less effectively, the performance improvements for larger devices are notably reduced. This pattern highlights TAKFL's ability to intelligently manage and utilize the available knowledge, selectively distilling information based on the intrinsic capacity and contributions of each prototype, and integrating only the most valuable knowledge from smaller devices when beneficial.

By contrast, our analysis of existing KD-based methods shows their failure to effectively discern and utilize the most informative knowledge across prototypes. These methods often overload the capacity of larger prototypes with suboptimal or irrelevant information, particularly in high heterogeneity environments, leading to not just stagnation but an accumulation of inefficiencies. These experimental observations align with our theoretical insights, as outlined in Remark 2, which emphasizes the crucial combinatorial constraint of capacity and diverse information. This further confirms the superiority of TAKFL's adaptive approach to knowledge distillation in diverse federated learning environments.

## D.2 Additional Experimental Results on CV Task

**Experimental Setup.** For additional evaluation of the CV task, we conducted experiments on Tiny-ImageNet [25], STL-10 [7], and CINIC-10 [9] datasets using pre-trained models. For TinyImageNet [25], we utilized STL-10 [7] as the unlabeled public dataset. STL-10 [7] and CINIC-10 [9] both employ CIFAR-100 [24] as their respective public datasets. We distributed the training datasets among the device prototypes in a 2:3:5 ratio for small (S), medium (M), and large (L) prototypes, respectively. The data portion for each prototype was further subdivided among its clients using a Dirichlet distribution: Dir(1.0) for TinyImageNet and Dir(0.3) for both STL-10 and CINIC-10. Client configurations were set with 4, 3, and 2 clients for S, M, and L, respectively, all with a sampling rate of 1.0. The architectures employed were MobileNetV3-Large [19] for S, Mobile-ViTV2 [34] for M, and ResNet-34 for L, sourced from the TIMM library.[1] The local training was conducted over 10 epochs using an Adam [23] optimizer with a learning rate of 1e-3 and weight decay of 1e-5. For server-side distillation, the epoch count was 10 for TinyImageNet and 1 for STL-10 and CINIC-10, with a batch size of 128, employing an Adam optimizer with a learning rate of 1e-5 and weight decay of 1e-5. For TinyImageNet, public dataset images from STL-10 were resized to 64×64, while for STL-10, images were resized to 32×32. No data augmentation was used. The communication rounds is fixed to 40. These experiments were conducted by only 1 trial. Table 15 details the configurations.

**Performance Results.** Table 5 presents the results. The superiority of TAKFL's performance across these challenging datasets using pre-trained models is evident here as well.

## D.3 Additional Experimental Results on Natural Language Processing (NLP) Task

**The experiments in this section complements the main experimental results in the main paper Section 7.2.**

**Experimental Setup.** For the evaluation of NLP tasks, we utilize four datasets: MNLI [52], SST-2 [45], MARC [22], and AG-news [60]. The corresponding unlabeled public datasets are SNLI [2] for MNLI, Sentiment140 [14] for SST-2, Yelp [61] for Amazon, and DBPedia [59] for AG-News. The training data is distributed among the device prototypes in a ratio of 1:3:6 for small (S), medium (M), and large (L) categories, respectively, with each portion further subdivided among its clients using a Dirichlet distribution (Dir(0.5)). The client configurations and their sampling rates are set as follows: 8, 4, and 2 clients for S, M, and L categories, respectively, with sampling rates of 0.3, 0.5, and 1.0. The architectures employed for each prototype size are BERT [47] -Tiny, -Small, and -Mini, respectively, each initialized from pre-trained parameters and tested over 20 communication rounds. Additional details regarding hyper-parameters and datasets are presented in Appendix F.2 and Table 17.

---

[1] https://github.com/huggingface/pytorch-image-models

Table 5: **Performance Results for CV task on TinyImageNet, STL-10, and CINIC-10 using pre-trained models.**

| Private | Public | Baseline | S | M | L | Average |
|---------|--------|----------|---|---|---|---------|
| TinyImageNet | STL-10 | FedAvg | 8.97 | 13.03 | 15.12 | 12.37 |
| | | FedMH | 15.08 | 17.10 | 17.83 | 16.67 |
| | | FedET | 10.60 | 16.39 | 17.62 | 14.87 |
| | | TAKFL | 16.10 | 17.60 | 19.03 | 17.58 |
| | | TAKFL+Reg | **16.55** | **17.98** | **19.74** | **18.09** |
| STL-10 | CIFAR-100 | FedAvg | 26.01 | 34.47 | 42.88 | 34.45 |
| | | FedMH | 28.64 | 34.55 | 39.25 | 34.15 |
| | | FedET | 29.87 | 33.00 | 38.26 | 33.71 |
| | | TAKFL | 29.57 | 37.57 | 42.53 | 36.56 |
| | | TAKFL+Reg | **30.78** | **37.89** | **43.38** | **37.35** |
| CINIC-10 | CIFAR-100 | FedAvg | 44.87 | 55.49 | 51.33 | 50.56 |
| | | FedMH | 45.52 | 55.75 | 53.48 | 51.58 |
| | | FedET | 46.31 | 57.43 | 53.01 | 52.25 |
| | | TAKFL | **48.21** | **57.81** | 52.74 | **52.92** |
| | | TAKFL+Reg | 47.66 | 57.54 | **53.25** | 52.82 |

Table 6: **Performance Results for NLP Task on 4 Datasets.**. Training data is distributed among S, M, and L device prototypes in a 1:3:6 ratio, subdivided among clients using Dir(0.5). Client configurations are 8, 4, and 2 clients for S, M, and L, with sample rates of 0.3, 0.5, and 1.0, respectively. Architectures include Bert-Tiny, Bert-Mini, and Bert-Small for S, M, and L, initialized from pre-trained parameters and fine-tuned for 20 communication rounds. See Appendix F.2 for more details.

| Private | Public | Baseline | S | M | L | Average |
|---------|--------|----------|---|---|---|---------|
| MNLI | SNLI | FedAvg | $36.15_{\pm0.46}$ | $54.47_{\pm2.48}$ | $57.51_{\pm2.79}$ | 49.37 |
| | | FedDF | $54.21_{\pm0.15}$ | $60.44_{\pm1.91}$ | $66.71_{\pm1.09}$ | 60.45 |
| | | FedET | $48.03_{\pm6.32}$ | $50.33_{\pm7.87}$ | $53.80_{\pm6.18}$ | 50.72 |
| | | TAKFL | $57.43_{\pm0.21}$ | $63.58_{\pm0.31}$ | $68.74_{\pm0.12}$ | 63.25 |
| | | TAKFL+Reg | $\mathbf{57.61_{\pm0.89}}$ | $\mathbf{63.91_{\pm1.05}}$ | $\mathbf{68.96_{\pm1.10}}$ | **63.49** |
| SST2 | Sent140 | FedAvg | $54.98_{\pm1.81}$ | $74.71_{\pm8.22}$ | $86.69_{\pm0.06}$ | 72.13 |
| | | FedDF | $74.41_{\pm2.62}$ | $80.71_{\pm1.63}$ | $84.35_{\pm1.66}$ | 79.82 |
| | | FedET | $66.63_{\pm9.14}$ | $65.89_{\pm16.35}$ | $70.05_{\pm15.83}$ | 67.52 |
| | | TAKFL | $74.73_{\pm0.55}$ | $82.17_{\pm0.31}$ | $86.93_{\pm0.42}$ | 81.28 |
| | | TAKFL+Reg | $\mathbf{74.88_{\pm0.43}}$ | $\mathbf{82.40_{\pm0.83}}$ | $\mathbf{87.33_{\pm0.63}}$ | **81.54** |
| MARC | Yelp | FedAvg | $33.76_{\pm1.13}$ | $49.08_{\pm1.28}$ | $59.26_{\pm1.43}$ | 47.36 |
| | | FedDF | $53.01_{\pm1.24}$ | $55.37_{\pm0.87}$ | $56.81_{\pm0.99}$ | 55.06 |
| | | FedET | $52.63_{\pm2.29}$ | $54.28_{\pm2.31}$ | $56.11_{\pm2.61}$ | 54.34 |
| | | TAKFL | $55.70_{\pm2.08}$ | $58.64_{\pm1.75}$ | $59.39_{\pm1.16}$ | 57.91 |
| | | TAKFL+Reg | $\mathbf{55.96_{\pm1.66}}$ | $\mathbf{59.18_{\pm1.13}}$ | $\mathbf{59.61_{\pm1.89}}$ | **58.25** |
| AG-News | DBPedia | FedAvg | $83.64_{\pm3.51}$ | $83.47_{\pm2.35}$ | $91.48_{\pm2.22}$ | 86.20 |
| | | FedDF | $85.97_{\pm2.45}$ | $89.10_{\pm1.85}$ | $91.37_{\pm1.10}$ | 88.81 |
| | | FedET | $75.27_{\pm3.85}$ | $81.13_{\pm3.21}$ | $83.19_{\pm4.58}$ | 79.86 |
| | | TAKFL | $87.37_{\pm1.31}$ | $90.11_{\pm1.56}$ | $92.48_{\pm1.12}$ | 89.99 |
| | | TAKFL+Reg | $\mathbf{87.66_{\pm1.83}}$ | $\mathbf{90.30_{\pm2.05}}$ | $\mathbf{92.61_{\pm1.72}}$ | **90.19** |

**Performance on NLP Task.** Table 6 presents the results on four different datasets: MNLI, SST-2, MARC, and AG-News. Similar to the CV task, TAKFL has consistently improved performance across all device prototypes of varying sizes, achieving state-of-the-art results. On MNLI, it has enhanced average performance across all prototypes by 3%, on SST-2 by ~2%, on MARC by 3%, and on AG-News by ~1.50%. As observed in the CV task, the suboptimality of existing KD-based methods is also evident here. Notably, FedET exhibits very poor performance compared vanilla FedAvg, failing to achieve satisfactory results on all datasets except for the MARC dataset. Particularly, the performance of the L prototype has consistently decreased across all datasets compared to vanilla FedAvg. This behavior can be attributed to FedET's reliance on neural network confidence scores for uncertainty estimates in its uncertainty-weighted distillation. However, neural networks, especially pretrained language models (PLMs), are often poorly calibrated and prone to overconfidence, which compromises their ability to provide reliable uncertainty estimates [50, 15, 5, 55].

 **Scalability Evaluation**

**This section complements the experimental results in the main paper Section 7.3.**

**Experimental Setup.** To evaluate the effectiveness and scalability of our method across a broad spectrum of device prototypes, ranging from very small to very large sizes, we conduct experiments involving 3 to 7 different prototypes. Our objective is to assess how effectively our method adapts from a uniform array of small-size prototypes (3 device prototypes) to a diverse mix that includes prototypes ranging from extremely small (XXS) to extremely large (XXL) (7 device prototypes). These experiments involve training image classification models from scratch on the CINIC-10 dataset, using CIFAR-100 as the unlabeled public dataset. We randomly distribute the dataset among prototypes with dataset ratios set to 1:2:3:4:5:6:7 from XXS to XXL. Each dataset portion is further distributed among clients using a Dirichlet distribution (Dir(0.5)). The number of clients ranges from 35 to 5 from XXS to XXL, respectively. Client sample rates are set at 0.1, 0.1, 0.15, 0.15, 0.2, 0.3, and 0.6 from XXS to XXL. We use a series of ResNet architectures—ResNet10-XXS, ResNet10-XS, ResNet10-S, ResNet10-M, ResNet10, ResNet18, and ResNet50—scaled appropriately for each prototype. The local training epochs are set at 2, 2, 2, 5, 10, 10, and 20 from XXS to XXL to account for resource constraints, with fewer epochs assigned to smaller devices. We employ the Adam optimizer with a learning rate of 1e-3 and a weight decay of 5e-5 for local training. For XL and XXL, a step learning rate scheduler reduces the learning rate by a factor of 0.1 at half epoch. Server-side distillation employs a fixed batch size of 128, using the Adam optimizer with learning rate of 1e-3 and weight decay of 5e-5. The softmax temperature is set at 3 for ensemble distillation and 20 for self-regularization. The number of communication rounds is fixed at 30. These experiments are conducted over 3 trials with different random seeds, and the average performance with standard deviation is reported. The entire device prototypes configurations are given in Table 16.

The detailed results are presented in Tables 7, 8, and 9.

Table 7: **Scalability Evaluation.** Detailed performance results for 7 device prototypes case.

| Baseline | XXS | XS | S | M | L | XL | XXL | Average |
|---|---|---|---|---|---|---|---|---|
| FedAvg | $23.17_{\pm1.26}$ | $30.66_{\pm0.14}$ | $32.81_{\pm0.21}$ | $31.77_{\pm0.21}$ | $37.69_{\pm0.08}$ | $41.78_{\pm0.05}$ | $50.52_{\pm0.01}$ | 35.49 |
| FedDF | $27.98_{\pm0.66}$ | $\mathbf{37.47}_{\pm\mathbf{0.33}}$ | $40.61_{\pm0.01}$ | $40.26_{\pm0.18}$ | $43.83_{\pm0.22}$ | $45.58_{\pm0.18}$ | $52.18_{\pm0.12}$ | 41.13 |
| FedET | $26.75_{\pm0.98}$ | $36.99_{\pm0.31}$ | $40.51_{\pm0.19}$ | $41.60_{\pm0.16}$ | $46.12_{\pm0.31}$ | $48.39_{\pm0.11}$ | $52.71_{\pm0.09}$ | 41.87 |
| TAKFL | $27.30_{\pm0.08}$ | $36.93_{\pm0.16}$ | $43.31_{\pm0.42}$ | $40.88_{\pm0.01}$ | $48.52_{\pm0.15}$ | $50.95_{\pm0.04}$ | $54.27_{\pm0.43}$ | 43.17 |
| TAKFL+Reg | $\mathbf{29.28}_{\pm\mathbf{0.16}}$ | $37.10_{\pm0.45}$ | $\mathbf{43.96}_{\pm\mathbf{1.65}}$ | $\mathbf{41.83}_{\pm\mathbf{0.73}}$ | $\mathbf{48.77}_{\pm\mathbf{0.37}}$ | $\mathbf{51.43}_{\pm\mathbf{0.46}}$ | $\mathbf{54.63}_{\pm\mathbf{0.84}}$ | **43.86** |

Table 8: **Scalability Evaluation.** Detailed performance results for 5 device prototypes case.

| Baseline | XXS | XS | S | M | XL | Average |
|---|---|---|---|---|---|---|
| FedAvg | $24.19_{\pm1.03}$ | $21.04_{\pm0.76}$ | $33.62_{\pm0.88}$ | $38.91_{\pm0.74}$ | $46.93_{\pm0.05}$ | 32.94 |
| FedDF | $\mathbf{28.31}_{\pm\mathbf{0.61}}$ | $34.66_{\pm0.00}$ | $39.91_{\pm0.07}$ | $38.24_{\pm0.36}$ | $46.81_{\pm0.11}$ | 37.59 |
| FedET | $26.88_{\pm0.95}$ | $34.11_{\pm0.27}$ | $\mathbf{41.15}_{\pm\mathbf{0.29}}$ | $40.81_{\pm0.87}$ | $48.14_{\pm0.06}$ | 38.22 |
| TAKFL | $27.91_{\pm0.12}$ | $37.09_{\pm0.11}$ | $40.46_{\pm0.34}$ | $41.06_{\pm0.02}$ | $49.02_{\pm0.35}$ | 39.11 |
| TAKFL+Reg | $28.24_{\pm0.46}$ | $\mathbf{37.30}_{\pm\mathbf{1.10}}$ | $40.76_{\pm0.94}$ | $\mathbf{43.09}_{\pm\mathbf{0.27}}$ | $\mathbf{50.86}_{\pm\mathbf{0.22}}$ | **40.05** |

Table 9: **Scalability Evaluation.** Detailed performance results for 3 device prototypes case.

| Baseline | XXS | S | M | Average |
|---|---|---|---|---|
| FedAvg | $24.19_{\pm1.03}$ | $33.62_{\pm0.88}$ | $\mathbf{38.91}_{\pm\mathbf{0.74}}$ | 32.24 |
| FedDF | $27.85_{\pm0.10}$ | $\mathbf{37.83}_{\pm\mathbf{0.12}}$ | $37.74_{\pm0.41}$ | 34.47 |
| FedET | $26.04_{\pm0.67}$ | $36.87_{\pm0.68}$ | $37.66_{\pm0.09}$ | 33.52 |
| TAKFL | $26.62_{\pm0.16}$ | $37.32_{\pm0.40}$ | $38.13_{\pm0.58}$ | 34.02 |
| TAKFL+Reg | $\mathbf{27.90}_{\pm\mathbf{0.98}}$ | $37.63_{\pm0.87}$ | $38.20_{\pm0.91}$ | **34.58** |

# E Ablation Studies

## E.1 Understanding Merging Coefficient

In this section, we conduct an ablation study to further understand how TAKFL customizes knowledge integration and understand how the merging coefficients $\lambda_i$ are achieving this. This experiment aims to further understand the trade-offs between customized knowledge integration approach from the one-size-fits-all strategy employed in vanilla ensemble distillation and prior works.

**Experimental Setup.** Our experimentation focuses on two device prototypes: XXS and XXL, selected from the scalability evaluation detailed in Section 7.3, Appendix D.4, and Table 16. We employ the image classification task on the CINIC-10 [9] dataset, starting from scratch. Each prototype receives a randomly selected, non-overlapping subset of the training dataset—3.57% for XXS and 25% for XXL—distributed among their clients in a non-i.i.d. manner using Dir(0.5). Both prototypes have three clients each. The architectures used are ResNet10-XXS for the XXS prototype and ResNet-50 for the XXL prototype. To focus solely on the evaulation of the server-side distillation process and its evolution with varying $\lambda$, we pre-train each prototype using standard FedAvg for 10 communication rounds, with a sample rate of 1.0. Local training involves 20 epochs for XXS and 20 epochs for XXL using an Adam optimizer with a learning rate of 1e-3 and weight decay of 5e-5. The XXL prototype employs a step learning rate scheduler that reduces the rate by a factor of 0.1 at local epoch 10. For server-side distillation, we utilize a batch size of 128 and an Adam optimizer with a learning rate of 1e-5 and weight decay of 5e-5. CIFAR-100 [24] serves as the unlabeled public dataset. We save the final updated client and server models from both prototypes for further experimentation, focusing on the impact of merging coefficients without self-regularization in TAKFL. The merging coefficient $\lambda$ is varied linearly from 0 to 1 in increments of 0.05. For simplicity, the XXS prototype is referred to as the small (S) prototype and XXL as the large (L) prototype.

**Discussion.** Figure 6 illustrates the significant impact of customized knowledge integration on the performance of both small and large device prototypes compared to the one-size-fits-all approach typical of vanilla ensemble distillation in the prior works, at different distillation epochs. Here, TAKFL adeptly manages customization for both small and large prototypes by controlling the merging coefficient $\lambda$. The merged model for both the small and large student prototypes is obtained using the formula $\boldsymbol{\theta}_{merged} = \boldsymbol{\theta}_{avg} + \left((1-\lambda)\boldsymbol{\tau}_S + \lambda\boldsymbol{\tau}_L\right)$. Notably, the performance is benchmarked at $\lambda \approx 0.5$ in all cases, reflecting similar results to vanilla ensemble distillation (FedDF), where no customization in knowledge transfer occurs. This baseline performance is critical for understanding the effects of further customization.

In small distillation epochs ($I_{distill} < 10$), minimal benefit is observed from customized knowledge integration, as both small and large prototypes achieve optimal performance at the non-customized $\lambda \approx 0.5$. However, as the distillation process progresses beyond 10 epochs, the influence of $\lambda$ becomes increasingly pronounced. For $\lambda > 0.5$, the knowledge from the large prototype's ensembles predominates, enhancing their impact, while for $\lambda < 0.5$, integration is more influenced by the small prototype's ensembles. This pattern suggests that increased distillation epochs enable more effective distillation of each prototype's unique knowledge for extreme cases of extremely small and large prototypes, thereby making the customization benefits evident. In scenarios with small distillation epochs, the absence of significant unique knowledge results in optimal performance at $\lambda \approx 0.5$. Conversely, as the number of distillation epochs rises ($I_{distill} \geq 20$), the one-size-fits-all strategy proves suboptimal, underscoring the importance of tailored knowledge integration strategies. Optimal performance increasingly occurs at $\lambda > 0.5$, indicating effective leveraging of each prototype's strengths to maximize overall performance. These findings confirm the necessity for customized knowledge integration in environments with significant prototype size variations and support our theoretical insights as detailed in Remark 1 and 2.

## E.2 Impact of Public Dataset

In this section, we explore the influence of the public dataset on the performance of TAKFL and existing KD-based methods when the public dataset used for server-side distillation is less similar to the private dataset, which is the actual learning objective. For this analysis we employ the same experimental setup previously outlined in Section 7.2 and Appendix D.1, using the CIFAR-10 homo-family architecture. To measure dataset similarity, we compute cosine similarity between the averaged features of datasets, extracted using an off-the-shelf pre-trained CLIP model [40] (CLIP ViT-B/32) available from the official GitHub repository.[2]

**Discussion.** Table 10 presents our results, highlighting a significant observation: the performance of existing methods drastically deteriorates as the similarity between the public dataset and private datasets decreases. In contrast, TAKFL exhibits robustness, suffering much less performance degradation under the same conditions. This demonstrates TAKFL's practical utility in real-world sce-

---

[2]https://github.com/OpenAI/CLIP

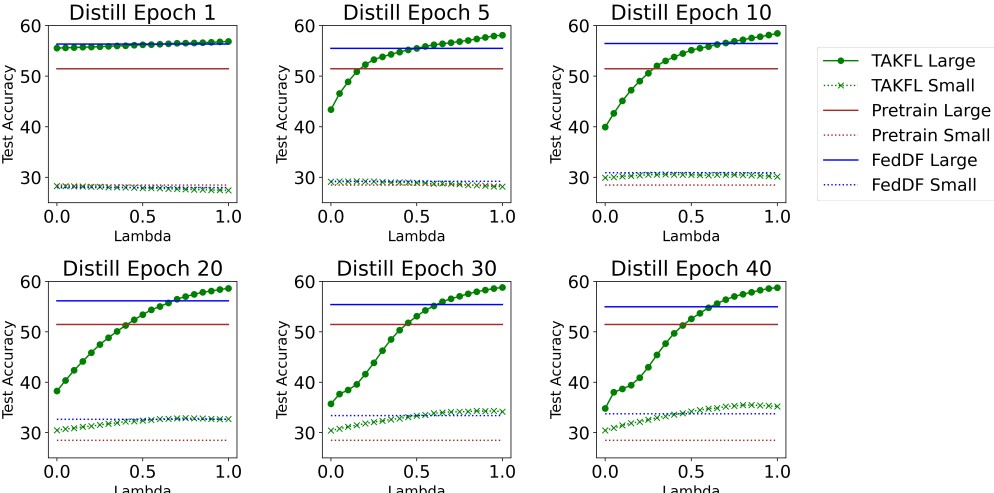

Figure 6: **Understanding the Impact of Merging Coefficients.** This figure showcases server-side knowledge distillation between two device prototypes, XXS and XXL, referred to as small and large, respectively, utilizing CIFAR-100 as the unlabeled public dataset. Both prototypes were pre-trained from scratch using standard FedAvg for 10 communication rounds. The CINIC-10 dataset was distributed between the small and large prototypes in ratios of 3.57% and 25%, respectively, and further subdivided non-i.i.d. among the clients using Dir(0.5). Each prototype has three clients with a sample rate of 1.0. The small prototype utilizes a ResNet10-XXS architecture, while the large prototype employs a ResNet-50.

narios where the server typically lacks knowledge of the private datasets to select a closely aligned public dataset for distillation. Notably, FedET underperforms significantly when using a less similar public dataset, performing worse than vanilla FedAvg in both low and high data heterogeneity scenarios. A similar pattern was observed with FedET in the NLP tasks discussed in Section 7.2 and Appendix D.3. This issue is likely due to FedET's dependence on the overconfident and poorly calibrated confidence scores from neural networks [15, 55] for uncertainty estimates in its uncertainty-weighted distillation approach.

Table 10: **Impact of Public Dataset on performance results.** Same experimental setting described in Section 7.2 and Appendix D.1 on CIFAR-10 homo-family setting is used for this experiment. The numbers in parentheses represent the similarity scores between private and public datasets, obtained using a pre-trained CLIP ViT-B/32 model.

| Public Dataset | Baseline | Low Data Heterogeneity (Dir(0.3)) | | | | High Data Heterogeneity (Dir(0.1)) | | | |
|---|---|---|---|---|---|---|---|---|---|
| | | S | M | L | Average | S | M | L | Average |
| — | FedAvg | $36.21_{\pm2.24}$ | $46.41_{\pm2.33}$ | $59.46_{\pm6.17}$ | 47.36 | $22.01_{\pm0.78}$ | $25.26_{\pm3.89}$ | $51.51_{\pm3.52}$ | 32.93 |
| CIFAR-100 (0.99) | FedDF | $49.31_{\pm0.15}$ | $50.63_{\pm0.73}$ | $49.82_{\pm0.98}$ | 49.92 | $34.71_{\pm1.48}$ | $35.27_{\pm4.74}$ | $51.08_{\pm4.04}$ | 40.35 |
| | FedET | $49.21_{\pm0.72}$ | $55.01_{\pm1.81}$ | $53.60_{\pm6.47}$ | 52.61 | $29.58_{\pm3.00}$ | $30.96_{\pm4.70}$ | $45.53_{\pm6.46}$ | 35.36 |
| | TAKFL | $55.90_{\pm1.70}$ | $57.93_{\pm3.49}$ | $60.58_{\pm2.35}$ | 58.14 | $37.40_{\pm1.68}$ | $38.96_{\pm0.17}$ | $51.49_{\pm6.15}$ | 42.62 |
| | TAKFL+Reg | $\mathbf{56.37}_{\pm0.46}$ | $\mathbf{58.60}_{\pm0.43}$ | $\mathbf{65.69}_{\pm1.28}$ | **60.22** | $\mathbf{40.51}_{\pm1.05}$ | $\mathbf{40.12}_{\pm1.24}$ | $\mathbf{53.24}_{\pm2.51}$ | **44.62** |
| TinyImagenet (0.92) | FedDF | $49.37_{\pm1.58}$ | $49.41_{\pm4.21}$ | $55.06_{\pm6.71}$ | 51.28 | $31.41_{\pm6.61}$ | $30.73_{\pm7.77}$ | $39.82_{\pm5.16}$ | 33.99 |
| | FedET | $33.95_{\pm0.92}$ | $37.26_{\pm1.64}$ | $39.77_{\pm3.44}$ | 36.99 | $24.12_{\pm1.84}$ | $24.58_{\pm2.13}$ | $28.91_{\pm1.09}$ | 25.87 |
| | TAKFL | $55.20_{\pm0.07}$ | $56.36_{\pm0.40}$ | $60.71_{\pm0.22}$ | 57.42 | $40.08_{\pm0.19}$ | $40.26_{\pm0.04}$ | $43.56_{\pm1.10}$ | 41.30 |
| | TAKFL+Reg | $\mathbf{56.28}_{\pm0.09}$ | $\mathbf{57.14}_{\pm0.03}$ | $\mathbf{60.90}_{\pm0.22}$ | **58.11** | $\mathbf{40.88}_{\pm0.11}$ | $\mathbf{41.10}_{\pm1.15}$ | $\mathbf{46.25}_{\pm5.95}$ | **42.74** |
| Celeb-A (0.77) | FedDF | $\mathbf{48.99}_{\pm0.37}$ | $50.06_{\pm0.43}$ | $55.12_{\pm4.95}$ | 51.39 | $29.80_{\pm0.39}$ | $32.28_{\pm4.41}$ | $44.0_{\pm4.60}$ | 35.36 |
| | FedET | $28.56_{\pm3.00}$ | $28.80_{\pm1.00}$ | $37.20_{\pm2.78}$ | 31.52 | $15.28_{\pm1.75}$ | $19.00_{\pm3.43}$ | $23.29_{\pm5.04}$ | 19.19 |
| | TAKFL | $45.65_{\pm2.72}$ | $54.53_{\pm1.72}$ | $58.13_{\pm0.13}$ | 52.77 | $\mathbf{31.02}_{\pm0.68}$ | $\mathbf{36.76}_{\pm1.58}$ | $48.33_{\pm0.53}$ | 38.70 |
| | TAKFL+Reg | $46.93_{\pm0.67}$ | $\mathbf{56.67}_{\pm1.26}$ | $\mathbf{60.13}_{\pm1.38}$ | **54.58** | $30.88_{\pm3.51}$ | $35.95_{\pm5.40}$ | $\mathbf{52.68}_{\pm1.90}$ | **39.84** |

## E.3 Impact of FL Optimizer

To further assess the impact of FL optimizer on the performance, we adopt FedOpt (Adam) [41], which is a SOTA FL optimizers as the base per prototype FL optimizer and conduct the same experimental setup on CV task discussed in Section xx. The results are presented in Table 11. As we can see, the performance of TAKFL is robust under different FL optimizers and is consistently achieving SOTA performance.

Table 11: **Impact of FL Optimizer.** Performance comparison results using FedOpt (Adam) [41] as per prototype FL optimizer for the same setting as Table 1 (Section 7.2).

| Dataset | Baseline | Low Data Heterogeneity (Dir(0.3)) | | | | High Data Heterogeneity (Dir(0.1)) | | | |
|---|---|---|---|---|---|---|---|---|---|
| | | S | M | L | Average | S | M | L | Average |
| CIFAR-10 | FedOpt | 35.40 | 41.69 | 58.29 | 45.12 | 23.18 | 22.03 | 37.82 | 27.67 |
| | FedDF | 43.74 | 43.48 | 63.36 | 50.19 | 29.23 | 29.46 | 37.45 | 32.04 |
| | FedET | 35.28 | 40.01 | 63.38 | 46.22 | 25.84 | 28.17 | 39.26 | 31.09 |
| | TAKFL | **55.95** | **56.25** | **65.14** | **59.11** | **39.92** | **43.26** | **45.64** | **42.94** |
| CIFAR-100 | FedOpt | 10.02 | 20.91 | 32.78 | 21.23 | 9.22 | 15.74 | 21.36 | 15.44 |
| | FedDF | 15.34 | 23.26 | 30.92 | 23.17 | 12.54 | 19.81 | 22.58 | 18.31 |
| | FedET | 9.01 | 19.59 | 32.26 | 20.28 | 6.85 | 15.30 | 23.44 | 15.19 |
| | TAKFL | **27.06** | **28.93** | **33.76** | **29.91** | **22.40** | **23.38** | **26.35** | **24.04** |

### E.4   t-SNE Visualization of Knowledge Transfer

To better understand the effectiveness of knowledge transfer in TAKFL, we conducted a visualization study using two device prototypes, XS and XL, on the CIFAR-10 dataset. The prototype configurations follow those used in our scalability evaluation in Section 7.3 (more details are available in Appendix D.4). We first pre-trained the two prototypes' global models using FedAvg for 40 communication rounds and then performed knowledge transfer using the TAKFL process for 10 distillation epochs. We have plotted the t-SNE visualizations of both prototypes for FedAvg and TAKFL in Figure 7.

As we can see in the t-SNE plots (Figure 7), TAKFL demonstrates a substantial enhancement in feature representation for both prototypes compared to FedAvg. The plots show better separation between classes for both XS and XL prototypes. The class clusters are more distinct, suggesting that TAKFL, by effectively transferring knowledge between the two prototypes, yielded a clearer representation of the features, even in the smaller XS prototype. Therefore, the t-SNE visualizations confirm that TAKFL is indeed effective in knowledge transfer, further corroborating its superiority and effectiveness.

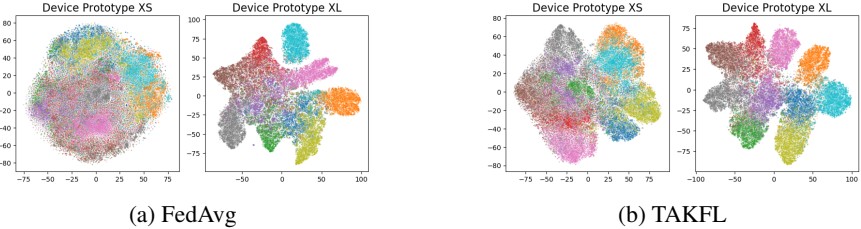

(a) FedAvg                         (b) TAKFL

Figure 7: **t-SNE Visualization Study.** To better understand the effectiveness of TAKFL's knowledge transfer, we conducted experiment on a setting involving two device prototypes XS and XL similar to our scalability evaluation setup in Section 7.3 on CIFAR-10 dataset.

# F Hyper-parameters and Implementation

In this section we bring the details of the hyper-parameters we used and our implementation. We implement our entire code in PyTorch [38] using FedZoo benchmark [36] and release it at `https://github.com/MMorafah/TAKFL`. We use two NVIDIA RTX 3090 gpus to conduct the entire experimentation in this paper.

## F.1 Computer Vision (CV) Task

For comprehensive evaluation of our method, we consider federated learning image classification training from scratch.

### F.1.1 Datasets

**Datasets.** We experiment with several image classification datasets: CIFAR-10, CIFAR-100, CINIC-10, TinyImageNet, STL-10. The details of each dataset is the following:

- **CIFAR-10** [24]: CIFAR-10 consists of 60,000 images of size 32×32 RGB across 10 classes, with each class containing 6,000 images.
- **CIFAR-100** [24]: CIFAR-100 comprises 60,000 images of size 32×32 RGB distributed across 100 classes, with 500 images per class.
- **CINIC-10** [9]: CINIC-10 has 270,000 images of size 32×32 RGB across 10 classes, each class containing 27,000 images.
- **TinyImageNet** [25]: TinyImageNet contains 100,000 images of size 64×64 RGB across 200 classes.
- **STL-10** [7]: STL-10 has 100,000 unlabeled images and 13,000 labeled images of size 96×96 RGB across 10 classes.

The ImageNet-100 as the unlabeled public dataset is constructed by randomly selecting 100 classes from the ImageNet [10] dataset.

### F.1.2 Architectures

**Experiments in Section 7.2 and Appendix D.1.** The architecture that we use are two distinct architectural settings: the *"homo-family"* setting, where all device prototypes' architectures are from the same family, and the *"hetero-family"* setting, where architectures do not necessarily belong to the same family. For the homo-family scenario, we employ ResNet-8 for S, ResNet-14 for M, and ResNet-18 for L. For the hetero-family scenario, we use ViT-S for S, ResNet-14 for M, and VGG-16 for L. All models are initialized from random initialization.

For the ResNet architecture configuration, we utilize the standard 'BasicBlock' as the building block. This consists of a convolutional block, followed by four residual block stages, an adaptive average pooling layer, and a classifier layer. The models within this family differ in terms of the number of repetitions of the residual block and the number of filters in each stage. The configurations for different capacities are detailed below:

- *ResNet-18* is configured with [64, 128, 256, 512] filters and repeats the 'BasicBlock' [2, 2, 2, 2] times.
- *ResNet-14* is configured with [64, 128, 256, 512] filters and repeats the 'BasicBlock' [1, 2, 2, 1] times.
- *ResNet-8* is configured with [64, 128, 256] filters, corresponding to the first three stages, and repeats the 'BasicBlock' [1, 1, 1] times.

For VGG-16 [44], we use the standard architecture which includes convolutional layers followed by max-pooling layers. The configuration of filters for each layer is as follows:

*VGG-16*: [64, 64, 'M', 128, 128, 'M', 256, 256, 256, 'M', 512, 512, 512, 'M', 512, 512, 512, 'M']

The final classification head consists of two linear layers with a hidden size of 512, followed by a ReLU activation, and a final linear classifier layer.

For ViT-S, we adopt the standard Vision Transformer [12] architecture implementation from Github.[3] We configure ViT-S with 6 attention blocks, each with 16 heads and a hidden dimension of 64. The final MLP dimension is set to 256. In our experiments with ViT-S, we set the patch size to 4, and the input image size is 32×32.

**Experiment in Appendix D.2.** The pre-trained MobileNetV3-Large [19], MobileViTV2 [34], and ResNet34 [51] were instantiated using the TIMM library.[4]

**Scalability Experiments in Section 7.3.** The architectures in these experiments are inspired by [20]. The details of architectures are as following:

- *ResNet10-XXS* is configured with [8, 8, 16, 16] filters and repeats the 'BasicBlock' [1, 1, 1, 1] times.
- *ResNet10-XS* is configured with [8, 16, 32, 64] filters and repeats the 'BasicBlock' [1, 1, 1, 1] times.
- *ResNet10-S* is configured with [16, 32, 64, 128] filters and repeats the 'BasicBlock' [1, 1, 1, 1] times.
- *ResNet10-M* is configured with [8, 16, 32, 64] filters and repeats the 'BasicBlock' [1, 1, 1, 1] times.
- *ResNet10* is configured with [64, 128, 256, 512] filters and repeats the 'BasicBlock' [1, 1, 1, 1] times.
- *ResNet18* is configured with [64, 128, 256, 512] filters and repeats the 'BasicBlock' [1, 1, 1, 1] times.
- *ResNet50* is configured with [64, 128, 256, 512] filters and repeats the 'BasicBlock' [3,4,6,3] times.

### F.1.3    FL configuration and Hyper-parameters

**Base Hyper-parameters.** The following hyperparameter values apply to all CV experiments unless stated otherwise. We set the diversity regularizer coefficient of FedET to 0.1 for our entire experimentation per the original paper [6]. We use the Adam optimizer with a learning rate of 1e-5, weight decay value of 5e-5, and a batch size of 128 for distillation. The softmax distillation temperature is set to 3, the distillation epoch to 1, and the self-regularizer softmax temperature to 20 for both CV and NLP experiments. Table 13 details the hyper-parameters.

**Experiments in Section 7.2 and Appendix D.1.** For tables 1 and 4, there are 100 clients with the S device prototype, 20 clients with the M device prototype, and 4 clients with the L device prototype. For each round, 10, 4, and 2 clients from each S, M, and L prototype are randomly sampled respectively for participation. 10% of the data goes to the S prototype, 30% to M, and 60% to L. The data is distributed to each client among each prototype in a Dirichlet distribution. Table 14 details the FL configuration.

**Experiments in Appendix D.2 and Appendix E.2.** For tables 5 and 10, there are 4, 3, and 2 clients for S, M, and L device prototypes, respectively. Each round, every client participates in FL. 20% of the data is distributed to prototype S, 30% to prototype M, and 50% to prototype L. Table 15 details the FL configuration.

**Scalability Experiments in Section 7.3 and D.4.** For tables 7, 8, and 9, there are 35, 30, 25, 20, 15, 10, and 5 clients for the prototypes XXS, XS, S, M, L, XL, and XXL, respectively. The sample rate is set to 0.1, 0.1, 0.15, 0.15, 0.2, 0.3, and 0.6 from XXS to XXL. The data is distributed for each prototype in the ratio 1:2:3:4:5:6:7 from XXS to XXL. Table 16 details the hyper-parameters and configuration.

**Validation Set.** For TAKFL, the validation set used for the heuristic method (see F.3) is 5% of the training dataset. The validation set and the private dataset does not overlap.

---

[3] https://github.com/lucidrains/vit-pytorch
[4] https://github.com/huggingface/pytorch-image-models

## F.2 Natural Language Processing (NLP) Task

For the NLP task, we fine-tune federated learning text classification task using pretrained models.

### F.2.1 Datasets.

All NLP datasets were provided by Hugging Face. [5]

- **MNLI** [52]: MNLI contains 433K sentence pairs, each sentence pair labeled as one of 'entailment,' 'neutral,' and 'contradiction.'

- **SNLI** [2]: SNLI is similar to MNLI, with 570K sentence pairs each labeled one of 3 labels.

- **SST2** [45]: SST2 consists of 67K phrases, each labeled as sentiment 'positive' or 'negative.'

- **Sentiment140** [14]: Sentiment140 is a dataset of 1.6M Twitter messages each labeled with one of 2 sentiment values.

- **MARC** [22]: MARC (Multilingual Amazon Reviews Corpus) is a dataset with online reviews in multiple languages from the Amazon delivery service website. Each review has a label which is one of 1-5 stars. We only use the English reviews from this dataset, which results in 260,000 English reviews total.

- **Yelp** [61]: The Yelp reviews dataset contains 700K reviews each labeled 1-5 stars from the Yelp service which publishes public reviews of businesses.

- **AG News** [60]: AG News contains 127,600 news article titles. Each article is one of four classifications of news articles.

- **DBpedia** [59]: The DBpedia dataset consists of 630K DBpedia article summaries each labeled one of 14 categorizations.

### F.2.2 Architectures

**Experiments in Section 7.2 and Appendix D.3.** We use three variations of the BERT architecture: BERT-Tiny, BERT-Mini, and BERT-Small from [47]. The weights were pre-trained on the BookCorpus dataset and extracted text from Wikipedia. Further details regarding each model are described extensively on Github.[6] The tokenizer used for these transformer models are the same ones provided by the authors of [47].

- *BERT-Tiny* contains 2 transformer layers and an embedding size of 128.
- *BERT-Mini* contains 4 transformer layers and an embedding size of 256.
- *BERT-Small* contains 4 transformer layers and and embedding size of 512.

### F.2.3 FL configuration and hyper-parameters

**Base Hyper-parameters.** For distillation, we use the Adam optimizer with a learning rate of 3e-5, no weight decay, and batch size of 32. The distillation epoch is set to 1, the ensemble distillation softmax temperature to 3, and the self-regularizer softmax temperature to 20 for all NLP experiments. Table 17 details the hyper-parameters.

**Experiments in Section 7.2 and Appendix D.3.** For tables 2 and 6, we limit the private dataset to 100,000 samples, randomly sampled from the original dataset i.i.d. The public dataset is limited to 30,000 examples sampled i.i.d as well. There are 8, 4, and 2 clients for the S, M, and L prototypes. The private data is split across each prototype in the following proportions: 0.1, 0.3, 0.6. Table 18 details the FL configuration.

**Validation Set.** The validation dataset used for TAKFL is 5,000 samples taken from the original training dataset that does not overlap with the 100,000 private dataset.

---

[5] https://github.com/huggingface/datasets
[6] https://github.com/google-research/bert

### F.3 Hyper-parameters of TAKFL

**Merging Coefficients.** We conducted extensive experiments with different merging coefficients on the main 3-device prototype setting of small (S), medium (M), and large (L) discussed in Section 7.2 and Appendix D.1. We empirically observed that the small (S) prototype typically achieves the best performance using a uniformly increasing merging coefficient, where the larger the prototype, the larger the merging coefficient, i.e., $\lambda_S \leq \lambda_M \leq \lambda_L$. As we move towards larger prototypes, they benefit more from increasingly skewed merging coefficients towards the larger ones. In the extreme case of the large (L) prototype, highly skewed merging coefficients generally led to better performance, i.e., $\lambda_S \ll \lambda_M \ll \lambda_L$. This pattern is intuitive as small prototypes can benefit from everyone while gaining more from the larger, more informative prototypes. However, larger prototypes benefit less from smaller ones, as they typically offer less information, especially in high data heterogeneity cases. Notably, in high data heterogeneity cases, more skewed merging coefficients seemed to be more advantageous as the smaller prototypes (S and M) possess lower quality knowledge.

Based on these observations, we designed a simple and cost-effective heuristic method that randomly instantiates merging coefficients following this intuition. Our heuristic method, presented in 1, leverages these observations by generating candidate merging coefficients that incorporate both uniformly increasing and different degrees of skewed merging coefficients. This dual approach enables us to explore a wide range of merging strategies and identify the most effective configurations for different prototypes. The optimal merging coefficient candidate is determined using the performance on the held-out validation set.

```
1  import numpy as np
2  def heuristic(num_devices=3, n_candidates=10):
3      candidates = [[1/num_devices for _ in range(num_devices)]]
4      for exponent in [1, 5, 10]:
5          for i in range(n_candidates):
6              candidate = np.random.beta(a=1, b=100, size=num_devices)
7              candidate = candidate ** exponent
8              candidate = np.sort(candidate)
9              candidate = candidate / np.sum(candidate)
10             candidates.append(candidate)
11     return candidates
```

Listing 1: Implementation of the heuristic method for merging coefficients in Python. The exponent term controls the degree of skewness or peaking in the merging coefficients.

Furthermore, we experiment with manually determining the merging coefficients and fixating them throughout the federation. We achieved similar results with this approach compared to adaptively finding the coefficients using the heuristic method and a small held-out validation set. We present the merging coefficient candidates that performed reasonably well during our experiments in Table 12.

Table 12: Details of the experimentally determined merging coefficients for the 3-device prototype setting discussed in Section 7.2 and Appendix D.1. Coefficients are ordered as $[\lambda_S, \lambda_M, \lambda_L]$.

| Merging Coefficient Candidate | Small Prototype | Medium Prototype | Large Prototype |
|:---:|:---:|:---:|:---:|
| 1 | [0.2, 0.3, 0.5] | [0.1, 0.2, 0.7] | [0.1, 0.2, 0.7] |
| 2 | [0.3, 0.3, 0.4] | [0.05, 0.15, 0.8] | [0.01, 0.09, 0.99] |
| 3 | [0.2, 0.3, 0.5] | [0.1, 0.2, 0.7] | [0.05, 0.2, 0.75] |
| 4 | [0.05, 0.1, 0.85] | [0.01, 0.19, 0.8] | [0.01, 0.09, 0.90] |
| 5 | [0.1, 0.15, 0.75] | [0.05, 0.15, 0.8] | [0.01, 0.09, 0.90] |
| 6 | [0.05, 0.1, 0.85] | [0.05, 0.05, 0.9] | [0.001, 0.009, 0.99] |
| 7 | [0.05, 0.15, 0.80] | [0.05, 0.2, 0.75] | [0.001, 0.009, 0.99] |
| 8 | [0.05, 0.15, 0.80] | [0.05, 0.1, 0.85] | [0.001, 0.009, 0.99] |
| 9 | [0.3, 0.35, 0.35] | [0.2, 0.3, 0.5] | [0.1, 0.2, 0.7] |

**Self-Regularization Coefficient.** Extensive experiments were conducted on the self-regulation coefficients for different device prototypes and settings. Although no consistent pattern emerged, we experimentally determined that optimal performance for the small prototype was achieved with self-

regulation coefficients $\gamma_S \in 0.1, 0.01, 0.001$. For the medium prototype, the coefficients were $\gamma_M \in 0.5, 0.1, 0.01, 0.001, 0.0001$, and for the large prototype, $\gamma_L \in 1.0, 0.8, 0.5, 0.1, 0.01, 0.001, 0.0001$ yielded the best results.

Table 13: Details of hyper-parameters for CV task in Section 7.2, Appendix D.1, and Appendix E.2.

| Local/Server | Hyperparameter | Small Prototype | Medium Prototype | Large Prototype |
|---|---|---|---|---|
| | Training epochs | 20 | 80 | 100 |
| | Batch Size | 64 | 64 | 64 |
| | Optimizer | Adam | Adam | Adam |
| Local Training | Learning Rate | 1e-3 | 1e-3 | 1e-3 |
| | Weight Decay | 5e-5 | 5e-5 | 5e-5 |
| | LR scheduler | None | None | StepLR(step_size = 10, gamma = 0.1) |
| | Optimizer | Adam | Adam | Adam |
| | Learning Rate | 1e-5 | 1e-5 | 1e-5 |
| | Weight Decay | 5e-5 | 5e-5 | 5e-5 |
| Server KD Training | Batch Size | 128 | 128 | 128 |
| | Training Epochs | 1 | 1 | 1 |
| | Ensemble Distillation Softmax Temperature | 3 | 3 | 3 |
| | Self-Regularizer Softmax Temperature | 20 | 20 | 20 |

Table 14: Details of Architecture parameters and FL configuration for CV task in Section 7.2 and Appendix D.1.

| Architecture Setting | Device Prototype | Architecture | CIFAR-10 Parameters | CIFAR-100 Parameters | Dataset Portion | Clients | Sample Rate |
|---|---|---|---|---|---|---|---|
| | Prototype S | ResNet8 | 1.23M | 1.25M | 0.1 | 100 | 0.1 |
| Homo-Family | Prototype M | ResNet14 | 6.38M | 6.43M | 0.3 | 20 | 0.2 |
| | Prototype L | ResNet18 | 11.17M | 11.22M | 0.6 | 4 | 0.5 |
| | Prototype S | ViT-S | 1.78M | 1.79M | 0.1 | 100 | 0.1 |
| Hetero-Family | Prototype M | ResNet14 | 6.38M | 6.43M | 0.3 | 20 | 0.2 |
| | Prototype L | VGG16 | 15.25M | 15.30M | 0.6 | 4 | 0.5 |

Table 15: Details of Architecture parameters and FL configuration for CV task experiment using pre-trained models in Appendix D.2.

| Device Prototype | Architecture | STL-10/CINIC-10 Parameters | TinyImageNet Parameters | Dataset Portion | Clients | Sample Rate |
|---|---|---|---|---|---|---|
| Prototype S | mobilenetv3-large-100 | 4.21M | 4.45M | 0.2 | 4 | 1.0 |
| Prototype M | mobilevitv2-175 | 13.36M | 13.53M | 0.3 | 3 | 1.0 |
| Prototype L | ResNet34 | 21.28M | 21.38M | 0.5 | 2 | 1.0 |

Table 16: Details of Architecture parameters for Scalability Section 7.3, and Appendix D.4.

| Device Prototype | CINIC-10 | | | | | |
|---|---|---|---|---|---|---|
| | Architecture | Parameters | Dataset Portion | Clients | Sample Rate | Local Epochs |
| Prototype XXS | ResNet10-XXS | 11K | 0.0357 | 35 | 0.1 | 2 |
| Prototype XS | ResNet10-XS | 78K | 0.0714 | 30 | 0.1 | 2 |
| Prototype S | ResNet10-S | 309K | 0.1071 | 25 | 0.15 | 2 |
| Prototype M | ResNet10-M | 1.2M | 0.1428 | 20 | 0.15 | 5 |
| Prototype L | ResNet10 | 4.9M | 0.1785 | 15 | 0.2 | 10 |
| Prototype XL | ResNet18 | 11M | 0.2142 | 10 | 0.3 | 10 |
| Prototype XXL | ResNet50 | 24M | 0.25 | 5 | 0.6 | 20 |

Table 17: Details of hyper-parameters for NLP task experiments in Section 7.2 and Appendix D.3.

| Local/Server | Hyperparameter | Prototype S | Prototype M | Prototype L |
|---|---|---|---|---|
| Local Training | training epochs | 1 | 1 | 1 |
| | batch size | 32 | 32 | 32 |
| | optimizer | Adam | Adam | Adam |
| | Learning rate | 3e-5 | 3e-5 | 3e-5 |
| | Weight decay | 0 | 0 | 0 |
| | lr scheduler | None | None | None |
| Server KD Training | optimizer | Adam | Adam | Adam |
| | learning rate | 3e-5 | 3e-5 | 3e-5 |
| | weight decay | 3e-5 | 3e-5 | 3e-5 |
| | batch size | 32 | 32 | 32 |
| | training epochs | 1 | 1 | 1 |
| | Ensemble distillation softmax temperature | 3 | 3 | 3 |
| | Self-regularizer softmax temperature | 20 | 20 | 20 |

Table 18: Details of Architecture parameters and FL configuration for NLP task experiment using pre-trained models in Section 7.2 and Appendix D.3.

| Device Prototype | Architecture | Parameters | Clients | Dataset Portion | Sample Rate |
|---|---|---|---|---|---|
| Prototype S | BERT-Tiny | 4.39M | 8 | 0.1 | 0.4 |
| Prototype M | BERT-Mini | 11.17M | 4 | 0.3 | 0.5 |
| Prototype L | BERT-Small | 28.77M | 2 | 0.6 | 1.0 |

