# OpenReview forum: "Towards Diverse Device Heterogeneous Federated Learning via Task Arithmetic Knowledge Integration"
_NeurIPS.cc/2024/Conference — NeurIPS 2024 poster_

### Official Review · Reviewer_ME5v · 2024-06-16

**Soundness:** 2
**Presentation:** 3
**Contribution:** 2
**Rating:** 5
**Confidence:** 3

**Summary:**

The authors aim to develop a knowledge distillation method that addresses the challenges posed by heterogeneous device prototypes in federated learning. By capturing the knowledge transfer among device prototypes, the proposed TAKFL tries to preserve each device's unique contribution and prevent knowledge dilution during the learning procedure. The method incorporates a self-regularization technique to address issues of the noisy and unsupervised ensemble distillation. Evaluation of some CV and NLP tasks shows the performance of model accuracy and scalability.

**Strengths:**

The authors focus on knowledge distillation in heterogeneous settings, which is meaningful to real-world tasks. The overall presentation is clear and easy to understand. The proposed TAKFL shows some primary theoretical analysis of the learning efficiency. The evaluation provides a quantitative analysis to compare model accuracy and scalability with previous methods.

**Weaknesses:**

While the paper presents a clear presentation and evaluation, there are a few aspects to be strengthened.

1. The concept of "task arithmetic" in the context of federated learning is vague. It would be beneficial to explain how this concept enhances the design of the federated learning process.
2. While the knowledge distillation process is described, it appears to be largely based on existing methodologies. It would be interesting to explore any novel design elements introduced in TAKFL. Additionally, what is the theoretical convergence order of the proposed method?
3. Although TAKFL demonstrates higher model accuracy in performance comparisons, the baselines used, such as FedAvg and FedDF, seem outdated. Incorporating more recent baselines could provide a more rigorous evaluation of TAKFL's effectiveness.

**Questions:**

Please refer to the concerns and questions mentioned in the weakness part.

---

> ### Author Rebuttal · Authors · 2024-08-07
>
> Thank you for your invaluable review and pointing out that our proposed method shows **some primary theoretical analysis of the learning efficiency**.
>
> ### **Response to 1**
> To clarify, we use the concept of task arithmetic to address limitations of previous approaches and enhance knowledge transfer across heterogeneous device prototypes in FL. We extend the notion of “task vector” from centralized learning to FL by considering the averaged locally updated parameters as the “pre-trained” model parameters and the distilled model parameters as the “fine-tuned” parameters (refer to Figure 2 in paper). Task vectors encapsulate the distinct contributions of each prototype's ensembles to the student model, representing transferred knowledge (Section 5.2).
>
> Our task arithmetic approach enhances the design of knowledge transfer across diverse heterogeneous device prototypes in FL by:
>
> **(1) Effective Distillation of Unique Knowledge:** Task arithmetic facilitates the distillation of unique knowledge from each prototype’s ensembles. Prior works neglect individual strengths and average logits, causing dilution and information loss (Sections 4, 5.1, Remark 1).
>
> **(2) Customized Knowledge Integration:** Our adaptive task arithmetic operation (Eq 8) allows the student model to customize knowledge integration based on its capacity and the helpfulness of other prototypes, enhancing performance. Prior works use a one-size-fits-all approach, ignoring the student model’s capacity and prototypes’ helpfulness (Sections 4, 5.2, Remark 2).
>
> We have further verified the enhancements of our task arithmetic approach through both theoretical analysis and extensive experiments (Sections 6, 7, D.1.1). To the best of our knowledge, our work is the first of its kind to introduce the notion of “task vector” in FL to enhance knowledge transfer across diverse heterogeneous device prototypes.
>
> ### **Response to 2**
> We highlight the key distinctions and novelties of our knowledge distillation methodology compared to existing methods:
>
> **(1) Diversity in Device Capabilities:** Existing methods often overlook device capability diversity and apply logit averaging, causing dilution and interference. TAKFL addresses this by treating knowledge transfer from each prototype as separate tasks, distilling them independently to ensure unique contributions are captured without interference.
>
> **(2) Customized Knowledge Integration:** Existing methods use a single integrated distillation target for all student models, failing to provide customized knowledge integration. TAKFL employs task vectors in FL as a unique representation of transferred knowledge from each prototype. This enables the student model to customize knowledge integration based on each prototype's knowledge quality and its own capacity via adaptive task arithmetic (Eq 8), enhancing performance.
>
> **(3) Handling Noisy and Unsupervised Distillation:** Existing methods often neglect challenges associated with noisy and unsupervised ensemble distillation, which may cause the student to forget its own knowledge and drift into erroneous directions. TAKFL incorporates a novel KD-based self-regularization technique to mitigate these issues.
>
> Regarding the theoretical convergence, TAKFL consists of two processes in each round: (1) per-prototype FL and (2) server-side knowledge distillation and integration via task arithmetic. The convergence order of per-prototype FL is the same as, FedAvg [1], with various per-prototype FL optimizer options, and the server-side KD is similar to centralized neural network training  [2], as the KL loss has the same functional form as cross-entropy. Thus, the complete task arithmetic model merging convergence is simply the sum of these two procedures. We will include this in a Remark in the final revision of our paper.
>
> We have also obtained the convergence plot for (CIFAR-10, Dir(0.3)) in Table 1 of the paper, included in Figure 2 of the PDF rebuttal. TAKFL shows similar convergence behavior to the baselines while achieving better performance. We will include this plot in the final version of the paper.
>
> ### **Response to 3**
> We appreciate the reviewer highlighting TAKFL's higher performance compared to the baselines. Existing body of works on global device heterogeneous FL consists of two distinct approaches and problem formulations:
>
> **(1) Partial Model Training Methods:** These aim to train a single global model while accommodating heterogeneous sub-models at the devices according to their compute resources, with several SOTA baselines.
>
> **(2) Knowledge Distillation Methods:** These address a more practical scenario where device prototypes with heterogeneous model architectures participate in FL to enhance their global model performance through knowledge distillation. The current SOTA baselines are FedDF and FedET, which we have comprehensively compared in our work. These methods mainly focus on settings where device prototypes have similar capabilities.
>
> Our work follows (2) and introduces TAKFL to address limitations of existing methods in the *underexplored* diverse device heterogeneous settings. We believe our work can significantly contribute to the FL literature and establish a new SOTA benchmark. For a detailed discussion on related works, we kindly refer the reviewer to Appendix A, where we reference recent surveys from 2023 and 2024.
> To further evaluate TAKFL's effectiveness, we experimented using FedOpt [3], a more advanced FL optimizer for per-prototype FL, and included the results in Table 3 of the PDF rebuttal. As shown, TAKFL achieves SOTA performance results independent of the specific FL optimizer used. We will include this experiment in the final version of our paper.
>
> ### References
> [1] Haddadpour, et al. "Local sgd with periodic averaging: Tighter analysis and adaptive synchronization." NeurIPS’19
>
> [2] Aggarwal. “Neural networks and deep learning”. Springer (2018)
>
> [3] Reddi et al. "Adaptive Federated Optimization." ICLR’21

---

> > ### Author Response · Authors · 2024-08-12
> > **Kind Reminder: End of Author-Reviewer Discussion Period is Tomorrow**
> >
> > Dear Reviewer ME5v,
> >
> > We sincerely appreciate your extensive efforts in reviewing and commenting on our work. We hope that our rebuttal has comprehensively addressed your comments and concerns.
> >
> > As the author-reviewer discussion period is approaching to its end by tomorrow on August 13th, we kindly ask you to reach out if you have any further questions or points of discussion. Your feedback is highly valued, and we are eager to address any additional remaining concerns you may have.
> >
> > Thank you very much for your attention to this matter.

---

### Official Review · Reviewer_F7vu · 2024-07-09

**Soundness:** 3
**Presentation:** 3
**Contribution:** 3
**Rating:** 5
**Confidence:** 4

**Summary:**

The paper focus on a problem that traditional federated learning methods fail to effectively handle scenarios where devices have widely varying capabilities. It improve existing Knowledge Distillation (KD) methods that are inadequate in these heterogeneous environments. Experimental results show the validity of their proposed method.

**Strengths:**

1. TAKFL treats knowledge transfer from each device prototype’s ensembles as separate tasks and distills them independently.
2. The paper is well written with comprehensive experiments

**Weaknesses:**

1. Assumption on Weight Disentanglement Property: The theorem 2 reply on the assumption of the weight disentanglement property  (line 679-681 in appendix)  is  too strong. In practice, achieving weight disentanglement is challenging. Studies [1,2] demonstrate that there are interferences among task vectors, making disentanglement difficult. [3] achieves disentanglement using Neural Tangent Kernel (NTK) and shows without disentanglement the performance is dropped. Consequently, asserting the disentanglement property is problematic, thereby limiting the theoretical impact.

2. Similar to weight conflicts when doing merge, averge logits also have conflicts, the paper only considers vanilla average logit as the KD loss. However, there are studies to resolve the issue, like [4,5] have proposed better ensemble KD loss designs, more studies online. Therefore, incorporating these methods for comparison is important.

3. The computation cost of the method is high as the number of distillation process is O(M^2) which exponentially increases with the prototype number.

4. The changes compared to [6] are minor. Initially, I believed this method could be simple and efficient. However, upon reviewing the weight disentanglement property, I have concerns about its practical validity, which limits the novelty of the approach.

5. A minor issue is only apply to data with prototype label, thus may limited its impact

[1] TIES-Merging: Resolving Interference When Merging Models

[2]  Language Models are Super Mario: Absorbing Abilities from Homologous Models as a Free Lunch

[3] Task Arithmetic in the Tangent Space: Improved Editing of Pre-Trained Models

[4] Adaptive Multi-Teacher Multi-level Knowledge Distillation

[5] Agree to Disagree: Adaptive Ensemble Knowledge Distillation in Gradient Space

[6] Ensemble distillation for robust model fusion in federated learning.

**Questions:**

please refer to Weakness.

**Limitations:**

please refer to Weakness.

---

> ### Author Rebuttal · Authors · 2024-08-07
>
> Thank you for your invaluable review and positive remarks on the strengths of our paper in **improving existing KD methods that are inadequate in diverse device heterogeneous environments** and **comprehensive experiments**.
>
> ### **Response to 1**
> Studies [1,2] primarily address parameter interference and redundancy, proposing methods to mitigate this. These studies focus on parameter interference, not weight disentanglement. Parameter interference is not equivalent to weight entanglement.
>
> Study [3] (NeurIPS’23), our cited reference (reference [37]), provides theoretical insights into task arithmetic, introducing weight disentanglement as a necessary condition. They show that “task arithmetic in non-linear models cannot be explained by NTK,” and weight disentanglement is actually the sole requirement for task arithmetic “where distinct directions in weight space correspond to changes of the network in disjoint regions of the input space ... This allows a model to perform task arithmetic by independently manipulating these weight directions.” They also demonstrate that weight disentanglement is an emergent property of pre-training and propose fine-tuning models in their tangent space to enhance this property. Please note that the quotes are the original text from [3]. Therefore, asserting weight disentanglement property is not problematic per [3].
>
> In our theoretical framework, we have cited [3] in our paper for weight disentanglement as it is the established theoretical framework to study task arithmetic. Our theoretical framework makes the following contributions: (1) It present the first theoretical framework using the concept of capacity dimension to understand the effectiveness of KD in device heterogeneous FL. (2) Our theoretical analysis of TAKFL is based on the theory in [3] (3) It presents an argument consistent with novel numerical experimental findings, showing that TAKFL performs KD without information loss more effectively.
>
> ### **Response to 2**
> We used vanilla average logits KD in our methodology for knowledge distillation from individual prototypes because it is the most representative and canonical form of KD. This avoids any bias towards the advantages or disadvantages of specific KD implementations, making our comparisons more transparent and informative.
>
> Regarding [4,5], these methods are designed for KD in a centralized learning setting, where teachers are fully trained, and the distillation dataset aligns with actual learning objective, including a cross-entropy supervision loss. However, in FL, the ensembles are noisy, and the distillation dataset differs from the private dataset, lacking a supervision cross-entropy loss (Section 5.1).
>
> With these considerations, we have compared TAKFL by directly implementing [4,5] in FL as well as incorporating them into TAKFL instead of vanilla KD. The results are presented in Table 2 of the PDF rebuttal. As seen, using [4,5] directly in FL performed poorly, while incorporating [5] into TAKFL achieved better performance in low data heterogeneity compared to TAKFL+Vanilla KD. TAKFL is independent of the specific KD method used. We will incorporate these comparisons into the final version of our paper.
>
> ### **Response to 3**
> We have presented the full algorithm details of TAKFL in Appendix B. All the distillation processes are performed in parallel and asynchronously (lines 9-10 in Algorithm 1). Therefore, the overall computation cost of the entire server-side distillation process is O(M), as device prototypes do not perform KD one at a time but combine information asynchronously into a singe task arithmetic operation (Eq. 8). Additionally, in FL, the server typically has substantial resources to manage the orchestration of many devices, making server-side computation cost not a concern.
>
> ### **Response to 4**
> We highlight the key distinctions and novelties of TAKFL compared to [6]:
>
> (1) [6] overlooks the diversity in device capabilities and applies uniform logit averaging, causing dilution and information loss. TAKFL addresses this by treating knowledge transfer from each prototype's ensembles as separate tasks, distilling them independently to ensure unique contributions are effectively captured.
>
> (2) [6] uses average logits as the distillation target for all student models, failing to provide customized knowledge integration. TAKFL employs a novel task arithmetic approach, allowing each student model to customize knowledge integration based on each prototype's quality and its own capacity, enhancing performance.
>
> (3) [6] neglect challenges associated with noisy and unsupervised ensemble distillation. TAKFL incorporates a novel KD-based self-regularization technique to mitigate these issues.
>
> Our theoretical and experimental findings highlight the deficiencies of [6] in diverse device heterogeneous FL (Remark 1, Proposition 1, Section 7.2, Appendix D.1.1). Our extensive experiments show that TAKFL not only outperforms existing methods but also offers a novel, simple, and efficient solution to a significant practical challenge. In regards to the weight disentanglement property, kindly please see our response to #1 above.
>
> ### **Response to 5**
> It seems that the reviewer is concerned about the impact of TAKFL. We have comprehensively evaluated TAKFL for CV and NLP tasks, including various datasets with differing architectures and data heterogeneity levels. We also introduced a new scalability evaluation showing TAKFL’s adaptability across devices ranging from XXS to XXL, underscoring its broad applicability and consistent performance improvements over existing methods. We believe that the impact of our method extends beyond specific datasets and prototype sizes, providing a versatile solution for diverse device heterogeneous federated learning applications. If this does not address the reviewer’s comment, could the reviewer please clarify and elaborate? The term “data with prototype label” is unclear to us.

---

> ### Comment · Reviewer_F7vu · 2024-08-09
>
> **Question 1:**
>
> Thank you for your response. I still insist that using disentanglement theory as proof is not fair because the model cannot achieve perfect disentanglement. In equation 6 of [3], they define disentanglement error to measure the weight disentanglement of a model. In Fig. 3 of [3], they found that non-linear models are not weight-disentangled enough, but linearized models are more weight-disentangled. Thus, they enhance task arithmetic performance via linearization, fine-tuning in the tangent space to their pre-trained initialization, which is the same as training a kernel predictor with kernel kNTK (as shown on pages 6-7 in [3]).
>
> Normally, due to the difficulty in achieving weight disentanglement, there is interference within different task vectors (which could be introduced by non-perfect disjoint regions of the input space). As a result, recent studies focus on resolving the interference when combining different task vectors [1,2,7].
>
> Therefore, assuming perfect weight disentanglement is unfair when comparing it with logit distillation. I feel that one way to consider a more concrete theory might be by showing that the degree of influence on the parameter space is smaller than in the logit space.
>
> **Question 2:**
>
> Thank you for your efforts in conducting extra experiments. In Table 2 in  rebuttal pdf the second row should be CIFAR-100?
> May I know how the new knowledge distillation losses were added? Were they used to replace the KD loss in FedDF? Because it is a little surprising that they perform much worse than the original FedDF.
>
> **Question 3:**
>
> It is a little tricky to say that parallelization can reduce the computational cost, since the amount of resource usage still grows in O(M^2).
>
> If you can resolve these remaining issues, I will increase my score.
>
> [1] TIES-Merging: Resolving Interference When Merging Models
>
> [2]   Language Models are Super Mario: Absorbing Abilities from Homologous Models as a Free Lunch
>
> [3]  Task Arithmetic in the Tangent Space: Improved Editing of Pre-Trained Models
>
> [7] AdaMerging: Adaptive Model Merging for Multi-Task Learning

---

> ### Author Response · Authors · 2024-08-10
> **Response to Q1**
>
> We appreciate the reviewer’s detailed feedback and engagement during the discussion period.
>
> ### **Response to Q1**
> We would like to clarify our position on the use of weight disentanglement (WD) in our work.
>
> **Response to Weight Disentanglement Concerns**
>
> Task Arithmetic (TA) was first introduced in reference [51] of our paper and has demonstrated practical effectiveness. [3] provides the theoretical foundation for TA, identifying WD as the only necessary condition to perform TA effectively. Importantly, [3] demonstrates that WD is an emergent property during pre-training, absent in randomly initialized models (Section 6.2 of [3]). Fig 3 of [3] shows that WD remains sufficiently strong within the merging coefficients range (0, 1] across tasks, even without linearization. Linearization primarily enhances WD outside this range, suggesting that models are weight-disentangled enough in practice to enable task arithmetic and achieve strong performance within the effective range.
> This effective range is crucial, as both the original TA paper [51] and [2,3] set task vector coefficients within (0, 1], ensuring that WD is maintained. Additionally, [7] uses `torch.clamp` to constraint task vector coefficients within this range, further validating the effectiveness of WD in this context. Our work, following [51,2,3,7], also ensures that the merging coefficients remain within this range, supporting robust WD and effective TA (Eq. 8, Section 5.2, Appendix F.3).
>
> **Response to Studies [1,2,7]**
>
> Studies [1,2] primarily address parameter interference and redundancy rather than WD. [1] discusses interference between task vectors due to redundant parameters and sign disagreement, proposing TIES-MERGING to mitigate this. [2] identifies redundant delta parameters in SFT LMs and introduces DARE to enhance model merging. [7] suggests that merging coefficients play a pivotal role in the average accuracy of the final MTL model and proposes AdaMerging to optimize these coefficients.
>
> We couldn’t find any argument in studies [1,2,7] stating that achieving WD is difficult, causing interference within different task vectors. It does not appear to us that there is any formal correspondence between parameter interference and redundancy as these are considered in [1,2,7] and the notion of  WD as defined in Eq. 4 of [3]. We would appreciate it if the reviewer could please clarify and elaborate, which would help us provide a more comprehensive response regarding [1,2,7]. Moreover, if the reviewer suggests that [7] handles the difficulty better through adaptive TA, it is important to note that our approach also employs adaptive TA, ensuring that we address this challenge effectively.
>
> **Conclusion**
>
> In conclusion, while perfect WD across all merging coefficient ranges is challenging to achieve, the strong WD within the effective range (0, 1] is sufficient for TA to perform well. Our work, following [51,2,3,7], ensures that the merging coefficients remain within this range, maintaining the robustness of WD and enabling effective TA as it is evident from the improvements in our extensive experiments across both CV and NLP tasks.
>
> Finally we believe that this concern is largely orthogonal to the specific theoretical results we claim. Regardless, we believe that this assumption is pedagogically useful, providing clarity and focus in our theoretical results without overcomplicating the analysis. The key takeaway is the "Delta" of information retention/loss between KD and TAKFL, which remains consistent regardless of the baseline error rate. Even if WD does not hold generically, our theoretical results are still informative as they indicate behavior in certain idealized, limited behavior circumstances, in particular cases with significant degrees of freedom of capacity. This is common practice in studying Federated Learning. Introducing additional complexity to account for WD variations would make the results more opaque without significantly altering the conclusions.
>
> We appreciate the reviewer's continued engagement in the discussion and are happy to address any further questions or concerns.

---

> ### Author Response · Authors · 2024-08-10
> **Response to Q2 and Q3**
>
> We appreciate the reviewer’s detailed feedback and engagement during the discussion period.
>
> ### **Response to Q2**
>
> Yes, the second row in Table 2 in the rebuttal PDF file refers to CIFAR-100. We apologize for any confusion, as we were working diligently to prepare the results. To address the reviewer's question, we directly replaced the KD loss in FedDF with AMTML-KD [4] and AE [5] in Table 2, resulting in the baselines labeled AMTML-KD [4] and AE [5]. Additionally, we replaced the vanilla KD loss in our TAKFL framework with [4,5], resulting in baselines TAKFL+[4] and TAKFL+[5].
>
> The reason why [4,5] are performing worse in FL could be that these methods are designed for KD in a centralized learning setting, where teachers or ensembles are fully trained and robust, and the distillation dataset aligns with the actual learning objective, including a cross-entropy supervision loss. However, in FL, the ensembles are noisy, and the distillation dataset differs from the private dataset, lacking a supervision cross-entropy loss. This discrepancy likely contributes to the reduced effectiveness of these methods in the device heterogeneous FL setting.
>
> ### **Response to Q3**
>
> We appreciate the reviewer's comment and would like to clarify the distinctions between overall computation time, computational load, and resource usage.
>
> * **Computation Time:** TAKFL's computation time is O(1) (constant) due to parallelization, as all distillation processes occur simultaneously.
> * **Computation Load:** The overall computational load scales as O(M) (linear) since the distillation tasks are performed independently for each prototype in parallel and merged into a singe task arithmetic operation (Eq. 8).
> * **Resource Usage (Memory):** The resource usage scales as O(M^2) (quadratic) because of the need to store and process multiple task vectors concurrently.
>
> While it's true that resource usage grows with O(M^2), it’s important to note that the entire KD process occurs on the server side. In FL, servers typically possess substantial resources. Given these resources, the resource usage incurred by TAKFL is generally less of a concern in practical applications. Moreover, in real-world scenarios, the number of device prototypes (M) tends to be limited, often encompassing categories such as IoT devices, mobile devices, and workstations. Therefore, while the theoretical resource usage grows with O(M^2), the practical impact is mitigated by the server's capabilities and the limited number of prototypes involved.
>
> We appreciate the reviewer's continued engagement in the discussion and are happy to address any further questions or concerns.

---

> ### Comment · Reviewer_F7vu · 2024-08-11
>
> Thank you response.
>
> Q1: First, I agree with the author's conclusion that "while perfect WD across all merging coefficient ranges is challenging to achieve, strong WD within the effective range (0, 1] is possible." Then, as stated in [3], the WD refers to different tasks (classes) as shown in the problem statement in Section 2, Property 3, and Fig. 6. In Property 1 (Task Arithmetic) of [3], it is noted that this occurs with non-intersecting task supports $\mathcal{D}_{t'} \cap \mathcal{D}_t = \emptyset$.
>
> However, in the problem setting (lines 131-135), this paper focuses on various network architectures, and in Table 13 of the appendix, it is shown that heterogeneity occurring within each device setting.  As a result  $\mathcal{D}_{t'} \cap \mathcal{D}_t \neq \emptyset$, thus might be too strong to use this assumption.
>
> Q2: the answer is clear to me.
> Q3: the answer is clear to me.
>
> As a result,  the inadequate Theory, the application of task arithmetic to FedDF, limited the novelty of this paper.

---

> > ### Comment · Reviewer_F7vu · 2024-08-11
> >
> > Although the novelty is limited, I will raise my score from 4 to 5 based on the authors' experiments. If there are any additional points I may have overlooked, I'm open to discussing them.

---

> ### Author Response · Authors · 2024-08-12
> **Response to Q1**
>
> We would like to extend our sincere gratitude for the reviewer's great effort in reviewing our work and continued engagement in the discussion. We also appreciate the reviewer for raising his score to 5.
>
> ### **Response to Q1**
> To answer Q1, we would like to note that the goal of device prototypes $i$ and $j$ participating in FL and doing server-side knowledge transfer with each other is benefiting and learning from their underlying datasets $\mathcal{D^i}$ and $\mathcal{D^j}$ (as noted in problem formulation Eq. 1), where indeed $\mathcal{D^i} \cap \mathcal{D^j} = \emptyset$ and $\bigcup_{i=1}^{M} \mathcal{D^i} = \mathcal{D}$ (lines 618-619, 679-680).
>
> In all of our experimentation, the data partitions of prototypes, i.e. $\mathcal{D^i}$'s, are all indeed disjoint with no sample overlap. Therefore,  $\mathcal{D^i} \cap \mathcal{D^j} = \emptyset$ does hold in our work and theoretical framework.
>
> We hope that this answers the reviewer's question.
>
> Once again, we appreciate the reviewer's continued engagement in the discussion and are happy to address any further questions or concerns in the remaining time till end of discussion period by tomorrow on August 13th.

---

> ### Comment · Reviewer_F7vu · 2024-08-12
>
> Your response is not true because even though you don't have overlapping samples, the classes themselves overlap.
>
> As mentioned in the problem statement in [3]:  Consider $T $ tasks, with every task $ t $ consisting of a triplet $\mathcal{D}_t, \mu_t, f_t^*) $ where $ f_t^* : \mathcal{D}_t \to \mathcal{Y} $ a target function (e.g., labels).
>
> Additionally, as illustrated in Fig. 6 in [3], the eigenfunction localization clearly demonstrates disentanglement in the case of $ x \in RESISC45 $ and $ x \in cars $. The former refers to Remote Sensing Image Scene Classification, while the latter pertains to car classification. The labeling functions $f_t^*$ are significantly different.
>
> However, in your data splitting, if we denote the data distribution of class $ c $ as $ D_c$, for any prototype group $ i $, there are samples belonging to the class $ x^i_c \in D_c $. Consequently, the intersection among different prototypes is not a null set $\mathcal{D}_{t'} \cap \mathcal{D}_t \neq \emptyset$.
>
> As a result,  the Theory is inadequate.

---

> ### Author Response · Authors · 2024-08-13
> **Response to Reviewer F7vu**
>
> We appreciate the reviewer’s attention to the technical details of weight disentanglement (WD). Formally, the reviewer is correct that the WD theory in [3] assumes exclusively partitioned class labels across device prototypes. In our experiments, there may be overlap in class labels, meaning our setup does not fully satisfy the assumption as formally presented in [3].
>
> However, we contend that this overlap does not undermine the theory’s applicability. When class labels overlap, the task vectors associated with training data from different prototypes can include vectors in the tangent space of both prototypes’ loss surfaces. This overlap creates an i.i.d. situation, which can actually make knowledge distillation (KD) easier, with the Vanilla Average Logits method likely experiencing minimal information loss.
>
> For instance, the task vectors corresponding to training on device prototype 2's data may now contain vectors in the tangent space of the loss surface of device prototype 1's architecture, fit to data exclusively belonging to prototype 2's original training samples. Additionally, vectors could now exist in the tangent space of the loss surface of device prototype 1's architecture, fit to prototype 1's data, due to the overlapping classes. This overlap creates an i.i.d. condition, simplifying KD. In this case, Vanilla Average Logits should also suffer minimal information loss.
>
> Relaxing the WD assumption in this context would simply add additional notation and unnecessary complexity to an already lengthy formal property statement without providing additional clarity. However, we are open to including a remark summarizing this point for completeness.
>
> While we appreciate the rigorous technical discussion, we respectfully disagree that this makes the theory inadequate. As mentioned earlier, this assumption is pedagogically useful, offering clarity and focus without overcomplicating the analysis. The key takeaway remains the "Delta" of information retention/loss between KD and TAKFL, which stays consistent regardless of the baseline error rate. Even if WD does not hold universally, our theoretical results still provide valuable insights into behavior under idealized conditions, which is a common practice in Federated Learning research. Introducing additional complexity to account for WD variations would make the results more opaque without significantly altering the conclusions.
>
> Finally, we believe that theoretical details should enhance the pedagogical and explanatory value of the theory. Simplifying assumptions, while not always factually accurate, are essential for clarity and understanding. For example, convergence theory often assumes continuously differentiable functions, despite the widespread use of ReLUs and max-pooling in modern architectures. Such theory remains valuable in providing insights into an algorithm’s properties and relative advantages, and we believe the same applies to our theoretical framework.
>
> Once again, we appreciate the reviewer's continued engagement in the discussion and are happy to address any further questions or concerns in the remaining time till end of discussion period by tomorrow on August 13th.

---

> > ### Comment · Reviewer_F7vu · 2024-08-13
> >
> > 1. As state in lines 679-681 in appendix, the proof  heavily rely on task arithmetic property thus the statement "Relaxing the WD assumption in this context would simply add additional notation and unnecessary complexity to an already lengthy formal property statement without providing additional clarity." is wrong.
> >
> > 2. I didn't see "When class labels overlap, the task vectors associated with training data from different prototypes can include vectors in the tangent space of both prototypes’ loss surfaces." why they are in tangent space?  More extreme, what if there is only one class, the distillation of other prototypes to prototypes $j$ ? Without concrete proof or citation, it is not proper to give such a conclusion.
> >
> > 3. Combining my question 1,2 this statement is not proper....“As mentioned earlier, this assumption is pedagogically useful, offering clarity and focus without overcomplicating the analysis. The key takeaway remains the "Delta" of information retention/loss between KD and TAKFL, which stays consistent regardless of the baseline error rate. Even if WD does not hold universally, our theoretical results still provide valuable insights into behavior under idealized conditions, which is a common practice in Federated Learning research.”

---

> ### Author Response · Authors · 2024-08-13
>
> We sincerely appreciate the reviewer's detailed constructive feedback to enhance or theoretical results. To provide a comprehensive response to the reviewer's point, we present the following cases:
>
> ### Case 1 (No Overlap)
> The prototype $i$ and $j$ datasets are disjoint, i.e. $\mathcal{D}^i \cap \mathcal{D}^j = \emptyset$, and $\bigcup_{i=1}^{M} \mathcal{D^i} = \mathcal{D}$. Then, our current proposition holds correct with no changes.
>
> ### Case 2 (Fully Overlap: Identical Class Labels)
> The prototype $i$ and $j$ datasets are fully overlapping, i.e. $\mathcal{D}^i \subseteq \mathcal{D}^j$, meaning that the two prototypes have the same classes $\mathcal{D}_c$, $\forall c \in D$. Then the changes to Proposition 1 and 2 in our theory are detailed in the following.
>
> Consider that in this case, the logits corresponding to $\mathcal{W}^{1,1}$ and $\mathcal{W}^{1,2}$ are the same. As such, $\mathcal{W}^{1,2} \in \mathcal{Q}^1$, and so the propositions change to, trivially:
>
> **Proposition 1** *(Information Loss in VED).* Consider the VED procedure. Consider two device prototypes with a device capacity and solution dimension of $Q^1, Q^2$ and $W^1, W^2$, respectively, and with associated eigenbases $\mathcal{Q}^i, \mathcal{W}^i$. Let the solution set of VED with prototype $i$ as student be $\hat{\Theta}^i_{VED}$ with $\dim(\hat{\Theta}^i_{VED}) = W^{v_i}$ with eigenbasis $\mathcal{W}^{v_i}$. In addition, denote $W^{s,t},\, s,t \in \{1,2\}$ the dimension of the solution set for the student model trained on the data from the teacher device's ensembles. We assume that self-distillation is executed appropriately, e.g., $W^{1,1} = W^1$ and $W^{2,2} = W^2$.
>
> 1. **Case 1:** Assume that $Q^1 = Q^2$ and $W^1 = W^2 = W^{1,2} = W^{2,1}$. Then it holds that, in expectation,
>
>    $$
>    \dim\left(\hat{\Theta}^1_{VED} \cap \left[\mathcal{Q}^1 \setminus \mathcal{W}^1 \right]\right) = 0
>    $$
>
>    This corresponds to the expected capacity of prototype 1 that is taken up for fitting logits that do not fit the data corresponding to prototype 1.
>
> 2. **Case 2:** Assume that $Q^1 > Q^2$ and $W^1 = W^{1,2} > W^2$. Then the same quantity as for Case 1 holds. Moreover,
>
>    $$
>    \dim\left(\hat{\Theta}_{VED} \cap \left[\mathcal{Q}^1 \setminus (\mathcal{W}^1 \cup \mathcal{W}^{1,2}) \right]\right) = 0
>    $$
>
>    This corresponds to the capacity of client 1 that has been allocated but fits, in the model of prototype 1, neither the data of prototype 1 nor the data of prototype 2.
>
> Please note that Proposition 2, that is the guarantees for TAKFL, remains the same.
>
> **...Response continued in the next comment...**

---

> ### Author Response · Authors · 2024-08-13
>
> **...Continued Response...**
>
> ### Case 3 (Partial Overlap)
> The prototype $i$ and $j$ datasets are partially overlapping, i.e. $\mathcal{D}^i \cap \mathcal{D}^j \neq \emptyset$, meaning that the two prototypes overlap on at least one of the classes $\mathcal{D}_c$. Then the changes to Proposition 1 and 2 in our theory are detailed in the following.
>
> In this case, $\mathcal{W}^{1,2} = \mathcal{W}^{1,2}_1 \cup \mathcal{W}_2^{1,2}$, that is, the class labels across data prototypes partially overlap (with $W^{1,2}_1$ the overlapping dimensionality). The Proposition for VED changes to:
>
> **Proposition 1** *(Information Loss in VED).* Consider the VED procedure. Consider two device prototypes with a device capacity and solution dimension of $Q^1, Q^2$ and $W^1, W^2$, respectively, and with associated eigenbases $\mathcal{Q}^i, \mathcal{W}^i$. Let the solution set of VED with prototype $i$ as student be $\hat{\Theta}^i_{VED}$ with $\dim(\hat{\Theta}^i_{VED}) = W^{v_i}$ with eigenbasis $\mathcal{W}^{v_i}$. In addition, denote $W^{s,t},\, s,t \in \{1,2\}$ the dimension of the solution set for the student model trained on the data from the teacher device's ensembles. We assume that self-distillation is executed appropriately, e.g., $W^{1,1} = W^1$ and $W^{2,2} = W^2$.
>
> 1. **Case 1:** Assume that $Q^1 = Q^2$ and $W^1 = W^2 = W^{1,2} = W^{2,1}$. Then it holds that, in expectation,
>
>    $$
>    \dim\left(\hat{\Theta}^1_{VED} \cap \left[\mathcal{Q}^1 \setminus \mathcal{W}^1 \right]\right) = O\left( \frac{(Q^1 - W^1)(W^1)!(Q^1 - W_2^{1,2})!}{Q^1!(W^1)!(Q^1 - W^1)! + Q^1! W_2^{1,2}!(Q^1 - W_2^{1,2})!} \right)
>    $$
>
>    This corresponds to the expected capacity of prototype 1 that is taken up for fitting logits that do not fit the data corresponding to prototype 1.
>
> 2. **Case 2:** Assume that $Q^1 > Q^2$ and $W^1 = W^{1,2} > W^2$. Then the same quantity as for Case 1 holds. Moreover,
>
>    $$
>    \dim\left(\hat{\Theta}_{VED} \cap \left[\mathcal{Q}^1 \setminus (\mathcal{W}^1 \cup \mathcal{W}^{1,2}) \right]\right) = O\left(\frac{(Q^1 - W^1)(W^1!)(W_2^{1,2} - W^2)!}{Q^1! W^1!(Q^1 - W^1)! + Q^1! W^2!(W_2^{1,2} - W^2)!}\right)
>    $$
>
>    This corresponds to the capacity of client 1 that has been allocated but fits, in the model of prototype 1, neither the data of prototype 1 nor the data of prototype 2.
>
> Please note that the presence of overlapping logits $W^{1,2}_1$ does not appear in the result. It is exactly the same as the original Proposition 1 in the original paper, just with $W_2^{1,2}$ in place of $W^{1,2}$.
>
> Please note that Proposition 2, that is the guarantees for TAKFL, remains the same. Indeed, in this case $W^{1,2}$ does not even appear in the original result, and there is no modification needed. Indeed, since the theoretical results consider the complement of $\mathcal{Q}^1$ and all overlapping logits are in $\mathcal{Q}^1$, they do not affect the results.
>
> ### **Conclusion Remark**
> We will supplement these cases to our theoretical results for completeness in the final version of the paper. We hope that this response comprehensively addresses the reviewer's remaining concerns.
>
> Once again, we appreciate the reviewer's continued engagement in the discussion and are happy to address any further questions or concerns in the remaining time till end of discussion period in the next few hours.

---

> > ### Comment · Reviewer_F7vu · 2024-08-14
> >
> > The problem is when class labels overlap  $\mathcal{D}_{t'} \cap \mathcal{D}_t \neq \emptyset$,  task arithmetic property is not held. As state in lines 679-681 in appendix, the proof heavily rely on task arithmetic property thus "Please note that Proposition 2, that is the guarantees for TAKFL, remains the same." is not true....

---

> ### Author Response · Authors · 2024-08-14
>
> We sincerely appreciate the reviewer's response and great effort for the continued engagement in the discussion.
>
> Yes, when the labels overlap (case 2 and 3) the task arithmetic property is not held, but that's not a problem. Please note that for case 2 and 3 the statement is trivial and does not require a proof since we already know the logits are in $\mathcal{Q}^1$. Therefore, as a result no need for task arithmetic property for this case as the statement is trivial and no proof is needed.
>
> Please note that Proposition 1 deals with the information loss of vanilla KD (or average logits ensemble distillation) in device heterogeneous FL, while Proposition 2 states the improvement of TAKFL beyond vanilla KD.
>
> Regarding the improvement of knowledge transfer (TAKFL's improvement) in Proposition 2, under the label overlapping case, the statement becomes trivial. The theorem specifically concerns whether the KD results in parameters that fall within or outside of the space $\mathcal{Q}^1$. When the logits are identical, they are already within that space, meaning there is nothing further to address. As a result, all propositions, which consider the dimensions of the model that do not fit prototype 1's data (i.e., $\mathcal{Q}^1$), remain unaffected by these overlapping logits. This exactly proves our earlier statement that the label overlapping case creates an i.i.d. situation, which can actually make knowledge transfer easier, with the vanilla KD method experiencing minimal information loss. Therefore, as a result TAKFL's guarantee (improvement of knowledge transfer) in proposition 2 becomes trivial.
>
> We would like to also sincerely appreciate the reviewer's great effort in continued engagement in the discussion and providing us feedback. The received feedback indeed helped us to provide a complete theoretical statements for different cases (disjoint and overlapping labels) which we will note this in the final version of the paper.

---

### Official Review · Reviewer_RvJG · 2024-07-12

**Soundness:** 2
**Presentation:** 3
**Contribution:** 3
**Rating:** 5
**Confidence:** 3

**Summary:**

The paper presents a novel framework called TAKFL, which addresses the challenge of transferring knowledge in federated learning across heterogeneous devices, ranging from small IoT devices to large workstations. TAKFL uniquely handles the knowledge distillation by treating the transfer from each device prototype as a separate task, allowing for tailored integration of knowledge to optimize performance. The approach is validated theoretically and through extensive experiments, demonstrating superior results over existing methods across both computer vision and natural language processing tasks.

**Strengths:**

(1) Practical Problem and Innovative Approach
The authors address a significant, real-world challenge in federated learning—knowledge transfer across heterogeneous devices. Their novel framework, TAKFL, innovatively treats each device's knowledge transfer as a separate task, allowing for customized integration. This tailored approach is both practical and theoretically sound, making it a substantial contribution to the field.

(2) Clarity and Organization
The paper is well-structured, facilitating easy understanding of complex concepts and methods. The clear presentation enhances the accessibility of the content, making it easier for readers to grasp the significance of the proposed solution and its impact on the field.

(3) Strong Experimental and Theoretical Support
The authors back their claims with extensive experimental results across multiple tasks and datasets, demonstrating the effectiveness of TAKFL in diverse scenarios. Moreover, the inclusion of theoretical analysis adds depth to the validation, reinforcing the reliability and scalability of their approach. This combination of empirical and theoretical evidence strongly supports the paper's contributions and conclusions.

**Weaknesses:**

(1) [main concern] Strong Dependence on Hyperparameters
The TAKFL framework introduced in the article significantly relies on the setting of hyperparameters, especially during the Task Arithmetic Knowledge Integration process, where the weights for different task vectors are set as hyperparameters and adjusted on a validation set. This might limit the method's generalizability across different real-world applications. To enhance the practicality and robustness of the method, it is recommended that the authors explore more automated hyperparameter optimization strategies to reduce the need for manual tuning and improve the adaptability of the model.

(2) [main concern] Strong Assumptions in the Selection of Public Datasets
The experimental design involves the use of public datasets, such as CIFAR-100 and ImageNet-100, for knowledge distillation. This choice seems to be based on two key assumptions: that the public datasets must exhibit high diversity and that the training data distribution can be approximately considered a subset of the public dataset. These assumptions may not always hold in practical applications, so it is advisable for the authors to thoroughly investigate the actual impact of these choices on model performance in future work. Additional experiments could validate the effectiveness of these assumptions, and considerations of these potential limitations should be explicitly stated in the manuscript.

(3) [minor concern] Quantification of Data Heterogeneity and Hyperparameter Selection
The authors utilize a Dirichlet distribution to quantify Data Heterogeneity, setting $Dir(\alpha)$ at 0.3 and 0.1 to simulate varying degrees of data heterogeneity. However, there is insufficient explanation for the choice of these specific values. To enhance the transparency and reproducibility of the research, it is recommended that the authors provide a detailed rationale behind these parameter choices, based on logic and references to previous studies. Moreover, to give readers a more intuitive understanding of the data distribution differences under different $\alpha$ settings, descriptive statistics or visualizations, such as the distribution of samples across categories, would be helpful.

**Questions:**

See weaknesses

**Limitations:**

N.A

---

> ### Author Rebuttal · Authors · 2024-08-07
>
> Thank you for your invaluable review and pointing out that our work makes a **substantial contribution to the field**. We also appreciate the reviewer for the positive remarks on the **practicality, innovation, and clarity of our TAKFL framework**, as well as the **strong experimental and theoretical support** for our approach.
>
> ### **Response to 1**
> We appreciate the reviewer’s feedback regarding the dependence on hyperparameters. We have designed a simple and efficient automated heuristic method to tune the merging coefficients, detailed in Appendix F.3. Based on our extensive experiments, we observed that larger prototypes benefit more from increasingly skewed merging coefficients, while smaller prototypes perform best with uniformly increasing coefficients. This pattern is intuitive as larger prototypes gain less from smaller ones, especially in high data heterogeneity cases.
>
> Our automated heuristic method randomly instantiates merging coefficients following this observed pattern, generating 30 candidate coefficients that include both uniformly increasing and various degrees of skewed coefficients (detailed implementation is presented in Listing 1 of Appendix F.3). The optimal merging coefficient is determined using maximum performance on a small held-out validation set. As detailed in Appendix F.3, we observed similar performance using this automated heuristic method compared to manual tuning (lines 1143-1145). Furthermore, we used this automated heuristic method for our scalability evaluation experiment in Section 7.3, Figure 3 of our paper. This approach reduces the need for manual tuning, enhancing the adaptability and robustness of the TAKFL framework across different real-world applications.
>
> ### **Response to 2**
> We appreciate the reviewer’s feedback regarding the impact of public datasets. Our public dataset selection mainly follows [1] for a fair and consistent comparison with existing works. To address the reviewer’s concern, we have explored the influence of the public dataset on TAKFL's performance and existing KD-based methods when the public dataset used for server-side distillation is less similar to the private dataset, as detailed in Appendix E.2.
>
> In this experiment, we measured the similarity between different public datasets and the private dataset using CLIP [2]. We fixed the private dataset to CIFAR-10 and used less similar public datasets such as Celeb-A. We observed that the performance of existing methods drastically deteriorates as the similarity between the public and private datasets decreases. In contrast, TAKFL exhibits robustness, suffering much less performance degradation and still achieving SOTA average performance under the same conditions. For instance as presented in Table 10, when using Celeb-A as the public dataset, which is significantly different from CIFAR-10, TAKFL still achieved more than 3% average performance across all prototypes compared to the baselines. This is in contrast to other methods like FedET, which substantially underperformed compared to vanilla FedAvg under the same condition.
>
> This demonstrates TAKFL's practical utility in real-world scenarios where the server typically lacks knowledge of the private datasets to select a closely aligned public dataset for distillation. Additionally, we limited the size of the public dataset to 60,000 or less across all experiments in throughout the paper.
>
> We will further highlight these findings in the final version of our paper to emphasize TAKFL's robustness and practical applicability.
>
> ### **Response to 3**
> We appreciate the reviewer’s feedback and would like to further clarify our data heterogeneity setting. We utilized Dirichlet distribution to quantify data heterogeneity, as it is a commonly used method in the FL literature to simulate varying degrees of data heterogeneity [1,3]. The level of data heterogeneity is controlled by $\alpha$; a lower $\alpha$ indicates higher data heterogeneity and presents a more challenging setup.
>
> The chosen values of $\alpha = 0.3$ for low data heterogeneity and $\alpha = 0.1$ for high data heterogeneity are standard in the FL literature and follow practical experimental design suggestions from [4] (cited in our paper in reference [36]). To give readers a more intuitive understanding of the data distribution differences under different $\alpha$ values, we have plotted visualizations showing how clients' data distributions look for CIFAR-10 using these $\alpha$ values. These visualizations are included in Figure 3 of the PDF rebuttal file. We will include this Figure in the final version of the paper.
>
> ### References
>
> [1] Lin et al. "Ensemble distillation for robust model fusion in federated learning." NeurIPS’20
>
> [2] Radford et al. "Learning transferable visual models from natural language supervision." ICML’21
>
> [3] Li et al. "Federated learning on non-iid data silos: An experimental study." IEEE ICDE (2022).
>
> [4] Morafah et al. "A Practical Recipe for Federated Learning Under Statistical Heterogeneity Experimental Design." IEEE TAI (2023).

---

> > ### Author Response · Authors · 2024-08-12
> > **Kind Reminder: End of Author-Reviewer Discussion Period is Tomorrow**
> >
> > Dear Reviewer RvJG,
> >
> > We sincerely appreciate your extensive efforts in reviewing and commenting on our work. We hope that our rebuttal has comprehensively addressed your comments and concerns.
> >
> > As the author-reviewer discussion period is approaching to its end by tomorrow on August 13th, we kindly ask you to reach out if you have any further questions or points of discussion. Your feedback is highly valued, and we are eager to address any additional remaining concerns you may have.
> >
> > Thank you very much for your attention to this matter.

---

### Official Review · Reviewer_V7t1 · 2024-07-29

**Soundness:** 3
**Presentation:** 3
**Contribution:** 3
**Rating:** 5
**Confidence:** 4

**Summary:**

This paper introduced a KD-based framework (TAKFL) to address the dilution and diversity issues in heterogeneous FL knowledge transfer learning. The TAKFL distills knowledge from prototypes of varying sizes and incorporates a self-regularization to mitigate noise simultaneously, then integrates these separately distilled knowledge by task arithmetic. Empirical evaluations across various CV and NLP datasets demonstrate the framework's effectiveness.

**Strengths:**

1. The paper is well-organized and easy to follow.
2. The paper novelty introduced a theoretical model to illustrate the efficacy of knowledge distillation in heterogeneous FL.
3. The paper proposed a new framework, TAKFL, for considering varying sizes of prototypes with different contributed information, and experiments on different CV and NLP datasets show its effectiveness.

**Weaknesses:**

1. Some baselines are lacking. For example, FedProto [1], which also employs prototypes within device heterogeneous FL, should be included for a more comprehensive comparison.
2. It seems that the proposed method incurs higher time and storage costs, as it requires the independent learning of multiple student models compared to the vanilla methods. The paper should provide an efficiency analysis that compares the proposed method with existing baselines, highlighting both time and storage metrics.
3. It would be better to provide a visualization study for a better understanding of the effectiveness of transfer learning from different prototypes.

[1] Tan, Yue, et al. "Fedproto: Federated prototype learning across heterogeneous clients." Proceedings of the AAAI Conference on Artificial Intelligence. Vol. 36. No. 8. 2022.

**Questions:**

see weakness

**Limitations:**

Yes

---

> ### Author Rebuttal · Authors · 2024-08-07
>
> Thank you for your invaluable review and pointing out the **novelty of our framework and theoretical model** and **effectiveness of our framework on both CV and NLP tasks**.
>
> ### **Response to 1**
> We appreciate the reviewer's suggestion and would like to highlight a few differences about FedProto and our work, TAKFL:
>
> **(1) Problem Formulation:** FedProto's problem formulation aims to optimize local models in FL (Eq. 2 in [1]), whereas our problem formulation follows [2], where we aim to optimize device prototypes’ global models (Eq. 1 in our paper).
>
> **(2) Feature Exchange Requirements:** FedProto requires the exchange of local and global average per-label features between devices and the server, necessitating that local architectures produce the same feature matrix dimensions. However, in more general cases where architectures are from different families, this does not hold. Our experimental setup includes the heterogeneous-family case too.
>
> **(3) Evaluation Metrics:** FedProto uses average local test accuracy as the evaluation metric, while our evaluation metric is device prototypes’ global model test accuracies, following [2].
>
> Given these differences, a direct comparison with FedProto is not entirely feasible or fair. However, considering these factors, we have adopted FedProto’s official implementation code and compared it within our homogeneous-family case experimental setting in Table 1 of our paper. The results, presented in Table 1 of the PDF rebuttal, show that TAKFL consistently outperforms FedProto for both CIFAR-10 and CIFAR-100.
>
> We will incorporate these comparisons into the final version of the paper to provide a more comprehensive evaluation.
>
> ### **Response to 2**
> To address the reviewer’s comment, we would like to mention that the full algorithm details of TAKFL have been presented in Appendix B. All the distillation processes are being performed in parallel and asynchronously (lines 9-10 in Algorithm [1]), and once done, the distillation task vectors are merged.
>
> Therefore, the computation time remains O(1), and the overall computation cost of the entire server-side distillation process is O(M), as device prototypes do not perform KD one at a time but combine information asynchronously into a singular task arithmetic operation (Eq. 8). Since there are M prototypes, the total computation is O(M).
>
> We have further conducted an efficiency analysis on a system with two RTX 3090 GPUs and compared our method with prior works for one epoch knowledge distillation process, including time in seconds, GPU memory, and CPU memory in MB for CIFAR-10 experimental setting in Table 1 of our paper. The results are as follows:
>
> | Baseline	| Time	| CPU Memory | GPU Memory|
> | ----------- | ------- | ---------- | --------- |
> | FedDF   	| 55.24   | 1051.89	| 104.90	|
> | FedET   	| 58.81   | 1113.12	| 107.65	|
> | TAKFL   	| 71.55   | 1080.60	| 266.57	|
>
> As we can see, our method achieves similar CPU memory usage compared to the baselines, with a slight increase in GPU memory and time. The additional 16 seconds incurred time can be attributed to our use of the high-level parallelism interface implementation, `concurrent.futures`, for executing callables asynchronously. Furthermore, in FL, servers are typically equipped with substantial computational resources, including multiple GPUs and extensive memory capacities, to manage the orchestration of many clients. These servers can employ highly efficient parallel asynchronous implementations to avoid the incurred time. Given that the entire distillation process occurs at the server, the computation and storage demands are less of a concern since the server has ample resources.
>
> We will incorporate these results and the efficiency analysis into the final version of the paper.
>
> ### **Response to 3**
> We appreciate the reviewer's suggestion. To better understand the effectiveness of knowledge transfer in TAKFL, we conducted a visualization study using two device prototypes, XS and XL, on the CIFAR-10 dataset. The prototype configurations follow those used in our scalability evaluation in Section 7.3 (more details are available in Appendix D.4). We first pre-trained the two prototypes' global models using FedAvg for 40 communication rounds and then performed knowledge transfer using the TAKFL process for 10 distillation epochs. We have plotted the t-SNE visualizations of both prototypes for FedAvg and TAKFL, included in Figure 1 of the PDF rebuttal file.
>
> As seen in the t-SNE plots, TAKFL demonstrates a substantial enhancement in feature representation for both prototypes compared to FedAvg. The plots show better separation between classes for both XS and XL prototypes. The class clusters are more distinct, suggesting that TAKFL, by effectively transferring knowledge between the two prototypes, yielded a clearer representation of the features, even in the smaller XS prototype. Therefore, the t-SNE visualizations confirm that TAKFL is indeed effective in knowledge transfer, further corroborating its superiority and effectiveness.
>
> We will enrich this visualization study by considering knowledge transfer between different size prototypes and incorporate this study into the final version of our paper.
>
> ### References
> [1] Tan, Yue, et al. "Fedproto: Federated prototype learning across heterogeneous clients." Proceedings of the AAAI Conference on Artificial Intelligence. Vol. 36. No. 8. 2022.
>
> [2] Lin et al. "Ensemble distillation for robust model fusion in federated learning." NeurIPS’20

---

> > ### Author Response · Authors · 2024-08-12
> > **Kind Reminder: End of Author-Reviewer Discussion Period is Tomorrow**
> >
> > Dear Reviewer V7t1,
> >
> > We sincerely appreciate your extensive efforts in reviewing and commenting on our work. We hope that our rebuttal has comprehensively addressed your comments and concerns.
> >
> > As the author-reviewer discussion period is approaching to its end by tomorrow on August 13th, we kindly ask you to reach out if you have any further questions or points of discussion. Your feedback is highly valued, and we are eager to address any additional remaining concerns you may have.
> >
> > Thank you very much for your attention to this matter.

---

> > > ### Comment · Reviewer_V7t1 · 2024-08-13
> > >
> > > Thanks for your response. However, for Q1, FedProto is indeed a method proposed to address the problem of model heterogeneity which utilizes a linear layer to project different feature dimensions into the same for prototype embeddings. Therefore, the targeted problem of FedProto and your method are the same. In addition, the experimental results provided in Table 1 of the PDF rebuttal seem unreasonable: the results of FedProto are even lower than FedAvg, which contradicts the result in FedProto's original paper.

---

> ### Author Response · Authors · 2024-08-13
>
> We appreciate the reviewer’s response and would like to clarify the fundamental distinctions between our methodology and FedProto, particularly in terms of problem formulation and evaluation metrics.
>
> **Problem Formulation:**
> While FedProto does address model heterogeneity, its primary focus is on optimizing heterogeneous local models in FL (Eq. 2 in [1]). In contrast, our method follows [2], where the objective is to optimize the global models of heterogeneous device prototypes (Eq. 1 in our paper). Although both methods tackle device heterogeneity, the specific goals and contexts of optimization differ, leading to very different approaches and outcomes.
>
> **Evaluation Metrics:**
> FedProto evaluates performance using average local test accuracy, while our evaluation metric is the global model test accuracy of device prototypes, as outlined in [2].
>
> Furthermore, we would like to note that our experimental setup significantly differs from the original setting in FedProto’s paper.  As a results, these significant fundamental differences in both problem formulations and evaluation metrics makes direct comparisons between FedProto and our method challenging and unfair and misleading.
>
> However, despite these fundamental differences we have adopted the FedProto’s official implementation code and conducted experiments within our homogeneous-family experimental setting (Table 1 of our paper), and presented the results in Table 1 of rebuttal PDF file. The differences in results, such as FedProto performing lower than FedAvg, could be due to these fundamental differing contexts and setups.
>
> In conclusion, while we agree with the reviewer that FedProto is a model heterogeneous FL method, it is important to note that FedProto is a "personalized FL" model-heterogeneous federated learning per its problem formulation in Eq. 2 of [1]. However, our work is a "global FL" KD-based device heterogeneous federated learning per our problem formulation in Eq. 1 of our paper, following [2]. This distinction is crucial in understanding the differences in performance outcomes.
>
> We sincerely appreciate the reviewer's constructive feedback and suggestions. If there are any further concerns or questions, we would be grateful if the reviewer could kindly elaborate so that we can provide additional clarity.
>
> We appreciate the reviewer's continued engagement and are happy to address any further questions or concerns as the discussion period concludes in the next few hours.
>
> ### References
>
> [1] Tan, Yue, et al. "Fedproto: Federated prototype learning across heterogeneous clients." Proceedings of the AAAI Conference on Artificial Intelligence. Vol. 36. No. 8. 2022.
>
> [2] Lin et al. "Ensemble distillation for robust model fusion in federated learning." NeurIPS’20

---

### Author Rebuttal · Authors · 2024-08-07

We sincerely appreciate all the reviewers' efforts in reviewing and commenting on our work. We are particularly grateful for the positive feedback highlighting the following aspects:

* **The novel theoretical model and framework illustrating the efficacy of knowledge distillation in heterogeneous federated learning (Reviewer V7t1, Reviewer RvJG, Reviewer ME5v).**
* **Effective handling of challenges in transferring knowledge across heterogeneous devices in federated learning and improving existing knowledge distillation methods (Reviewer RvJG, Reviewer F7vu).**
* **Strong experimental validation on CV and NLP tasks with various datasets, demonstrating superior results over existing methods (Reviewer V7t1, Reviewer RvJG, Reviewer F7vu).**
* **Addressing significant real-world challenges in federated learning with meaningful theoretical and practical contributions (Reviewer V7t1, Reviewer RvJG, Reviewer ME5v).**
* **Clear and well-organized presentation, making the paper easy to follow and understand (Reviewer V7t1, Reviewer RvJG, Reviewer F7vu, Reviewer ME5v).**

In summary, we have introduced a novel framework, TAKFL, to address the fundamental limitations of existing knowledge distillation (KD) methods in the *underexplored* diverse device heterogeneous federated learning environments, ranging from small IoT devices to large workstations. Our framework effectively distills the unique knowledge of each device prototype's ensembles by treating them as separate tasks and adaptively integrates them into the student model via a novel task arithmetic approach. Unlike existing methods, TAKFL avoids dilution and information loss, providing customized knowledge integration for each student model based on the knowledge quality and helpfulness of each prototype's ensembles and its intrinsic capacity. Furthermore, in response to the issues associated to noisy and unsupervised ensemble distillation, we have incorporated a novel KD-based self-regularization loss. Our work makes the following key contributions:

* **Framework:** Introducing a new framework to address the fundamental limitations of existing methods in diverse device heterogeneous FL called TAKFL.
* **Theoretical Contribution:** Presenting the first theoretical framework using the concept of capacity dimension to understand the effectiveness of KD in device heterogeneous FL.
* **Experimental Validation:** Conducting comprehensive experimental evaluations for both CV and NLP tasks with various datasets, architectures, and levels of data heterogeneity, demonstrating the effectiveness of our method and achieving state-of-the-art performance by significantly outperforming existing methods. Additionally, we include a new scalability experiment encompassing devices from XXS to XXL to further demonstrate the efficacy of our method.

Our extensive experiments show that TAKFL not only outperforms existing methods but also offers a novel, simple, and efficient solution to a significant practical challenge. Furthermore, to the best of our knowledge, we are the first to introduce the notion of "task vector" in FL to enhance knowledge transfer across diverse heterogeneous device prototypes. We have derived new consistent theoretical and experimental findings detailed in Appendix D.1.1. We believe our work makes a significant contribution to the FL field (as pointed out by Reviewer RvJG), addresses an important practical challenge (as highlighted by Reviewer ME5v), and establishes a new benchmark. We believe that the impact of our method extends beyond specific datasets and prototype sizes, providing a versatile solution for diverse device heterogeneous federated learning applications.

We appreciate the reviewers' constructive comments and have strived to comprehensively address them. To this end, we have provided at least one additional experiment, table, and/or figure for each reviewer:

* In response to Reviewer V7t1's comments #1 and #3, we have provided Table 1 and Figure 1 in the PDF rebuttal.
* In response to Reviewer RvJG's comment #3, we have provided Figure 3 in the PDF rebuttal.
* In response to Reviewer F7vu's comment #2, we have provided Table 2 in the PDF rebuttal.
* In response to Reviewer ME5v's comments #2 and #3, we have provided Figure 2 and Table 3 in the PDF rebuttal.

The PDF rebuttal file containing the associated figures and tables is uploaded with this message.

Once again, we sincerely appreciate the reviewers' time and effort in reviewing our work. If our response comprehensively addresses the reviewers' comments, we kindly ask the reviewers to consider raising their scores. We are happy to address any remaining comments or concerns during the author-reviewer discussion period.

Thank you!

---

### Author Response · Authors · 2024-08-10
**Kind Reminder: End of Author-Reviewer Discussion Period Approaching**

Dear Reviewers,

We sincerely appreciate your extensive efforts in reviewing and commenting on our work. We hope that our rebuttal has comprehensively addressed your comments and concerns.

As the author-reviewer discussion period is approaching to its end on August 13th, we kindly encourage you to reach out if you have any further questions or points of discussion. Your feedback is highly valued, and we are eager to address any additional remaining concerns you may have.

Thank you very much for your attention to this matter.

Anonymous Authors

---

### Decision · Program_Chairs · 2024-09-25

**Decision:**

Accept (poster)

**Comment:**

The paper is of high quality, and there is not objection from four reviewers. It requires minor refinements based on the reviewers' comments.